# CRISPRi–TnSeq maps genome-wide interactions between essential and non-essential genes in bacteria

Bimal Jana[1,7], Xue Liu[2,3], Julien Dénéréaz ®[3], Hongshik Park[1], Dmitry Leshchiner[1], Bruce Liu[1], Clément Gallay ®[3], Junhao Zhu[4], Jan-Willem Veening ®[3] ✉ & Tim van Opijnen ®[5,6] ✉

Genetic interactions identify functional connections between genes and pathways, establishing gene functions or druggable targets. Here we use CRISPRi–TnSeq, CRISPRi-mediated knockdown of essential genes alongside TnSeq-mediated knockout of non-essential genes, to map genome-wide interactions between essential and non-essential genes in *Streptococcus pneumoniae*. Transposon-mutant libraries constructed in 13 CRISPRi strains enabled screening of ~24,000 gene pairs. This identified 1,334 genetic interactions, including 754 negative and 580 positive interactions. Network analyses show that 17 non-essential genes pleiotropically interact with more than half the essential genes tested. Validation experiments confirmed that a 7-gene subset protects against perturbations. Furthermore, we reveal hidden redundancies that compensate for essential gene loss, relationships between cell wall synthesis, integrity and cell division, and show that CRISPRi–TnSeq identifies synthetic and suppressor-type relationships between both functionally linked and disparate genes and pathways. Importantly, in species where CRISPRi and Tn-Seq are established, CRISPRi–TnSeq should be straightforward to implement.

A variety of genome-wide functional genomics tools including transposon insertion sequencing (Tn-Seq)[1] and CRISPR interference (CRISPRi)[2] have been used to associate (bacterial) genetic elements with a specific perturbation. These data in turn inform on leads for gene function, including genes associated with antibiotic transport, cell wall synthesis, membrane integrity and global responses to overcome stress[3–5]. While either tool has its advantages, uncovering which genetic elements in a genome work together (that is, interact) and how they generate a phenotype remains a key challenge.

Overall, two random genes in a genome are hypothesized to work independently and not interact. This means that two simultaneously knocked out (Tn-Seq), or knocked down (CRISPRi), genes should have a combined phenotype effect. However, a deviating phenotype (for example, growth) is categorized as a negative interaction if growth is significantly slower, or positive if growth is faster than expected from the multiplicative function. One of the most comprehensive genetic interaction networks has so far been constructed through synthetic genetic array (SGA) analyses[6] for *Saccharomyces cerevisiae*, leading to

[1]Department of Biology, Boston College, Chestnut Hill, MA, USA. [2]Department of Pathogen Biology, Base for International Science and Technology Cooperation: Carson Cancer Stem Cell Vaccines R&D Center, International Cancer Center, Shenzhen University Health Science Center, Shenzhen, China. [3]Department of Fundamental Microbiology, University of Lausanne, Lausanne, Switzerland. [4]CAS Key Laboratory of Pathogenic Microbiology and Immunology, Institute of Microbiology, Chinese Academy of Sciences, Beijing, China. [5]Broad Institute of MIT and Harvard, Cambridge, MA, USA. [6]Boston Children's Hospital, Division of Infectious Diseases, Harvard Medical School, Boston, MA, USA. [7]Present address: Center for Genomic Medicine, Massachusetts General Hospital, Boston, MA, USA. ✉e-mail: jan-willem.veening@unil.ch; tim.vanopijnen@childrens.harvard.edu

the development of hierarchical genetic models of cellular functions[7]. Genetic interactions have also been mapped in *E.coli* using SGA[8], and in mammalian cells by using RNA interference[9] and by dual-CRISPRi[10]. While these and other approaches are powerful and highlight the importance of generating interaction networks, they all have their pros and cons, and some are only available for a select number of model strains and organisms, and/or may not be easily scalable[11,12].

Transposon insertion sequencing-based methods including Tn-Seq have enabled, on a genome-wide scale, the identification of essential genes, the importance of individual genes to specific phenotypes, and genetic interaction mapping in non-model microorganisms[1,3,13–15]. This has led to the identification of new gene functions including regulatory relationships[3,13], pathway and transporter redundancies[16], membrane integrity[17] and cholesterol processing[18,19]. A key limitation of deletion arrays or Tn-based approaches is that they cannot directly sample essential genes that are inviable[20]. To bypass this issue, chemical-genetics approaches have been used[21], albeit with limited success as compounds are only available against some essential gene products and often trigger off-target effects. CRISPRi has made studying essential gene function in bacteria possible[22–24], which has led to uncovering new genotype–phenotype relationships including for genes involved in cell wall synthesis, division and competence[22,25,26]. While CRISPRi often needs extensive development to implement it in a new strain or species, and Tn-Seq cannot sample essential genes, we here exploit the advantages of either approach by developing and implementing CRISPRi–TnSeq. Genome-wide interactions are mapped for 13 (strain-dependent) essential genes in *Streptococcus pneumoniae* and by performing extensive analyses, chemical genetics, knockout and microscopy studies, the applicability of CRISPRi–TnSeq in constructing genetic interaction networks is presented.

## Results

### Essential–non-essential interaction mapping with CRISPRi–TnSeq

CRISPRi–TnSeq identifies essential–non-essential genetic interactions by simultaneous knockdown of an essential gene (CRISPRi) and knockout of a non-essential gene (Tn-Seq). CRISPRi–TnSeq identifies synthetic and suppressor relationships between essential and non-essential genes genome-wide and follows 5 main steps (Fig. 1a). To develop and optimize CRISPRi–TnSeq, we selected CRISPRi strains for 12 essential and 1 strain-dependent-essential (*clpP*) genes in *Streptococcus pneumoniae* involved in different biological processes (Fig. 1b and Supplementary Table 1). These CRISPRi strains are adopted from our previous study, which demonstrated no leakiness[25]. Here we constructed Tn-mutant libraries for individual CRISPRi strains and confirmed functionality of CRISPRi in Tn mutants by growing six libraries with or without isopropyl β- d-1-thiogalactopyranoside (IPTG), followed by growth (Extended Data Fig. 1) and qPCR (Source Data Fig. 1c) analysis. Increased growth inhibition and target knockdown (Fig. 1c) with increased IPTG (except, as expected, conditionally essential *clpP*) confirm successful and tunable CRISPRi knockdown of a target in a Tn-mutant library.

### CRISPRi–TnSeq identifies mechanistically linked genes

To identify genetic interactions, Tn-mutant libraries were grown with or without IPTG (Supplementary Table 2). Fitness without IPTG represents non-essential gene disruption, while fitness with IPTG is a combination of non-essential gene disruption and essential gene knockdown. When fitness with IPTG ($W_{IPTG}$) is significantly lower or higher than without IPTG ($W_{noIPTG}$), this indicates a deviation from the expected multiplicative fitness, representing a negative or positive genetic interaction, respectively. Tn libraries in 13 CRISPRi strains (Fig. 1b and Supplementary Table 2) enabled screening of ~24,000 gene–gene pairs, identifying 1,334 significant genetic interactions, including 754 negative and 580 positive interactions (Extended Data Fig. 2a and Source Data Fig. 2a). Permutation testing shows enrichment of negative interactions, which is similar to synthetic genetic array screens of *E. coli*[8]. We have previously shown that the fitness effect of a gene disruption increases with increasing stress[4]. To confirm this and to simultaneously assess reproducibility, CRISPRi–TnSeq was performed at two sub-inhibitory IPTG concentrations (Supplementary Table 2). Indeed, more interactions were identified at the highest concentration, while on average there is a 65% overlap in genetic interactions between IPTG concentrations (Supplementary Table 4 and Fig. 1d).

All genetic interactions were mapped on a circular genome (Extended Data Fig. 3a). Genes adjacent to oriC (SPD2014–SPD2064) possess a significantly higher number of interactions (Extended Data Fig. 3b and Source Data Extended Data Fig. 3b), which is probably due to increased gene dosage to overcome replication stress[27]. Overall, negative interactions dominate positive interactions in all categories except for translation (Extended Data Fig. 4a, inset), suggesting that shutting down of the most energy-costly processes helps cells cope with stress. Furthermore, genetic interactions with *atpF* are mostly positive (Source Data Fig. 2a), which fits with findings that disruption of ATP homoeostasis can buffer against stress[4]. Gene set enrichment analysis shows that knockdown of an essential gene often leads to genetic interactions with functionally linked non-essential genes. For example, DNA repair and lipid metabolism genes are enriched in the *parC* and *fabH* knockdown datasets, respectively (Extended Data Fig. 4b and Source Data Extended Data Fig. 4b). Moreover, these functional connections are also present when individual genes are considered at the pathway level, including genes involved in purine biosynthesis and purine degradation (Extended Data Fig. 5). Overall, these results suggest that CRISPRi–TnSeq can identify non-essential genes that are mechanistically linked to corresponding essential genes.

---

**Fig. 1 | CRISPRi–TnSeq identifies essential–non-essential genetic interactions at genome scale. a**, Schematic overview of CRISPRi–TnSeq: (1) An *S. pneumoniae* D39 CRISPRi strain carries an essential gene-specific sgRNA with a P3 constitutive promoter and *dcas9* under the P*lac* inducible promoter blocked by LacI expressed constitutively from the PF6 promoter[74]. IPTG triggers dCas9 expression resulting in tunable essential gene knockdown; (2) In a CRISPRi strain, a Tn-mutant library is constructed as described previously[13]; (3) A library is grown with or without IPTG; (4) Tn-Seq sample preparation sequencing and analysis[1,13,20] determines the fitness effect for each non-essential gene with or without essential gene knockdown; (5) Interactions are scored as a significant deviation from the expected multiplicative function of the fitness of the knocked-out non-essential and knocked-down essential gene, enabling the mapping of a genetic interaction network. Note that CRISPRi knockdown of an essential gene reduces the growth of the entire mutant library in an equal manner, except for those that interact, which are a small minority population. As a result, the relative fitness of non-interacting individual mutants remains similar in the absence and presence of IPTG. In contrast, if an essential and non-essential gene pair genetically interact, a transposon's frequency will significantly decrease or increase, representing a negative and positive interaction, respectively.
**b**, Tn-mutant libraries were constructed in 13 CRISPRi strains. **c**, Sub-inhibitory IPTG concentrations inhibit growth (top) and knockdown gene expression (bottom), validating the functionality of the CRISPRi system. Cell density was measured after ~3 h of growth. Green, blue and yellow bars represent no, low and high IPTG, respectively. Average of $n = 3$ biological replicate growth OD values and corresponding expression changes are plotted; independent repetition (2) of the experiment showed similar results. **d**, Example volcano plots for *fabH*-CRISPRi. Tn-mutant libraries were grown at 0, 15 and 20 μM IPTG, followed by CRISPRi–TnSeq analysis. Increased knockdown by high IPTG identifies an increased number of genetic interactions (coloured by bioprocesses), with an average of 65% overlap at both IPTG concentrations for all CRISPRi strains and the highest overlap (82%) for *fabH*. Hypergeometric test with 'lower.tail=FALSE' showed that overlap is significant with $P = 0.012$. Source data are presented in Supplementary Tables 1 and 4, and Source Data Figs.

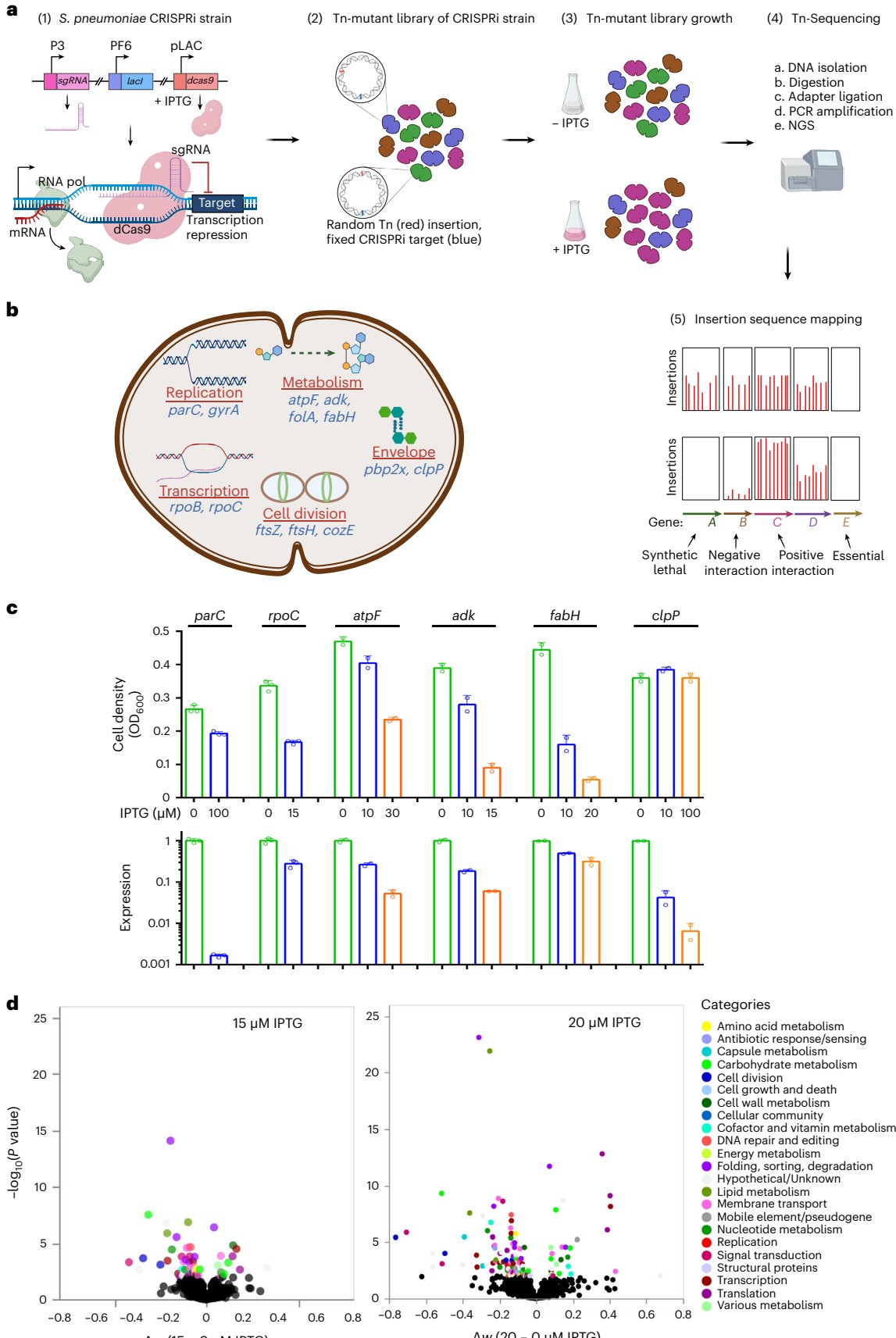

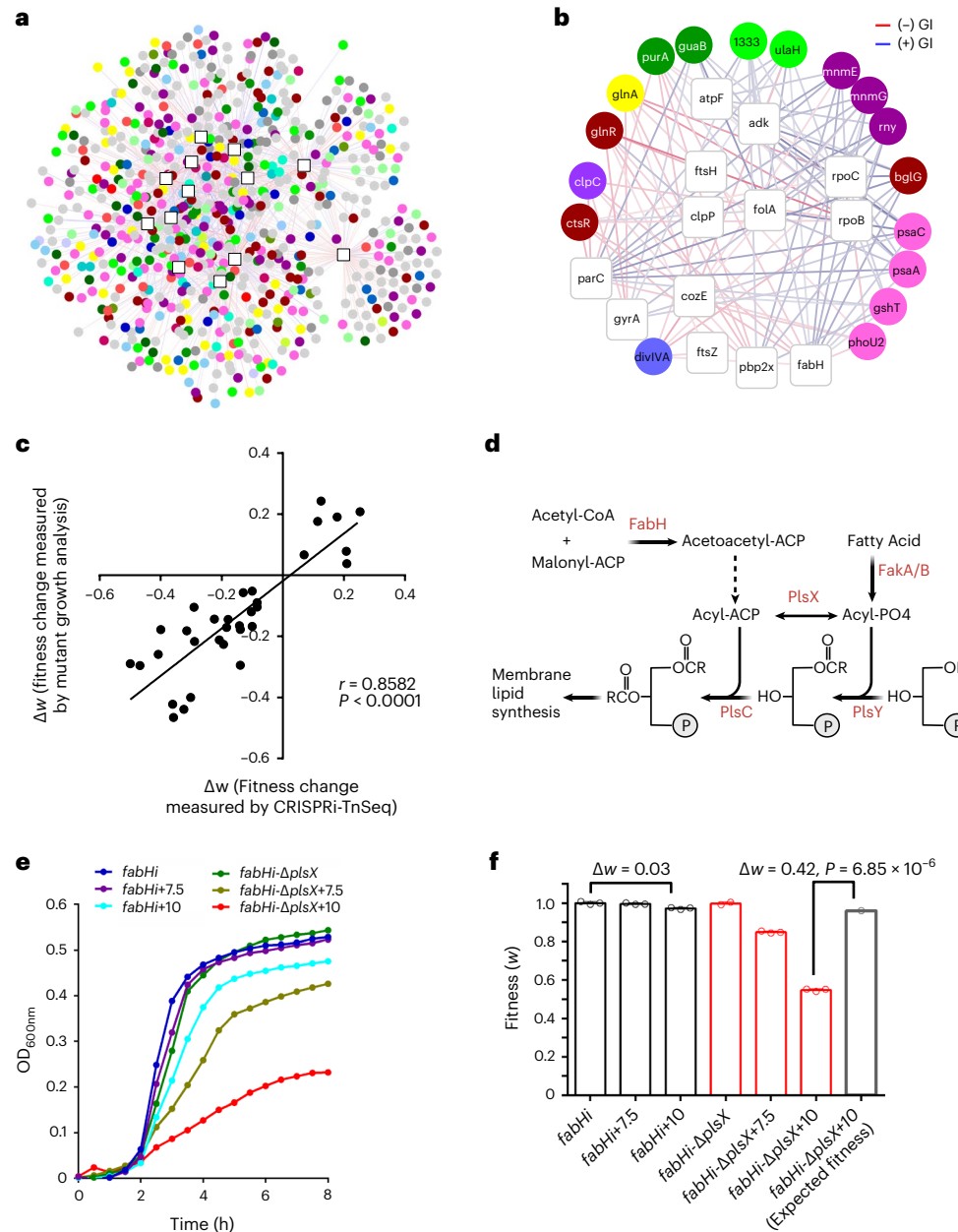

**Fig. 2 | A CRISPRi–TnSeq essential–non-essential genetic interaction network. a**, The genome-wide CRISPRi–TnSeq network contains 13 (strain-dependent) essential and 853 non-essential genes, and 1,334 genetic interactions. CRISPRi targets are presented as white squares and non-essential genes as circles coloured on the basis of their bioprocess (colour coding as in Fig. 1e). Coloured edges refer to negative (red) or positive (blue) interactions. The intensity of the edge colour is proportional to the strength of the interaction z-score ranging from −10 to +10. **b**, A subnetwork highlights a set of 17 non-essential genes interacting with over half of the targets. Note, a *bglG* knockout is probably reducing the phosphorylation and transport of the IPTG inducer. It is thereby impairing the CRISPRi system and should not be considered a suppressor. **c**, Thirty-two genetic interactions were validated by constructing non-essential gene deletion mutants (Source Data Fig. 2c) in the corresponding CRISPRi strains. Fitness (that is, growth) of deletion mutants is compared with corresponding CRISPRi–TnSeq fitness, indicating high reproducibility (two-tailed Pearson correlation coefficient $r = 0.85$, $P = 3.4 \times 10^{-10}$). **d**, A part of the

membrane lipid biosynthesis pathway highlights the relationship of *plsX* and *fabH* and illustrates how *plsX* can compensate for *fabH* knockdown by increasing flux between Acyl-ACP and Acyl-PO$_4$. This means that when FabH is knocked down, acyl-ACP levels are reduced which can be supplemented through PlsX's ability to generate acyl-ACP through the reversible transfer of acyl-PO$_4$. **e**, Growth curves of CRISPRi-*fabH* (*fabHi*) and *fabHi*-Δ*plsX* show, as predicted from CRISPRi–TnSeq, a significant interaction reflected in a growth defect in *fabHi*-Δ*plsX* in the presence of IPTG. Average of $n = 3$ biological replicate growth curves is plotted with standard error; independent repetition (3) of the experiment showed similar results. **f**, Growth curves of *fabHi* and *fabHi*-Δ*plsX* transformed into a bar graph showing their observed relative fitness under each condition, as well as the expected multiplicative fitness of the double mutant. Observed fitness is significantly lower than the expected multiplicative fitness (paired *t*-test $P = 6.85 \times 10^{-6}$), indicating a strong *fabH-plsX* negative interaction. Average of $n = 3$ biological replicate growth fitness values is plotted. Source data are presented in Source Data Fig.

To further confirm that CRISPRi–TnSeq identifies biologically relevant interactions, we determined the overlap between CRISPRi–TnSeq and antibiotic-TnSeq datasets. Tn-mutant libraries were grown in the presence of an antibiotic (for example, rifampicin) that impairs the function of the corresponding CRISPRi-target essential gene product (for example, *rpoC*), followed by Tn-Seq analysis[4] (Supplementary Table 3). As functionally linked gene sets clustered[7], profile similarity analysis provides an opportunity to validate the dataset. Hierarchical

clustering of 13 CRISPRi–TnSeq and 8 antibiotic-TnSeq datasets (Source Data Fig. 2a and Source Data Extended Data Fig. 6) placed target-specific CRISPRi–TnSeq and antibiotic-TnSeq datasets closely together (for example, *rpoC*-CRISPRi and rifampicin-TnSeq) (Extended Data Fig. 6). Furthermore, clustering also identifies modules of non-essential genes involved in a specific bioprocess (Extended Data Fig. 6b). For example, *smc-scpA-scpB* form a cluster that has not been previously identified in *S. pneumoniae*, although this was expected as bacterial condensins consist of the SMC-ScpA-ScpB triad[28]. Clustering of fitness and genetic interaction data can thus be leveraged to identify functional modules.

## *S. pneumoniae* contains pleiotropic genes that modulate stress

Genetic interactions were assembled into a network containing 853 non-essential genes, which were ranked on the basis of their total number of interactions (Fig. 2a). Of the highest-ranked genes, 17 have interactions with more than half of the targeted essential genes (Fig. 2b). We validated 7 of these interactions (Fig. 2c and Source Data Fig. 2c), confirming that these non-essential genes interact with multiple essential genes and can thereby modulate various bioprocesses. Seven of these genes are dominated by negative interactions and are involved in stress responses (*ctsR* and *glnR*, *clpC*), division (*divIVA*) and metabolism (*purA*, *phoU* and *glnA*). Knockout of these genes sensitizes the organism to knockdown of essential genes from a variety of bioprocesses, making them the most pleiotropic genes in the network. A canonical pleiotropic example is heat shock protein 90 of *Saccharomyces cerevisiae*, a chaperon that maintains signalling networks and corresponding bioprocesses[29]. The pleiotropic genes identified here seem to introduce robustness into the system, protecting against stress and making them potential drug-sensitizing targets. The 10 remaining high-degree genes have mostly positive interactions, displaying suppressor phenotypes. As they are mainly involved in protein synthesis (*mnmE*, *mnmG* and *rny*) and transport (*psaA*, *psaC* and *gshY*), their knockouts can slow down growth and thereby mask the effect of essential gene knockdown, which is similarly reflected in antibiotic tolerance of slow-growing mutants[4].

## Essential–non-essential interaction network validation

To further evaluate the CRISPRi–TnSeq dataset, 32 genetic interactions were validated by constructing non-essential gene deletion mutants in the background of specific CRISPRi knockdown strains (Supplementary Table 1 and Source Data Fig. 2c). Growth rates were measured for each strain with or without IPTG, thereby enabling calculations of each strain's fitness affected by the non-essential gene deletion (−IPTG) and simultaneous CRISPRi-mediated target knockdown (+IPTG) (Source Data Fig. 2c). Note that multiple IPTG concentrations are used to show the sensitivity and robustness of an interaction. Genetic interactions of essential and non-essential genes determined in individual growth studies correlate well with the interactions of corresponding gene pairs identified in CRISPRi–TnSeq (Fig. 2c and Source Data Fig. 2c). We further explored several interactions to investigate underlying biological mechanisms, including a negative interaction between *fabH*, which is involved in membrane lipid biosynthesis[30], and *plsX*, which catalyses the reversible formation between acyl-phosphate and acyl-ACP[31] (Fig. 2d).

To validate this interaction, we assessed growth of *fabH*i (*fabH*-CRISPRi) and *fabH*i-Δ*plsX* with or without IPTG, confirming a significant growth defect in *fabH*i-Δ*plsX* with IPTG (Fig. 2e). Converting growth curves to fitness further highlights the strong impact of *fabH* knockdown on the growth of Δ*plsX* (Fig. 2f), confirming the CRISPRi–TnSeq-predicted negative genetic interaction, which stems from the ability of PlsX to compensate for a loss in FabH functionality by diverting acyl-ACP production (Fig. 2d).

## Septal and peripheral peptidoglycan syntheses need to be balanced

Cell division is a complex process requiring multiple pathways including cell wall (CW) synthesis, which is subdivided into peripheral (elongation) and septal (division) peptidoglycan (PG) synthesis. While PBP2b and PBP1a drive peripheral PG, PBP2x is key for septal PG synthesis[32]. How septal and peripheral PG syntheses at these two different cellular locations are orchestrated is unclear. By performing CRISPRi–TnSeq, we find that PG precursor synthesis genes *murM*, *murN* and *murA1* become important when *pbp2x* is knocked down (Fig. 3a). In contrast, the negative effect of *pbp2x* knockdown can be suppressed by knocking out PG synthesis genes *pbp1a*, *pbp1b* or *pbp2a* (Fig. 3a). These data suggest that the rates of septal vs peripheral PG synthesis need to be balanced to ensure proper cell shape and division. Our data also show that this rebalancing is mediated through *pbp3* which has a strong negative interaction with *pbp2x* (Fig. 3a). *pbp3* is a class C PBP that hydrolyses D-Ala from the pentapeptide and delocalizes from the future division site, thereby spatially directing availability of PG precursors towards the septum[33]. In the absence of PBP3, active PG precursors remain spread out across the membrane, reducing the effective concentration at the septum, while targeting *pbp2x* further disturbs this imbalance and exacerbates the problem. Furthermore, the importance of balanced PG synthesis and the connection with cell division is highlighted by the strong negative interactions of *pbp2x* with *sepF* and *pvaA*, and *ftsZ* with *pbp3*, *sepF* and *pvaA* (Fig. 3a). SepF has been shown to anchor and activate Z-ring assembly in *Bacillus subtilis*[34], while *pvaA*, a lysozyme-like protein, is important for proper localization of FtsZ[35]. The interactions with *sepF* confirm that PvaA has a similar function of Z-ring assembly activation in *S. pneumoniae*. Overall, these negative interactions among PG synthesis and Z-ring assembly genes demonstrate that both processes are intimately coordinated to ensure successful cell division.

## Cyclic-di-AMP helps maintain cell survival upon CW insult

*pde1* encodes a phosphodiesterase that hydrolyses the second messenger cyclic-di-AMP (c-di-AMP). Δ*pde1* can suppress the effects of *pbp2x* knockdown (Fig. 3a,b), leading us to hypothesize that Δ*pde1* leads to increased intracellular c-di-AMP, which maintains growth and morphology under reduced levels of PBP2x. Indeed, both intracellular and extracellular c-di-AMP levels are high in Δ*pde1* (Fig. 3c and Extended Data Fig. 7c). However, while increased c-di-AMP levels in the growth medium can partially rescue *pbp2x* knockdown (Extended Data Fig. 7d), high intracellular c-di-AMP levels are needed to buffer against the inhibition of septal PG synthesis, possibly by decreasing turgor[36]. Fluorescence microscopy shows that *pbp2x* knockdown leads to enlarged cells at a

**Fig. 3 | A *pde1* knockout can buffer the negative phenotypic effects of *pbp2x* knockdown. a**, A network highlighting the genetic interactions between *pbp2x* and non-essential genes involved in directly related processes including CW synthesis (green) and division (blue), as well as interactions with hypothetical genes (grey). Additional essential genes that interact with the same non-essential genes as *pbp2x* are highlighted. CRISPRi-targeted genes are presented as white nodes and non-essential genes as circles coloured on the basis of their bioprocess (colour coding as in Fig. 1e). Negative and positive genetic interactions (GI) are presented as red and blue edges, respectively. The intensity of the edge colour is proportional to the strength of the interaction *z*-score ranging from −10 to +10. **b**, Growth curves of *pbp2x*-CRISPRi (*pbp2xi*) and *pbp2xi*-Δ*pde1* without (violet

and green) and with (cyan and red) 20 μM IPTG. The average of *n* = 3 replicate growth curves is plotted with standard error; independent repetition (3) of the experiment showed similar results. **c**, c-di-AMP levels in *pbp2xi* and *pbp2xi*-Δ*pde1* with and without 20 μM IPTG. The average of *n* = 2 biological replicate measurements is plotted; independent repetition (2) of the experiment showed similar results. **d,e**, Fluorescence microscopy images of *pbp2xi* (**d**) and *pbp2xi*-Δ*pde1* (**e**), with (bottom) or without (top) 100 μM IPTG. Representative phase contrast, Nile red (red) and DAPI (blue)-stained micrographs are presented with a 2 μm scale bar; independent repetition (2) of the experiment showed similar results. A *pde1* deletion suppresses the deleterious impact of *pbp2x* knockdown by increasing c-di-AMP levels, which restores morphology.

coccus-to-rod transition state with defective septa (Fig. 3d), while Δ*pde1* rescues wild-type-like morphology (Fig. 3e), albeit with shorter chain lengths, suggesting that determinants such as LytB are altered[37,38]. Furthermore, we confirmed the interaction between the c-di-AMP synthase *cdaA* and *pde1* (Extended Data Fig. 8a–c). As a secondary messenger, c-di-AMP regulates multiple biological processes[39–41]. However, its role in CW synthesis is unexplored in *S. pneumoniae*. Intracellular c-di-AMP

directly interacts with effector protein CabP, which reduces the activity of the TrkH potassium transporter, thereby inhibiting $K^+$ uptake[42] and affecting cellular turgor[36]. We hypothesized that increased c-di-AMP levels in Δ*pde1* lead to reduced intracellular $K^+$, which in turn reduces cellular turgor and thereby maintains growth and morphology upon *pbp2x* knockdown. Indeed, only when extracellular $K^+$ is increased from 52 to 100 mM does Δ*pde1* fail to rescue *pbp2x* knockdown (Fig. 4a).

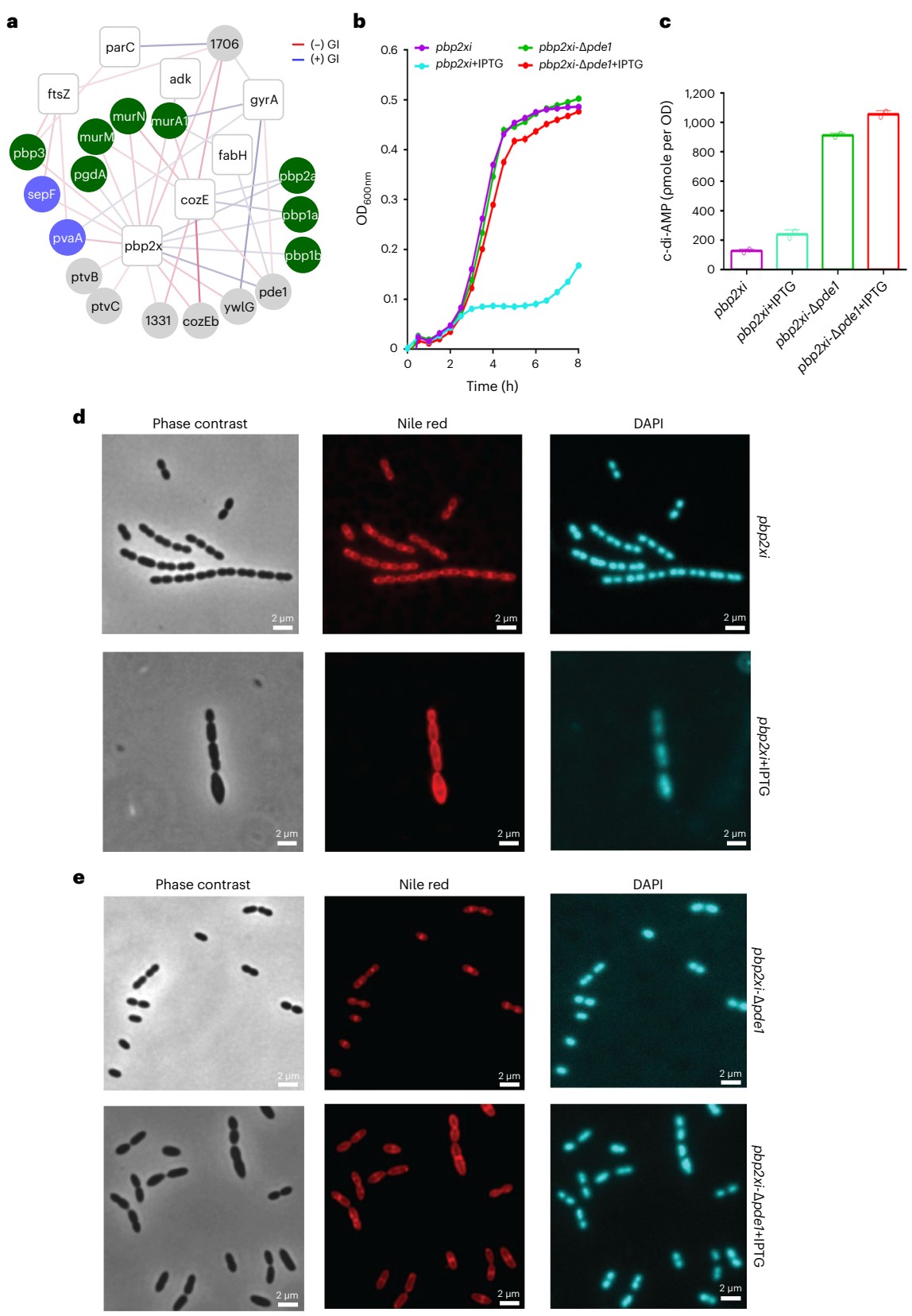

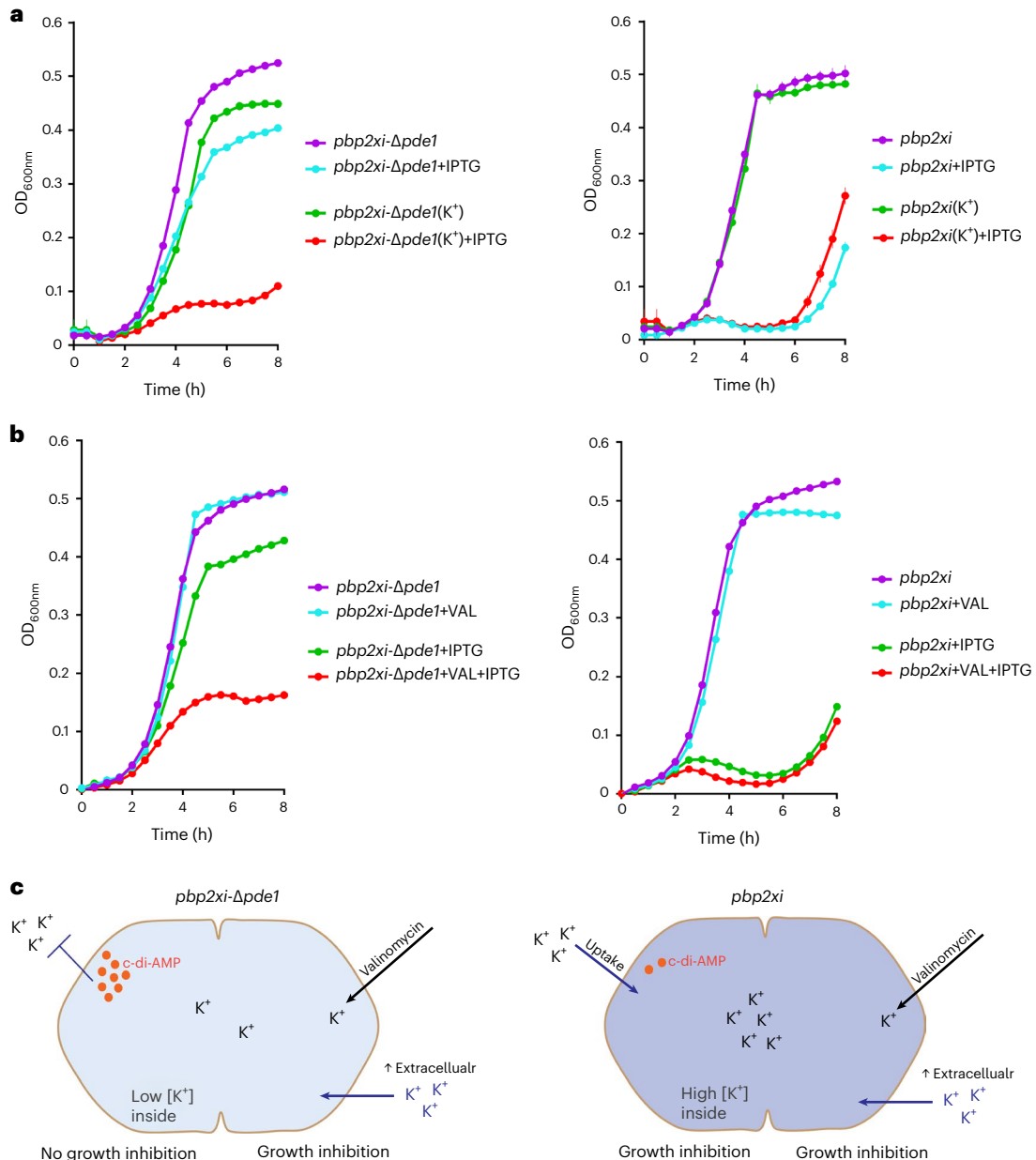

**Fig. 4 | *pbp2x* knocked-down cells can be rescued by decreasing K⁺ uptake and turgor. a**, Growth curves of *pbp2xi* (right) and *pbp2xi-Δpde1* (left) with and without 30 μM IPTG and/or additional 100 mM potassium (K⁺). While Δ*pde1* can mask the negative fitness effect of *pbp2x* knockdown (see Fig. 3b), the addition of K⁺ nullifies this masking effect. The average of $n = 3$ biological replicate growth curves is plotted with standard error; independent repetition (2) of the experiment showed similar results. **b**, Growth curves of *pbp2xi* (right) and *pbp2xi-Δpde1* (left) with or without IPTG and/or 0.5 μg ml⁻¹ valinomycin, a K⁺

carrier. The average of $n = 3$ biological replicate growth curves is plotted with standard error; independent repetition (2) of the experiment showed similar results. **c**, Modulation of the K⁺ concentration in *pbp2xi* (right) and *pbp2xi-Δpde1* (left). c-di-AMP reduces the activity of the TrkH potassium transporter, thereby inhibiting K⁺ uptake and affecting cellular turgor. Increased c-di-AMP levels in *pbp2xi-Δpde1* in the presence of IPTG (that is, during *pbp2x* knockdown) leads to reduced intracellular K⁺, which reduces cellular turgor and rescues growth and morphology.

Moreover, valinomycin (a K⁺ carrier[43]) also neutralizes the rescuing effect of Δ*pde1* (Fig. 4b). This indicates that a lower intracellular K⁺ concentration mediated by increased c-di-AMP levels can protect cells from CW synthesis insult (Fig. 4c). Importantly, these data show that c-di-AMP plays a protective role in *S. pneumoniae* by controlling turgor (possibly similar to *Bacillus subtilis* and *Listeria monocytogenes*[44]) and can thereby modulate susceptibility to perturbations to the CW synthesis machinery, including by inhibitors such as penicillin.

## Midcell PG synthesis is modulated by CozEa and CozEb

In the interaction network, *pbp2x* is part of a tight cluster of genes that are either directly or indirectly involved in CW metabolism and maintenance.

*cozEa* shares all its interactions with *pbp2x* and they are tightly connected in the network, indicating a shared mechanistic involvement. Both have negative interactions with *murM*, *murN* and *murA*, and suppressive interactions with *pbp1a* and *pbp2a* (Fig. 5a), indicating that *cozEa* may also affect septal PG synthesis. CozEa was recently proposed as a member of the MreCD morphogenic complex of *S. pneumoniae*, controlling cell shape by positioning PBP1a in the midcell[45]. These relationships are confirmed in our interaction network (Fig. 5a). In addition, CozEa negatively interacts with *divIVA* involved in cell shape[46], and titration of *cozEa* makes *cozEb* extremely important (Fig. 5a), indicating (partial) functional redundancy. *cozEb* similarly negatively interacts with *pbp2x* (Fig. 5a,b), a knockout is hypersusceptible to the CW disruptor

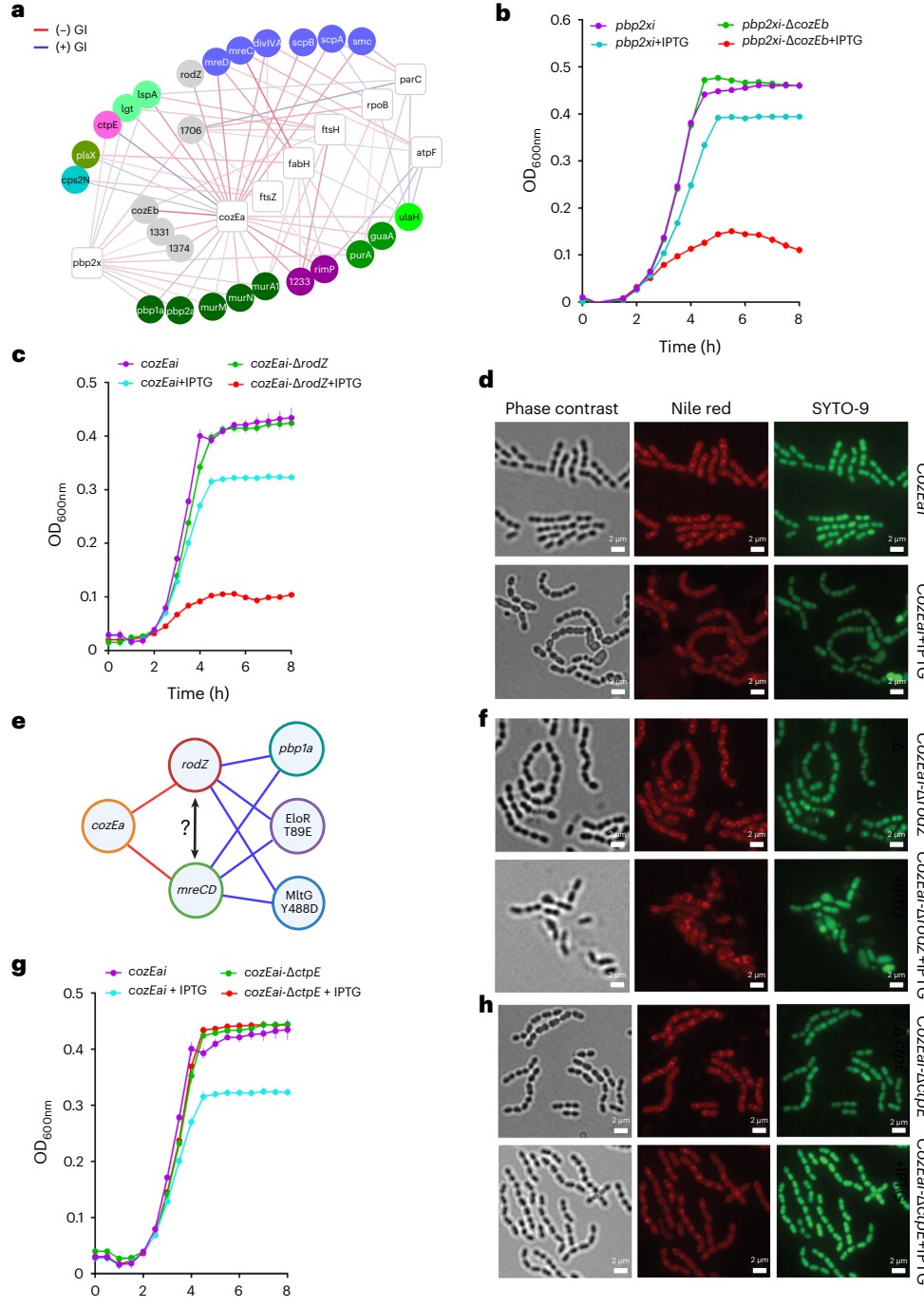

**Fig. 5 | *cozEa* interacts with genes involved in maintaining cell wall integrity.**
**a**, A network highlighting genetic interactions between *cozEa* and non-essential genes involved in processes including CW synthesis (green) and division (blue), as well as hypothetical genes (grey). Additional essential genes that interact with the same non-essential genes as *cozEa* are highlighted. CRISPRi-targeted genes are presented as white nodes and non-essential genes as circles coloured on the basis of their bioprocess (colours as in Fig. 1e). Negative and positive GI are presented as red and blue edges, respectively. The intensity of an edge colour is proportional to the strength of the interaction based on its *z*-score ranging from −10 to +10. **b**, *pbp2xi* and *pbp2xi-ΔcozEb* growth curves with or without 12.5 μM IPTG. **c**, *cozEai* and *cozEai-ΔrodZ* growth curves with or without 100 μM IPTG. **d**, Representative phase contrast, Nile red (red) and SYTO-9 (green)-stained images of *cozEai* with (bottom) or without (top) 100 μM IPTG. **e**, Summary of genetic interactions for *rodZ* and *mreCD*. In *Escherichia coli*, RodZ regulates the localization of MreB[75], a cytoskeleton actin-like protein that spatially synchronizes cell wall insertion directing the cell towards a rod shape[75]. *S. pneumoniae* (an ovococcus) lacks MreB but does possess RodZ, which seems to

have an important role in CW synthesis[48,49], and due to the lack of MreB, it could have additional roles in ovococci. Moreover, MreCD is a morphogenic complex and similar to CozEa, it is implicated in peripheral PG synthesis[45,76]. This suggests that there is a functional link between *cozEa*, *rodZ* and *mreCD*, which are all key to maintaining proper morphology; in addition, *rodZ* may share redundancy with *cozEa* and possibly with *mreCD*. **f**, Representative *cozEai-ΔrodZ* fluorescent micrographs with (bottom) or without (top) 100 μM IPTG. While a RodZ knockout does not result in a significant growth defect, possibly due to the rapid acquisition of suppressor mutations, it does trigger morphological changes, including irregularities in chain length, cell size and chromosome distribution. **g**, *cozEai* and *cozEai-ΔctpE* growth curves with or without 100 μM IPTG. For growth studies (**b**, **c** and **g**), the averages of *n* = 3 biological replicate growth curves are plotted with standard error; independent repetition (3) of the experiment showed similar results. **h**, Fluorescence micrographs of *cozEai* and *cozEai-ΔctpE* with (bottom) or without (top) 100 μM IPTG. Independent repetition (2) of the microscopy experiments showed similar results. Source data are presented in Source Data Fig. 2a,c.

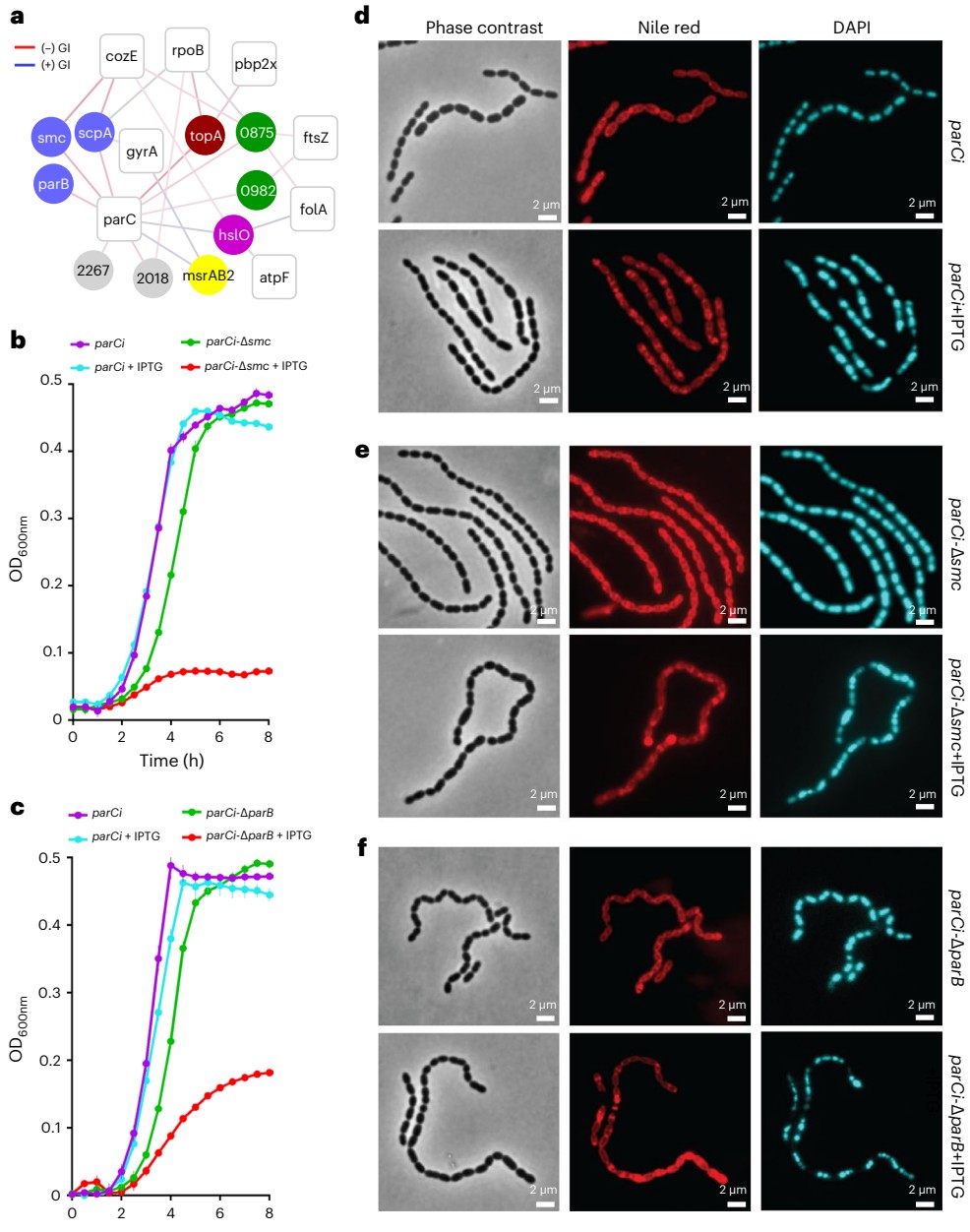

**Fig. 6 | *parC* interaction mapping connects disparate biological mechanisms including chromosome decatenation and segregation, cell division and CW synthesis. a**, A network highlighting the connections between disparate processes such as negative interactions of *cozEa* with *rimP* and SPD1233, two translation-related genes, as well as with a cluster of DNA replication-associated genes, including *smc*, *scpA*, *parC*, *parB* and *topA*. Further notable genetic interactions include those between *parC* and non-essential genes involved in CW synthesis (green), division (blue) and hypothetical genes (grey). Additional essential genes that interact with the same non-essential genes as *cozEa* are highlighted. CRISPRi-targeted genes are presented as white nodes and non-essential genes as circles coloured on the basis of their bioprocess (colour coding as in Fig. 1e). Importantly, these connections underscore the intricate link between transfer, replication, translation, CW synthesis and cell division. Negative and positive GI are presented as red and blue edges, respectively. The

intensity of an edge colour is proportional to the strength of the interaction *z*-score ranging from −10 to +10. **b**, Growth curves of *parCi* and *parCi*-Δ*smc* with or without 20 μM IPTG. **c**, Growth curves of *parCi* and *parCi*-Δ*parB* with or without 17.5 μM IPTG. For growth studies (**b** and **c**), the average of $n = 3$ biological replicate growth curves is plotted with standard error; independent repetition (3) of the experiment showed similar results. **d**–**f**, Fluorescence microscopy profiles of *parCi* (**d**), *parCi*-Δ*smc* (**e**) and *parCi*-Δ*parB* (**f**) with (bottom) or without (top) 100 μM IPTG. Negative interactions of *parC* with *parB* result in anucleate cells ranging from 11.13% in wild type to 43.82% in Δ*parB*. *parC* knockdown combined with Δ*parB* or Δ*smc*, combines increased cell (~0.2 μM) and chain (~12 μM) lengths. Representative phase contrast, Nile red (red) and DAPI (blue)-stained micrographs are presented with a 2 μm scale bar. The statistical analysis of microscopy images is presented in Extended Data Fig. 10 and Source Data Extended Data Fig. 10. Source data are presented in Source Data Fig. 2a,c.

daptomycin (Extended Data Fig. 8d,e), and CozEb can be targeted in vivo with an antibody[4], suggesting its presence and function in the CW. Moreover, three-dimensional structural illumination microscopy (3D-SIM) and total internal reflection fluorescence (TIRF) microscopy with green fluorescent protein (GFP)-tagged CozEb (Extended Data Fig. 9a–c) with

or without daptomycin (which does not influence its dynamics) show that early on during division, CozEb can be mostly found around the septum before it distributes and seems to move rapidly along the membrane (Extended Data Fig. 9d and Supplementary Movie 1). Recently, CozEa and CozEb were shown to be present in a complex with PBP1a[47], and

combined with CozEb's mobile nature as well as its interactions with multiple peripheral PBPs, we suggest that CozEa and CozEb may restrict class A PBPs in the midcell through physical interactions.

## CozEa and RodZ maintain morphology and CW synthesis

CozEa also interacts with RodZ, suggesting involvement in cell shape and morphology (Fig. 5a). While a RodZ knockout does not result in a significant growth defect (Fig. 5c and see below), it does trigger morphological changes, including irregularities in chain length, cell size and chromosome distribution (Fig. 5d,f). A *rodZ* knockout combined with *cozEa* knockdown results in drastically reduced growth and exacerbated morphological defects with reduced chain lengths, ballooning and lysis (Fig. 5c,d,f). Furthermore, Δ*rodZ* is highly sensitive to penicillin exposure (Source Data Extended Data Fig. 6) and *rodZ* shares multiple interactions with *mreCD* (Fig. 5e)[48,49], suggesting a functional link between *cozEa*, *rodZ* and *mreCD* in maintaining morphology. Of note, a recent study indicated that *rodZ* is essential[50], which would argue for the presence of a suppressor mutation in the Δ*rodZ* presented here. However, this would probably be a partial suppressor, as growth and microscopy data show that *rodZ* is crucial to maintain growth and morphology when *cozEa* is knocked down.

## CozEa controls CW integrity through CtpE and Ca$^{2+}$ homoeostasis

The *cozEa*–*ctpE* interaction (Fig. 5a,g) seems to further extend CozEa's role in maintaining envelope integrity and calcium homoeostasis. In *Mycobacterium smegmatis*, CtpE is involved in Ca$^{2+}$ homoeostasis and possibly in maintaining envelope integrity[51]. While knockdown of *cozEa* has a mild impact on morphology and growth, these defects can be suppressed by knocking out *ctpE* (Fig. 5g,h). This suggests that a reduced amount of CozEa results in CtpE dysregulation and intracellular calcium, resulting in cell envelope integrity defects. Importantly, under high calcium concentrations, disabling *ctpE* seems to restore calcium homoeostasis and compensate for the impact of *cozEa* knockdown, suggesting that CozEa has a regulatory effect on CtpE. Moreover, negative interactions between *cozEa* and at least two lipoprotein maturation-related genes, *lspA* and *lgt*[52,53] (Fig. 5a and Extended Data Fig. 9e,f), further support CozEa's role in maintaining envelope integrity.

## Chromosome segregation affects DNA decatenation and morphology

The integration of seemingly disparate processes is further highlighted by negative interactions between *cozEa* and translation-related genes, as well as with a cluster of DNA replication-associated genes (Fig. 6a). The latter cluster reveals some important relationships between these genes. For successful cell division, the decatenated chromosome needs to be segregated to daughter cells. Previously, *smc* and *parB* have been shown to influence chromosome segregation in *S. pneumoniae*[54], while ParC is involved in chromosome decatenation[55]. Fluorescence microscopy shows that negative interactions of *parC* with *parB* (Fig. 6a–c) result in anucleate cells and uneven chromosome distribution (Fig. 6d, Extended Data Fig. 10 and Source Data Extended Data Fig. 10), highlighting that chromosome distribution is dependent on decatenation by *parC*. Deletion of *smc* or *parB* results in enhanced chaining (Fig. 6e,f), indicating that SMC is recruited to the *oriC* by ParB, and SMC-mediated chromosome segregation affects downstream cell division. *parC* knockdown combined with Δ*parB* or Δ*smc*, produces visual deleterious effects that combine increased cell and chain lengths (Fig. 6e,f). The negative interactions of ParC with partitioning protein ParB and segregation protein SMC confirm functional dependency between chromosome decatenation and segregation, and the critical link with cell division and cell wall synthesis.

## Discussion

By exploiting the combined power of essential gene knockdown and non-essential gene knockout, we develop CRISPRi–TnSeq to map bacterial essential–non-essential genetic interactions. By focusing on key biological processes, we screened over 24,000 gene–gene pairs, identifying 1,334 genetic interactions. In conjunction with other tools, CRISPRi–TnSeq highlights the importance of mapping genetic interactions. On one hand, the identification of suppressor interactions is a strength of CRISPRi–TnSeq as it uncovers hidden redundancy. This, in turn, can be an indicator of the ease with which an organism can find a solution against a specific perturbation (for example, a drug and resistance emergence). On the other hand, aggravating interactions highlight the dependencies within a genome and thereby potential synergistic targets. CRISPRi–TnSeq does come with several considerations. First, employing CRISPRi–TnSeq in additional conditions should result in more interactions. Second, CRISPRi often introduces polar effects[56]. However, neither here nor in our previous study, where these CRISPRi strains were characterized[25], did we observe significant polar effects. Importantly, while CRISPRi–TnSeq is a powerful and sensitive tool, it will by no means be the last. For instance, we envision at least two alternative versions: (1) a dual Tn-Seq version based on creating double knockout Tn libraries; and (2) a dual-CRISPRi version incorporating two single guide RNAs into a single cell. While we are actively working on both approaches, we do foresee that each tool will have its advantages and disadvantages concerning questions that can be answered, sensitivity, ease of implementation, optimization and scalability. Moreover, in two recent studies, a new CRISPRi approach[57,58] was introduced using a CRISPR adaptation system (CALM) that randomly produces CRISPR (cr)RNA libraries using transformed genomic DNA. CALM can be used to knock down genes to study genetic interactions; however, broad implementation is yet unclear, and due to crRNAs' random nature, multiple crRNAs are required to knock down a single gene. This issue scales when intending to knock down two genes simultaneously, necessarily leading to large crRNA libraries. Besides such libraries probably being difficult to generate, they would be challenging to track, while quantification of crRNAs with variable knockdown and fitness effects may make it harder to accurately determine the precise fitness effect of both individual and dual gene-targeting crRNAs. In contrast, CRISPRi–TnSeq will be most valuable in precisely mapping interactions between a set of accurately fine-tunable (essential) genes and the rest of the genome. Moreover, the experimental set-up, sample preparation, data collection and analyses outlined here can be used as detailed guidelines without the need for much further development. We thus expect the implementation of CRISPRi–TnSeq to be relatively easy, especially in species where both tools have been applied, and prove to be a powerful tool that can untangle fundamental network relationships and thereby uncover new biological insights.

## Methods

### Strains, growth conditions and transformation

CRISPRi strains of *S. pneumoniae* D39V (NZ_CP027540.1) were adopted from our previous study[25] and are detailed in Supplementary Table 1. *Streptococcus pneumoniae* D39V and its derivatives were cultivated either in C + Y media, pH 6.8 (ref. 27), Todd Hewitt broth supplemented with yeast extract (THY), 5 μg ml$^{-1}$ Oxyrase (Oxyrase) and 150 U ml$^{-1}$ Catalase (Worthington Bio Crop), or on 5% sheep's blood agar plates at 37 °C in a 5% CO$_2$ atmosphere. Unless otherwise indicated, for growth studies, cells were grown to mid-exponential phase, diluted in the same growth media and incubated in a BioSpa automated incubator with integrated plate reader (BioTek). Growth curves were plotted using GraphPad Prism 9. Transformation of *S. pneumoniae* was performed as previously described[14]. To reduce opportunities for suppressor mutations arising (commonly in *dcas9*), taking over the population and being carried forward to follow-up experiments, strains were always grown in rich medium without IPTG. In addition, multiple stock tubes were prepared from the same culture and stored, and used for the entire study, while whole genome sequencing was used to confirm strain stability.

## Transposon-mutant library construction and selection experiments

Transposon-mutant libraries of individual CRISPRi strains using the mariner transposon Magellan6 were constructed as previously described[3,13,14]. The transposon carries a kanamycin resistance cassette that lacks a transcriptional terminator therefore allowing for readthrough transcription and has stop codons in all three frames in either orientation to prevent aberrant translation products. DNA was isolated from individual CRISPRi strains, followed by in vitro transposition. Transposed DNA was used to transform the corresponding CRISPRi strain leading to the construction of a mutant library. Six independent transposon-mutant libraries, each consisting of ~10,000 mutants, were constructed for each CRISPRi strain. Mutant libraries were grown in C + Y media up to mid-exponential phase, which represents the initial population (T1) in a Tn-Seq experiment. Subsequently, a culture was diluted to optical density (OD) = 0.003 and grown in the same media in the absence and presence of two sub-inhibitory concentrations of IPTG or different antibiotics. Cells were collected after 6–8 doublings when culture density reached OD ≈ 0.8. Note that, while we introduced transposon insertions in the genome of individual CRISPRi strains, this could also be done in the opposite way. Specifically, gene-specific CRISPRi cassettes could be transformed into the same transposon-mutant library. However, it will be challenging to ensure the incorporation of a CRISPRi cassette into all transposon mutants in a pool with equal efficiency. The result could be that a large number of transposon mutants drop out from the pool, possibly resulting in a skewed distribution. Therefore, generating a transposon-mutant library in a CRISPRi strain background will probably result in a more complex pool.

## Expression analysis

Total RNA was isolated from cells grown with and without IPTG using the RNeasy mini kit (Qiagen), followed by DNase treatment with the TURBO DNA-free kit (Invitrogen). Complementary DNA was synthesized from 1 µg RNA with random hexamers using the iScript Reverse Transcription Supermix (Bio-Rad). Quantitative PCR was performed on a MyiQ (Bio-Rad) using i*Taq* Universal SYBR Green Supermix (Bio-Rad). Each sample was measured in technical and biological replicates, and CRISPRi-target gene expression was normalized against the 50S ribosomal RNA gene SPD2031 (*rplI*).

## Analyses of CRISPRi-transposon-mutant libraries

Mutant libraries for each CRISPRi strain were grown in the absence and presence of IPTG, followed by DNA isolation using the DNeasy Blood and Tissue kit (Qiagen). Tn-Seq sample preparation, Illumina sequencing and fitness calculations were performed as described previously[3,13,14]. In brief, for each insertion, fitness $W_i$ was calculated by comparing the fold expansion of the mutant relative to the rest of the population with the following equation: $W_i = (\ln(N_{i(t2)} \times d/N_{i(t1)}))/ (\ln((1 - N_{i(t2)}) \times d/(1 - N_{i(t1)})))$[3,13], where $N_{i(t1)}$ and $N_{i(t2)}$ are the frequencies of the mutant in the population at the start and end of the experiment, respectively, and $d$ (expansion factor) represents the growth of the bacterial population during library selection. The expansion factor was directly measured for each library by plating appropriate dilutions of the bacterial library at $t_1$ and $t_2$ and dividing the number of bacteria counted at each timepoint ($d$ = number of bacteria at $t_2$/number of bacteria at $t_1$). As we have shown repeatedly, this way, fitness for each insertion mutant ($W_i$) represents the actual growth rate per generation, making fitness independent of time, which allows comparisons between conditions and strains[3,13]. To determine whether fitness values are significantly different between conditions, three requirements were imposed: (1) average relative fitness $W_i$ was calculated from 3 or more data points; (2) the fitness difference between conditions was larger than 0.1 (10%) and (3) the fitness difference was statistically significant ($P_{adj} < 0.05$) in a one-sample $t$-test with Bonferroni or Benjamini–Hochberg correction for multiple testing. Fitness datasets are presented in Supplementary Table 3. All fitness change values are presented in a volcano plot using Vega-Lite[59].

## Genetic interaction mapping and considerations

Genetic interactions of two genes are defined as a deviation from the multiplicative model[13]. If individual knockout mutants of genes $a$ and $b$ have fitness $W_a$ and $W_b$, respectively, a genetic interaction was scored when fitness of the double mutant ($W_{ab}$) deviated from the multiplicative fitness $W_a \times W_b$ (refs. [11–13]). In CRISPRi–TnSeq, the fitness difference between the absence ($W_{noIPTG}$) and presence ($W_{IPTG}$) of IPTG was calculated for the transposon mutant of individual non-essential genes. Since CRISPRi knockdown of an essential gene target reduces the growth of the entire mutant library, the relative fitness of individual insertion mutants remains similar in the presence of IPTG, unless the corresponding essential and non-essential gene pair genetically interact. Thus, when an essential and non-essential gene pair genetically interact, a transposon's frequency will significantly decrease or increase, representing a negative or positive interaction, respectively. A significant deviation in fitness of a transposon mutant in the absence and presence of essential gene knockdown thus constitutes a genetic interaction when the observed fitness ($W_{ab}$) is significantly different from the expected fitness ($W_a \times W_b$), where $\Delta w$ ($\Delta w = W_{IPTG} - W_{noIPTG}$) represents the size of the fitness effect[3,13,20].

The impact on growth due to essential gene knockdown depends on multiple factors, including stability and abundance of the corresponding gene product. Moreover, the fitness effect of a genetic interaction increases with the extent of growth inhibition (that is, due to the amount of added IPTG). To enable comparisons of genetic interactions across different datasets produced for individual CRISPRi targets (for example, those generated with different amounts of IPTG), individual fitness differences ($\Delta w$) and genetic interaction datasets were $z$-score normalized. All significant genetic interactions ($Z > 1.5$ and $Z < -1.5$) with $P_{adj} \leq 0.075$ were mapped in a genetic interaction network using Cytoscape[60]. $Z$-score normalization and cut-off eliminate the dominance of datasets produced under strong growth inhibition. It should be noted that random transposon insertions are likely in the large *dcas9* gene. Those mutants are expected to be identified with increased fitness in CRISPRi–TnSeq due to the lack of an active knockdown system. As expected, *dcas9* mutant fitness increased with increased IPTG concentrations for each library (Source Data Extended Data Fig. 2a), which confirms successful target knockdown and growth inhibition of the entire mutant library, except for the mutants of *dcas9*. Therefore, every Tn insertion into *dcas9* essentially serves as a positive control in the CRISPRi–TnSeq datasets.

Importantly, the CRISPRi–TnSeq data indicate that a situation can be created where IPTG is increased to such an extent that the resulting knockdown of the essential gene becomes so large that any other disruption to the system (for example, a non-essential gene knockout) can tip the balance and cause a disruption that results in a very strong fitness defect, indicative of a genetic interaction. However, this defect is probably not the result of a genetic interaction but due to the organism being on the 'verge of collapse'[61]. This situation is similar to that when a Tn-mutant library experiences a very narrow bottleneck, resulting in a large random loss of Tn mutants, which are then excluded from the dataset[3,4,16,17,20]. Consequently, libraries with a loss of ≥10% of the Tn insertion mutants were disregarded here as well. Such a situation occurred with *parCi* libraries grown twice at 100 mM IPTG, resulting in a loss of >26% of Tn mutants, hence this dataset was excluded from further analysis.

## Genetic interaction distributions, enrichment, visualization and gene clustering

*Streptococcus pneumoniae* D39 genome was plotted using Circos 0.69 (ref. [62]). The distribution of interactions across the genome was

determined for each 1 kb region and plotted as a histogram. Enrichment was determined for each 50 kb region in the genome by comparing it to the distribution of interactions of random regions within the genome. The Jaccard index was determined by defining the size of the intersection of the significant gene pool divided by the size of the union of the significant gene pool. Gene set enrichment analysis (GSEA)[63] was performed to test the overrepresentation of genes from specific biological processes and summarized in a heat map as $q$-values indicating false discovery rates. Pathway analyses were performed for individual datasets using BioCyc[64]. Genetic and chemical–genetic interaction datasets were visualized with Cytoscape[60]. Spearman's rank correlation coefficient-based clustering was performed on CRISPRi- and antibiotic-TnSeq datasets to identify functional units of non-essential genes. Non-essential genes (nodes) were clustered on the basis of their fitness profiles (attributes) under CRISPRi knockdown or antibiotic exposure. Experiments and non-essential genes were clustered on the basis of the entire datasets (Source Data Extended Data Fig. 6) and 5 selected clusters are presented in the heat map of Fig. 3b. Functional categories were adopted from the NCBI Reference Sequence file NC_008533.2.

### Gene deletion mutant construction

Gene knockout mutants were constructed by substituting the entire coding sequence with a chloramphenicol resistance cassette constructed through overlap extension PCR as previously described[13,65]. Deletion mutant strains and oligos can be found in Supplementary Tables 1 and 5, respectively. As we have previously shown[1,3,13,20,66,67], fitness ($W_i$) calculated from Tn-Seq data can be directly related to a strain's doubling time by, for instance, taking the ratio of the doubling time of a mutant to that of the wild type. For instance, if a mutant has a fitness of 0.5, it means that its doubling time is twice that of the wild type (for example, 30 min vs 60 min). Doubling time was calculated from exponential growth following [ln(2)/ln(Ratio)/Time]×60. Ratio = Final OD/Initial OD, and Time = Hours to grow from Initial to Final OD. To ensure the robustness of the scored interactions, genetic interactions were tested in validation experiments both below and above IPTG concentration(s) used in CRISPRi–TnSeq experiments.

### Construction of *gfp*-labelled strains

Strains and primers used in this study are listed in Supplementary Tables 1 and 5, respectively. Strain CG89 (*S. pneumoniae* TIGR4, *bgaA*::P$_{Zn}$-*m(sf)gfp-sp_1505*) was constructed by transformation of *S. pneumoniae* TIGR4 wild type with plasmid pCG15. This plasmid was built by ligating the product amplified by PCR with primers SP1505_F-SpeI and SP1505-R-NotI from *S. pneumoniae* DNA into plasmid pCG6, allowing for N-terminal GFP fusion of SP_1505 (CozEb) under the control of a ZnCl$_2$-inducible promoter (P$_{Zn}$). Strain CG90 (TIGR4, *sp1505*::*kan-m(sf)gfp-sp1505*) was built by transformation of TIGR4 wild type with product *kan-m(sf)gfp-sp1505* to replace the original SP_1505 gene. The DNA product was obtained by Gibson assembly of three PCR fragments: (1) *m(sf)gfp-sp1505* amplified with primers SP_1505-up-Htra-GFP-F and SP_1505-down-R on CG89 genomic DNA; (2) *kan*, kanamycin resistance marker, amplified with primers kan-F and kan-R-SP_1505-up and (3) *sp1505* upstream region amplified with primers SP_1506-down-F and SPD_1506-up-R-kan on TIGR4 wild type genomic DNA.

### Cyclic-di-AMP assay

Strains were grown in C + Y media at 37 °C with 5% CO$_2$ up to OD$_{600nm}$ of 0.2–0.4. Cultures were diluted to OD = 0.003 in C + Y media and regrown with and without IPTG up to OD = 0.5–1.0. Cultures (OD = 5 × ml volume) were pelleted by centrifugation, cell pellets were resuspended in lysis buffer (20 mM Tris pH 7.4, 1% Triton X-100, 100 μg ml$^{-1}$ lysozyme, 5 U ml$^{-1}$ DNase I), followed by incubation at room temperature for 20 min. Lysates were centrifuged at 15,000 × *g* for 5 min at 4 °C.

Supernatants were used in a c-di-AMP ELISA assay (Cayman Chemicals, 501960) following manufacturer instructions. This kit measures cellular c-di-AMP against HRP-tagged c-di-AMP by competitive ELISA binding. Cyclic-di-AMP levels were calculated on the basis of the standard curve, normalized by culture OD and presented in mole per OD.

### Fluorescence microscopy of CRISPRi strains and mutants

Morphological changes associated with the knockdown of target genes were analysed by microscopy as described previously[25]. Briefly, cells were grown in C + Y media at 37 °C for 2.5–3 h with and without IPTG. Nile red (membrane dye, 1 μl, 1 mg ml$^{-1}$) was added to 1 ml of cell suspension and incubated at room temperature for 4 min, followed by 1 μl 1 mg ml$^{-1}$ DAPI (DNA dye) and incubation for another minute. Next, cells were precipitated by centrifuging at 8,000 × *g* for 2 min and resuspended in 30 μl of fresh C + Y medium. Cell suspension (0.5 μl) was spotted on a PBS agarose pad prepared on the slide. Finally, cells were visualized with an Olympus IX83 (Olympus) or Deltavision Elite (GE Healthcare) fluorescence microscope. Microscopy images were analysed with ImageJ[68].

### Fluorescence microscopy image analysis

Microscopy image preprocessing and segmentation was conducted using customized Python scripts empowered by the Python package, Scikit-image. Bacterial chain and cellular features were extracted using a revised version of a previously established Python package, MOMIA[69] (GitHub: https://github.com/jzrolling/CleanSpace/tree/master/momia2). Briefly, phase contrast images were first processed using a dual-bandpass frequency filter to suppress background aberration due to uneven illumination and non-cellular micropatterns of the agarose pad. The processed phase contrast intensities were then normalized by calculating the ratio of two Gaussian smoothed images, with Gaussian kernel Sigmas of 0.5 and 10, respectively. Binary masks of the bacterial chains were generated using the iso-data thresholding method, followed by morphological operations to smoothen the chain masks. The core parts of individual cells were identified using the Shape index filter and used to guide cell segmentation based on the watershed method. To account for occasional out-of-focus issues that mostly occurred near the corner of the imaging fields, we manually annotated out-of-focus regions and excluded the affected cells and chains from downstream analysis. A record of the analysis is deposited on GitHub at https://github.com/jzrolling/SP_manuscript_deposit (ref. 70) and on Zenodo at https://doi.org/10.5281/zenodo.11465385 (ref. 71).

### Time-lapse microscopy

*Streptococcus pneumoniae* TIGR4 strains were pregrown in C + Y media at 37 °C until OD$_{600}$ = 0.3. Cells were diluted 100-fold in fresh C + Y media, supplemented when indicated with daptomycin and cultured until OD$_{600}$ = 0.1. Cells were collected by centrifugation for 1 min at 10,000 × *g*, washed once in fresh C + Y media and resuspended into 1/10th of the centrifuged volume. A volume of 1 μl was spotted onto a C + Y-10% polyacrylamide pad (incubated twice for 1 h in C + Y medium supplemented with daptomycin when applicable) inside a Gene Frame (Thermo Fisher) sealed with a cover glass. The resulting slide was then placed into the microscope chamber pre-incubated at 30 °C or 37 °C. Phase contrast time-lapse microscopy was performed as previously described[72] using a DV Elite microscope (GE Healthcare) with a sCMOS (PCO-edge) camera, a DV Trulight solid-state illumination module (GE Healthcare) and a ×100/1.40 oil-immersion objective. Conventional epifluorescence time-lapse microscopy was performed on the same microscope using a GFP filter (Ex: 475/28 nm, BS: 525/80 nm, Em: 523/36 nm). TIRF/HILO fluorescence time-lapse microscopy was performed on a Leica DMi8 microscope with a sCMOS DFC9000 (Leica) camera using a ×100 oil-immersion Plan APO TIRF objective, a 488 nm excitation laser module and a GFP filter (Ex: 470/40 nm Chroma ET470/40x, BS: LP 498 Leica 11536022, Em: 520/40 nm Chroma ET520/40 m).

Images were acquired with either LasX v.3.4.2.18368 (Leica) or Soft-WoRx v.7.0.0 (GE Healthcare) and processed with FIJI[73].

## 3D-structural illumination microscopy

Live bacterial cells for 3D-SIM were spotted on PBS–10% acrylamide pads. Acquisition was performed using a DeltaVision OMX SR microscope (GE Healthcare) equipped with a ×60/1.42 NA objective (Olympus) and 488 nm excitation laser. Nine $Z$-sections of 0.135 µm each were acquired in Structure Illumination mode with 20 ms exposure and 25% laser power. The 135 images obtained were reconstructed with a Wiener constant of 0.01 using SoftWoRx (GE Healthcare).

## Reporting summary

Further information on research design is available in the Nature Portfolio Reporting Summary linked to this article.

## Data availability

All sequence data can be found in the NCBI Sequence Read Archive under BioProject PRJNA813307. Functional categories are adopted from the NCBI Reference Sequence file NC_008533.2. Source data for the figures are provided with this paper.

## Code availability

Python codes used for the analysis of the fluorescence microscopy images are deposited on GitHub at https://github.com/jzrolling/SP_manuscript_deposit/tree/main (ref. 70 ) and on Zenodo at https://doi.org/10.5281/zenodo.11465385 (ref. 71).

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

## Acknowledgements

DNA sequencing was performed at the Boston College Sequencing Core. We thank J. Anthony for establishing the *Aerobio* sequencing analyses platform, and F. Rosconi, J. Ortiz-Marquez and K. Baker for valuable discussions. This work was supported by NIH R03 AI135737, R21 AI156203, R01 AI148470, U01 AI124302 and U19 AI158076 to T.v.O. X.L. was supported by the National Nature Science Foundation of China (NSFC, 82270012) and the Science and Technology Project of Shenzhen (JCYJ20220818095602006).

## Author contributions

B.J. and T.v.O designed the research and wrote the manuscript. B.J., X.L., J.D., H.P., B.L. and C.G. performed experiments and collected data. B.J., D.L. and J.Z. performed all bioinformatic and statistical analyses. T.v.O. and J.-W.V. supervised the project. All authors contributed to manuscript writing and approved the final paper.

## Competing interests

The authors declare no competing interests.

## Additional information

**Extended data** is available for this paper at https://doi.org/10.1038/s41564-024-01759-x.

**Correspondence and requests for materials** should be addressed to Jan-Willem Veening or Tim van Opijnen.

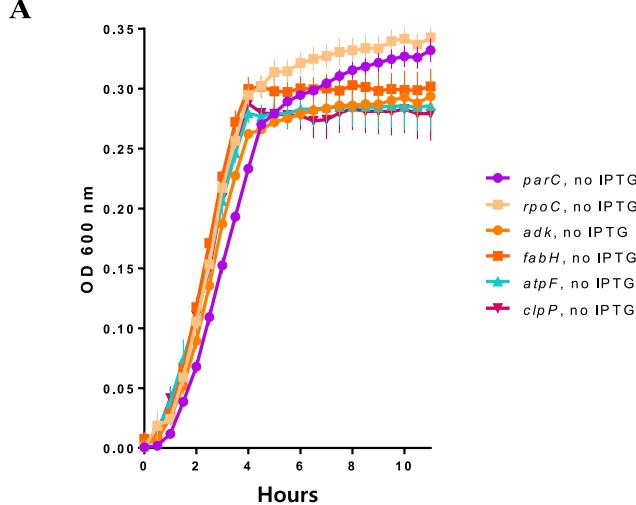

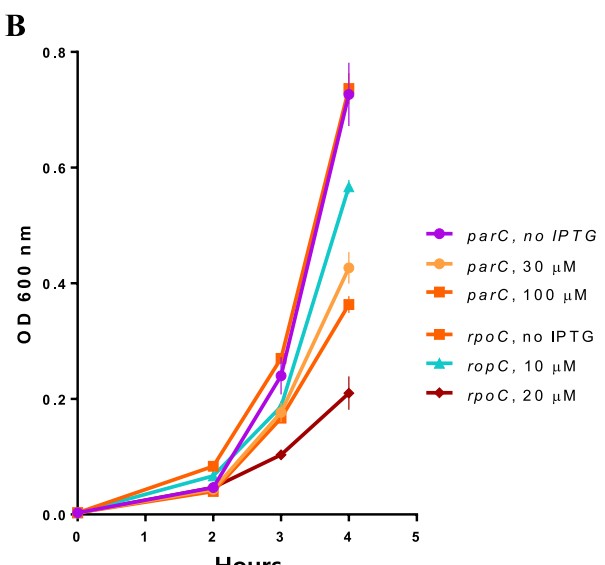

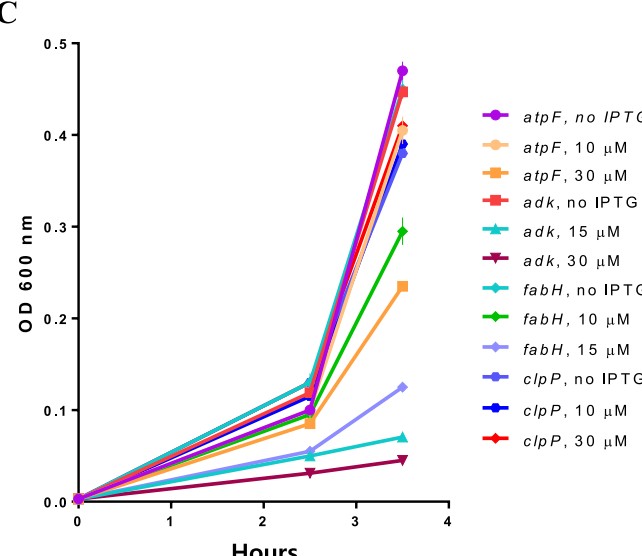

**Extended Data Fig. 1 | Growth curves of *S. pneumoniae* D39 CRISPRi strains with and without IPTG.** (**a**) Growth for six CRISPRi strains in the absence of IPTG. All growth curves were originally presented in Liu et al., 2017, and are similar to those highlighted here. (**b**) Growth of *parC* and *rpoC* strains with transposon libraries in their background in the absence and presence of IPTG. (**c**) Growth of *adk*, *fabH*, *atpF* and *clpP* in the absence and presence of different concentrations of IPTG. CRISPRi strains were grown in 96-well plates and Tn-mutant libraries were grown in 15 mL tubes. Average of (n = 3) biological replicate growth curves are plotted with standard error.

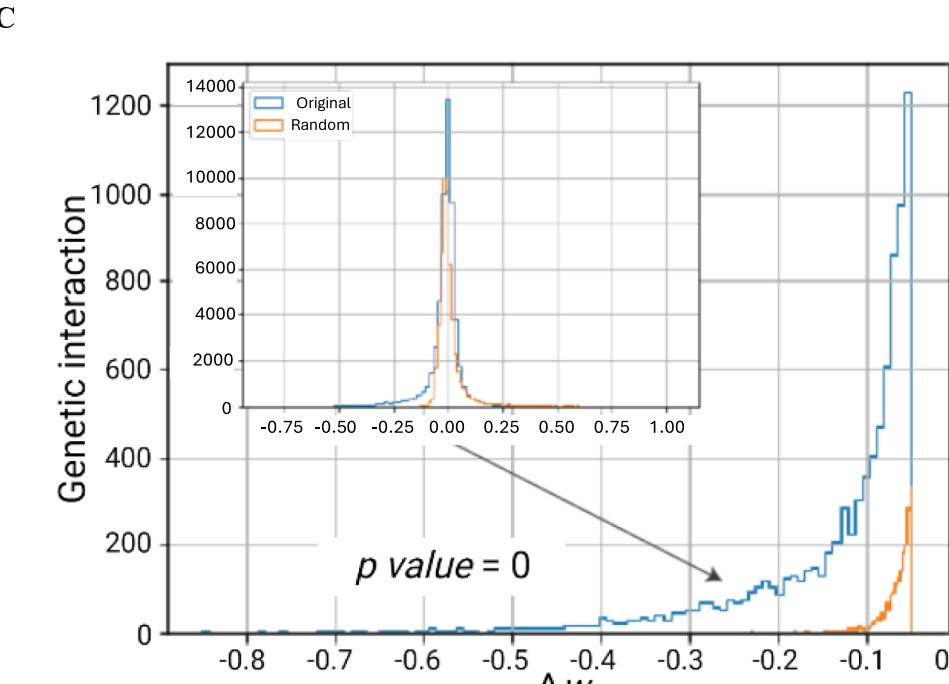

**A**

**B**

**C**

**Extended Data Fig. 2 | See next page for caption.**

**Extended Data Fig. 2 | CRISPRi-TnSeq data of *S. pneumoniae* D39 CRISPRi strains. a**. CRISPRi-TnSeq data of 13 *S. pneumoniae* D39 CRISPRi strains performed in the presence of different IPTG concentrations. Genetic interactions are z-score normalized (scale ranging from +10 to -10) and summarized in a heatmap. Increased and decreased fitness changes correspond to positive (blue) and negative (red) genetic interactions, respectively. **b**. Genetic interactions of *atpF* are highlighted in a separate network, which is enriched for

positive connections. *atpF* (CRISPRi-target) is presented as a white square, and nonessential genes as circles colored based on their bioprocess (colors as in Fig. 1e). **c**. CRISPRi-TnSeq datasets are enriched for negative genetic interactions. Comparison of real and random genetic interaction datasets are presented with fitness change ($\Delta w$) and number of interactions (inset). The shoulder of negative fitness change is highlighted. *p*-value of Mann-Whitney U test on observed and random generated negative genetic interaction datasets is presented.

A

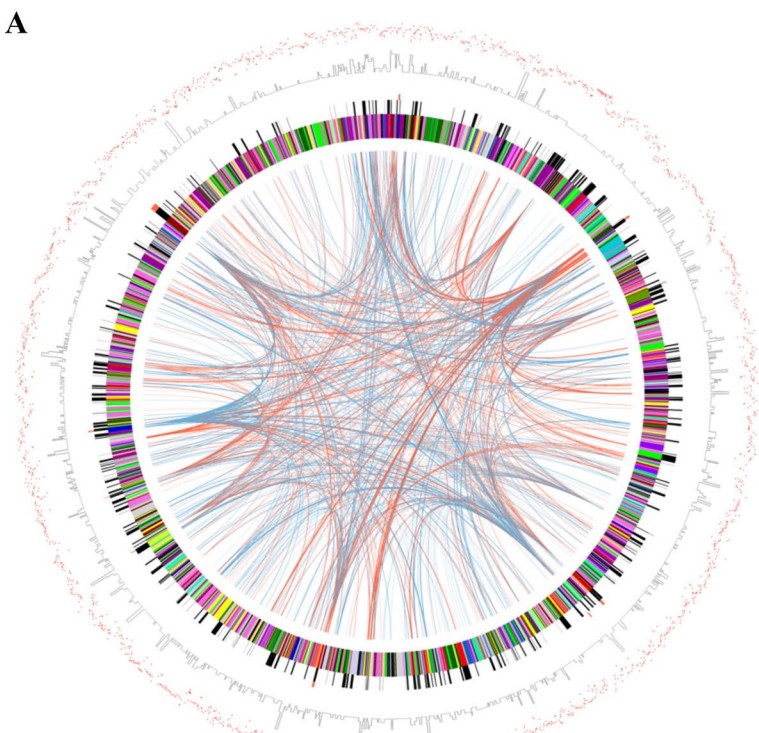

B

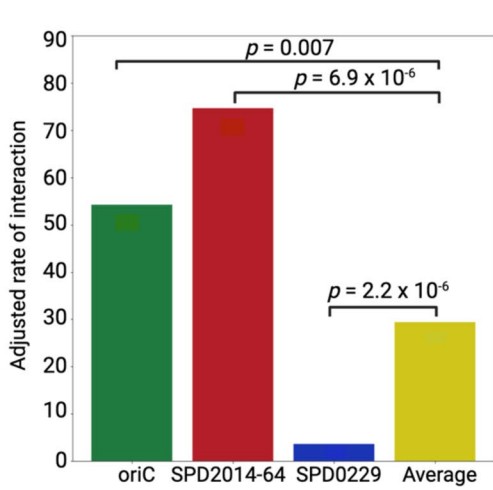

**Extended Data Fig. 3 | Genome-wide interactions between essential and non-essential genes. a**. Genome-wide interactions between essential genes and non-essential genes are depicted in a circos-plot. The red and blue lines represent positive and negative interactions, respectively. The figure highlights the distribution of interactions across the entire genome. In the first layer, the genome is presented, where perpendicular lines represent individual genes colored by biological processes (colors as in Fig. 1e). In the second layer, essential genes are marked by black lines. Genetic interaction frequencies of individual nonessential genes and expression levels are presented in black and red circles, respectively. **b**. Some areas of the genome carry significantly higher or lower numbers of interactions compared to the average number of interactions. Interactions per 50 kb and the *p*-value (one-sided Fisher's exact test) compared with the average number of interactions are shown for 3 regions (oriC, SPD0014-64, and SPD0229).

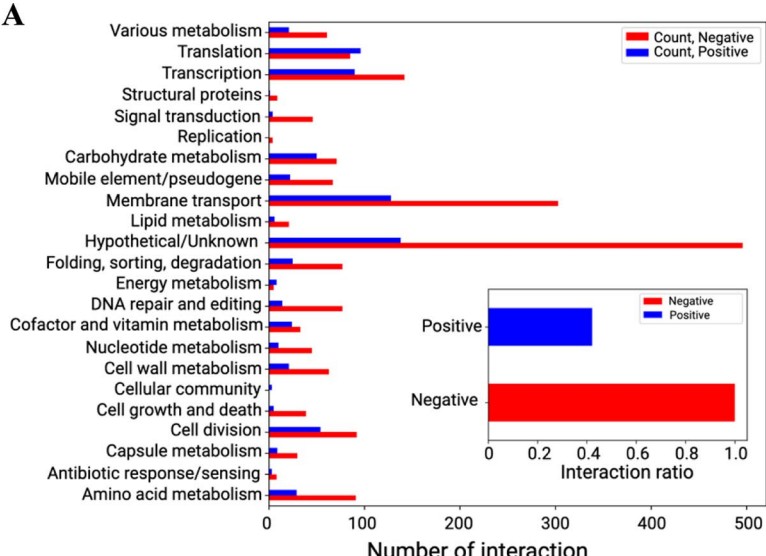

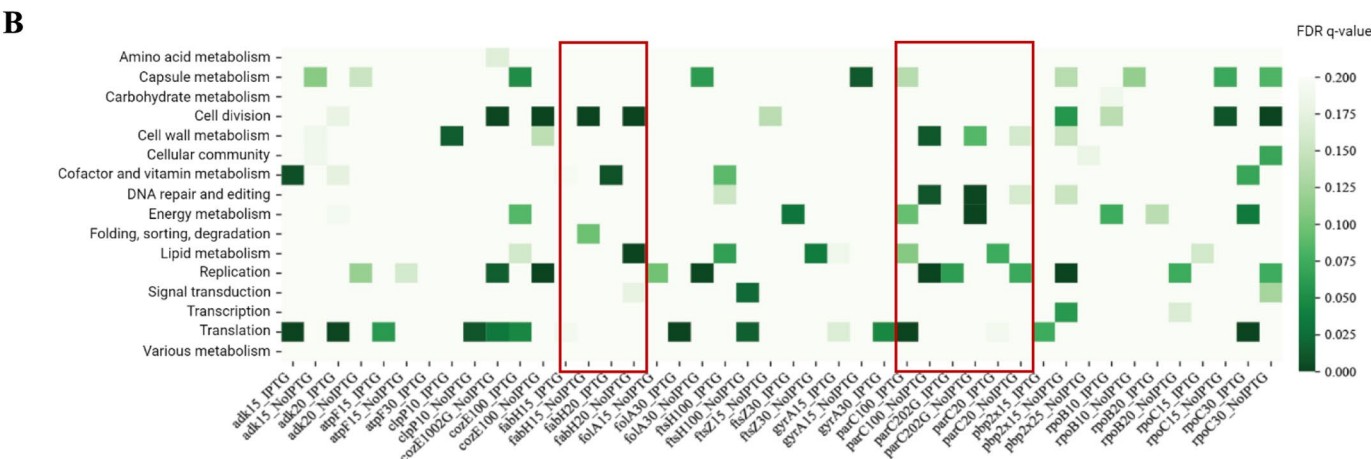

**Extended Data Fig. 4 | Distribution of genetic interactions over biological processes. a.** Genetic interactions are distributed over different biological processes. Negative interactions (red) are more prevalent than positive (blue) interactions. **b.** Nonessential genes functionally linked to corresponding target essential genes are enriched in CRISPRi-TnSeq datasets identified by gene set enrichment analysis (GSEA).

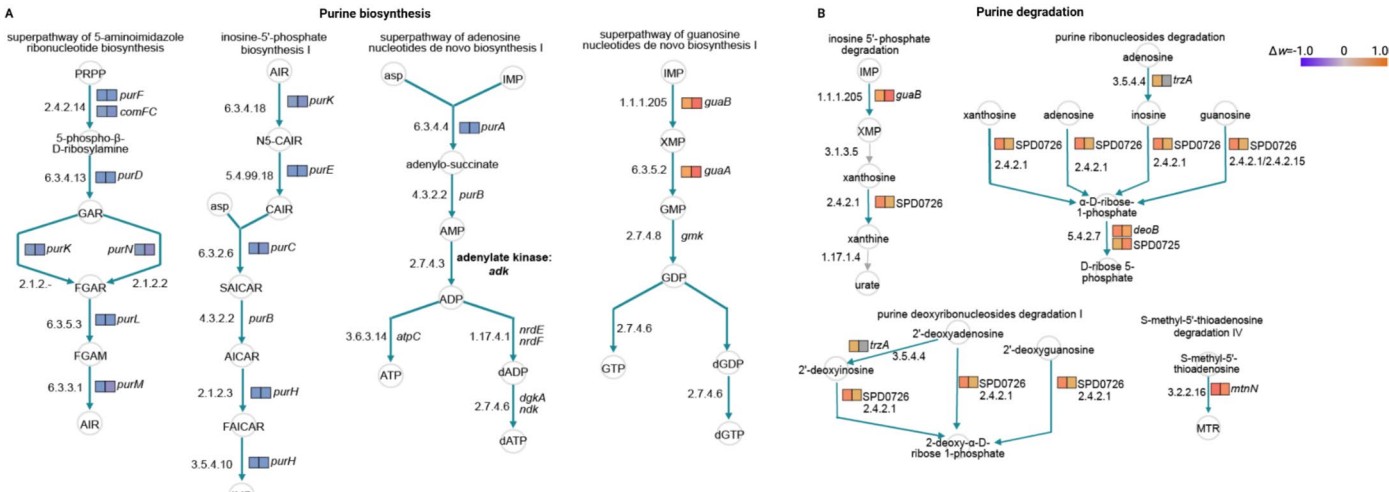

**Extended Data Fig. 5 | Purine metabolism is tightly linked to *adk* level.**
**a.** Purine biosynthesis pathways become important when *adk* is knocked down. Nonessential genes involved in AMP biosynthesis are identified with negative genetic interactions in the presence of 15 or 20 μM IPTG presented by blue boxes. In contrast, nonessential genes involved in GMP biosynthesis are disadvantageous under *adk* knockdown and therefore identified with positive genetic interactions presented by orange boxes. **b.** Purine degradation pathways become disadvantageous when *adk* is knocked down. Genes involved in purine degradation pathways are identified with positive genetic interactions in the presence of 15 or 20 μM IPTG presented by red boxes. Since AMP is the substrate of adenylate kinase Adk to produce ADP, reduction in the levels of AMP substrate likely impacts the growth strongly when *adk* is already knocked down. Therefore, compensatory pathways that are involved in AMP biosynthesis become crucial.

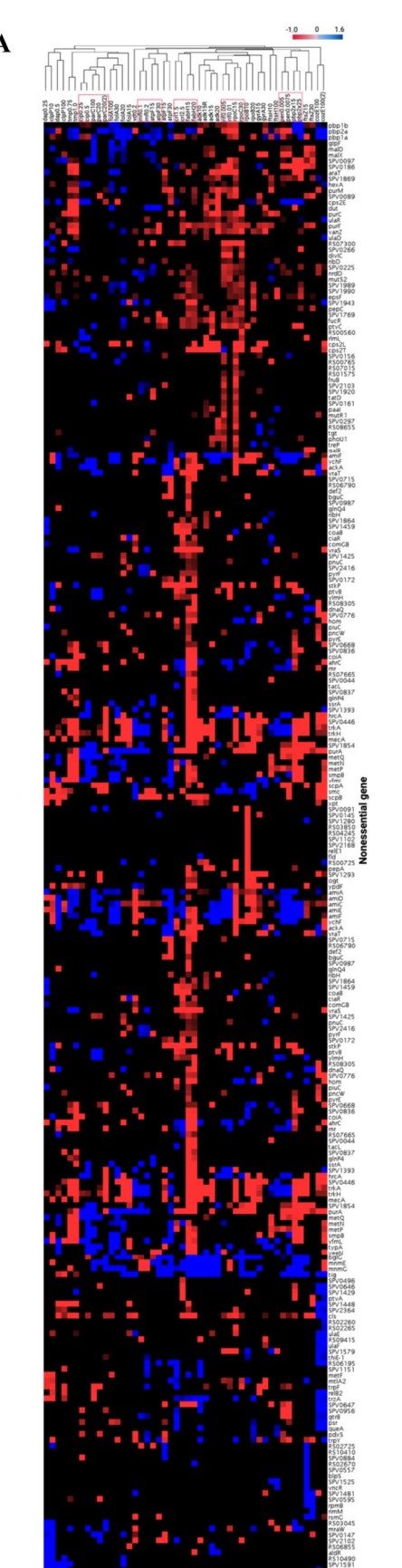

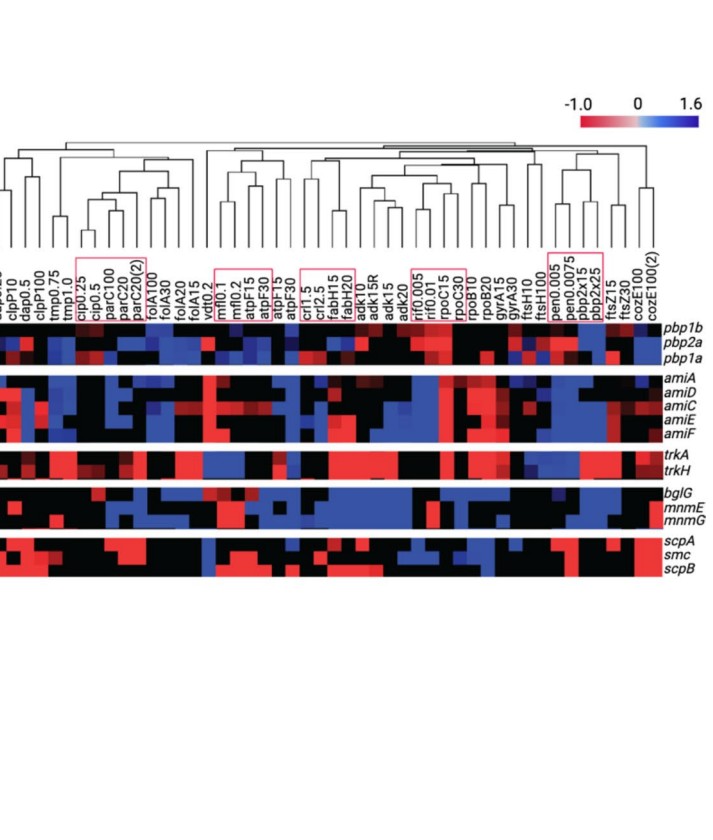

**Extended Data Fig. 6 | Clustering of interaction datasets. a**. Spearman's rank correlation coefficient-based clustering of CRISPRi- and antibiotic-TnSeq datasets. Nonessential genes (nodes) are clustered based on their fitness profiles (attributes) under CRISPRi knockdown or antibiotic exposure. Z-scored fitness values are plotted in the heatmaps, ranging from −1 to 1.6. **b**. 5 selected clusters from Figure A are highlighted separately.

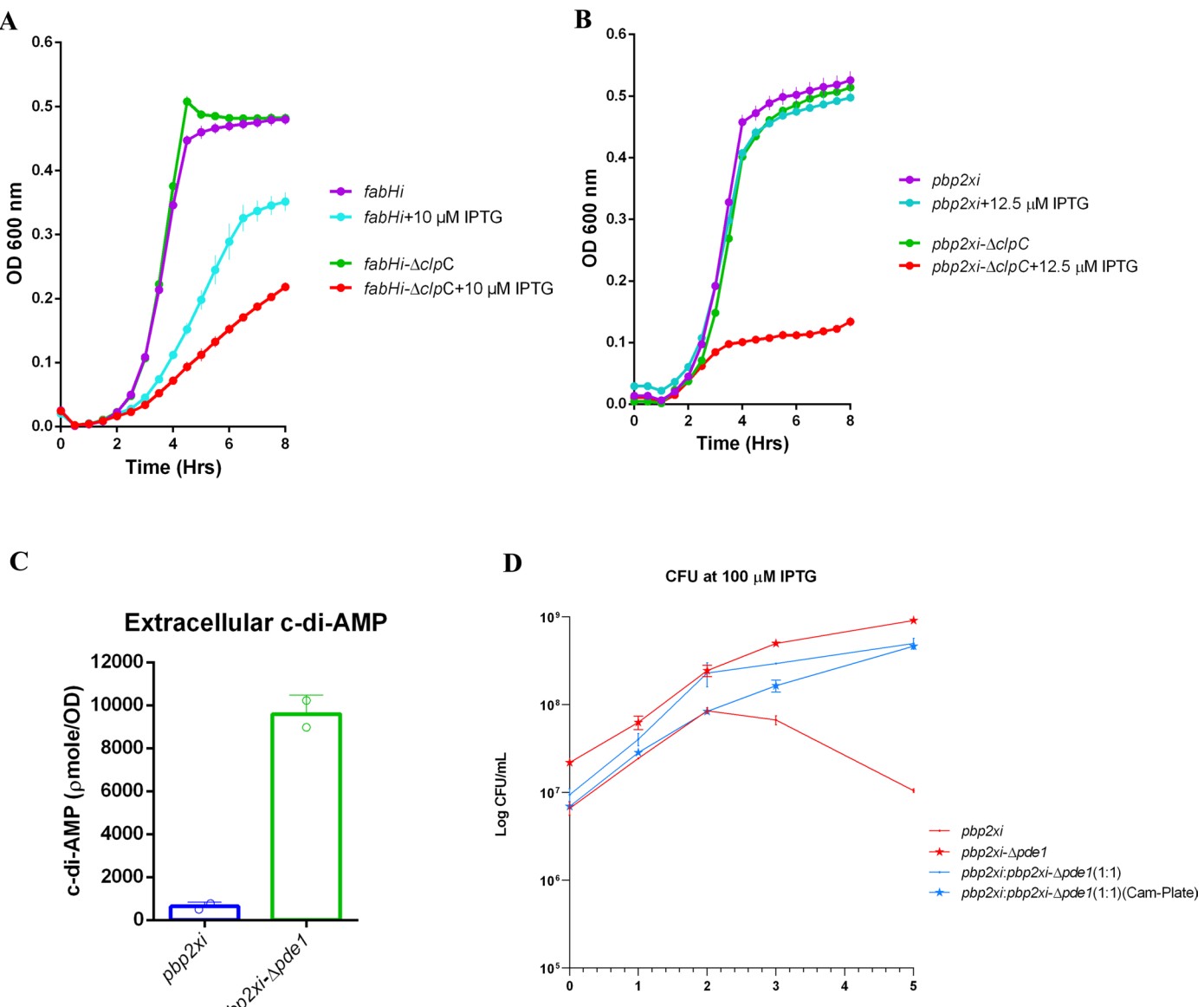

**Extended Data Fig. 7 | Growth, survival, and c-di-AMP levels of *S. pneumoniae* D39 CRISPRi strains.** Growth of ΔclpC and wild-type (wt) of *S. pneumoniae* D39 CRISPRi strains with (**a**) *fabH* or (**b**) *pbp2x* knockdown induced by supplementation of IPTG in C + Y growth medium. Average OD of n = 3 replicates is plotted with standard error. **c**. c-di-AMP levels in the growth medium of wild-type and Δpde1 in a *pbp2x*-CRISPRi background, measured by ELISA and presented in ρmole/OD after normalization by cell OD 600 nm. Average of (n = 2) biological replicate measurements is plotted with standard deviation; independent repetition (2) of the experiment showed similar results. **d**. CFU of *S. pneumoniae* D39 *pbp2x*-CRISPRi wild-type (wt), Δpde1, 1:1 mix of wt:Δpde1, and 1:1 mix of wt:Δpde1 grown on a chloramphenicol supplemented plate to select for Δpde1 colonies. Cultures were grown in C + Y with 100 μM IPTG, and then diluted serially before spotting on the plate. Wild-type CFU increased for the first 2 hours and then started to decline, indicating knockdown of *pbp2x*. Wild-type in a 1:1 mix with Δpde1 also showed a similar trend. As expected, Δpde1 CFU increased over time both in pure culture and a 1:1 mix with wild-type. Average CFU of 3 independent replicates is plotted with standard error.

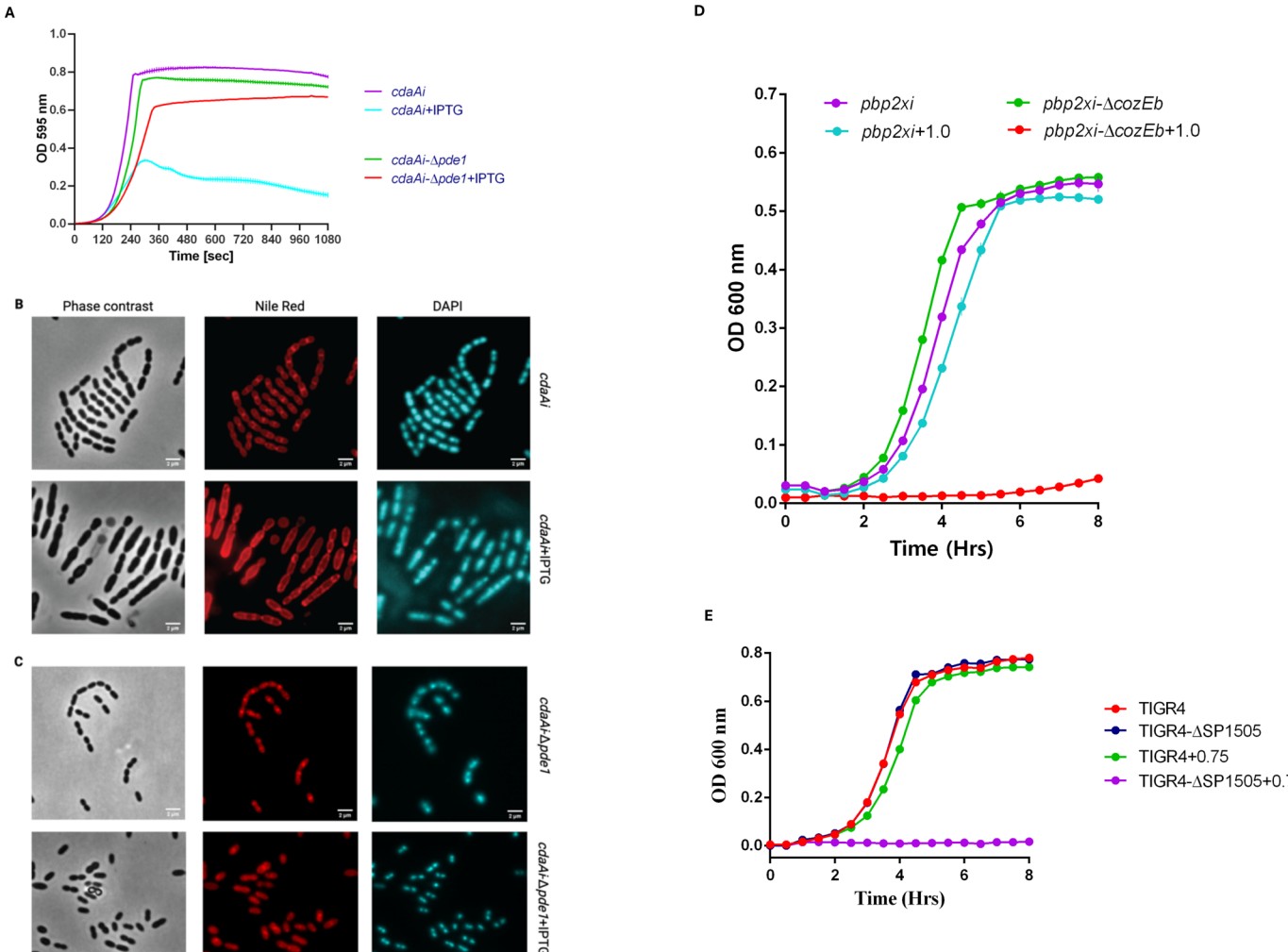

**Extended Data Fig. 8 | Growth and micrographs of wild-type and mutant of *S. pneumoniae* with and without (w/wo) CRISPRi knockdown. (a)** Both wild-type and Δ*pde1* were grown w/wo 100 µM IPTG in C + Y medium and OD was taken at 595 nm. Average OD of 4 replicates is plotted with standard error; independent repetition (2) of the experiment showed similar results. Microscopy images of *cdaAi* (**b**) and *cdaAi-Δpde1* (**c**) were captured w/wo 40 µM IPTG. Representative phase contrast, Nile red (red), and DAPI (blue) stained micrographs are presented with a 2 µm scale bar. Independent repetition (2) of the experiment showed similar results. c-di-AMP synthase or diadenylate cyclase (*cdaA*) synthesizes and maintains cellular c-di-AMP levels. Here, knockdown of *cdaA* inhibited the growth of wild-type but the impact was mild in Δ*pde1* (**a**). Enlarged cells at a coccus-to-rod transition, uneven chromosome distribution, and defective septa were

observed when *cdaA* was knocked-down in the wild-type (**b**), but it did not impact the morphology of Δ*pde1* (**c**). The impacts of *cdaA* knockdown were thereby to some extent like what was observed when *pbp2x* was knocked down presented in Fig. 3. This shows that knockdown of *cdaA* also impacts morphology. Such impact can also be suppressed by knocking out *pdeI*, possibly by restoring c-di-AMP levels. **d.** Growth of *S. pneumoniae* D39 *pbp2x*-CRISPRi wild-type (*pbp2xi*) and Δ*cozEb* (*pbp2xi-ΔcozEb*) with and without 1 µg/mL daptomycin. Average OD of 3 replicates is plotted with standard error; independent repetition (2) of the experiment showed similar results. **e.** Growth of Δ*cozEb* (TIGR4-ΔSP1505) and wild-type *S. pneumoniae* TIGR4 strains with and without 0.75 µg/mL of daptomycin. Average OD of 3 replicates is plotted with standard error; independent repetition (2) of the experiment showed similar results.

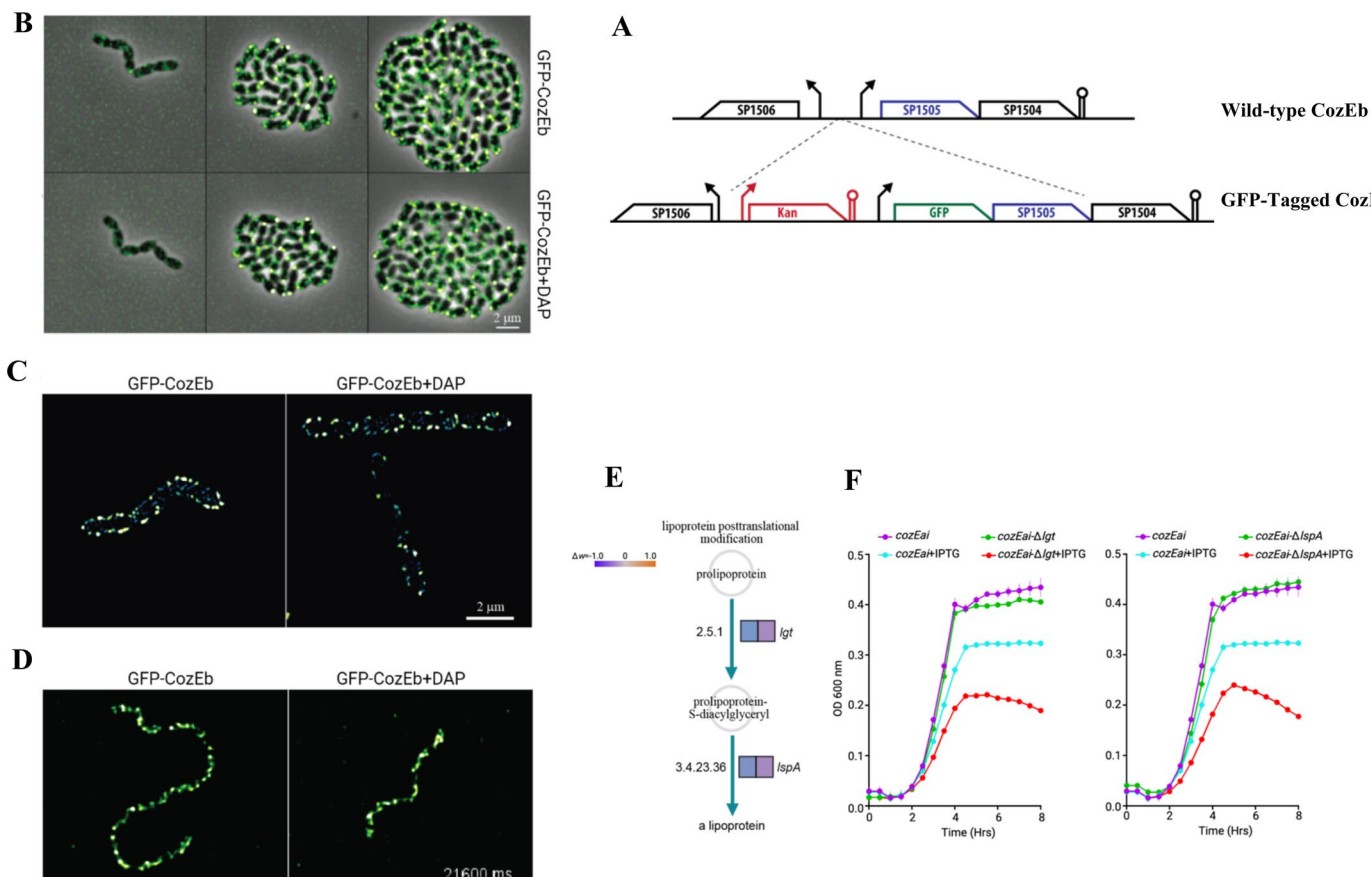

**Extended Data Fig. 9 | Cloning GFP-tagged CozEb (SP1505), growth, and microscopy of *S. pneumoniae*. a**. Cloning GFP-tag in front of CozEb (SP1505). The genomic region of *S. pneumoniae* TIGR4 from SP1506 to SP1504 is highlighted. GFP and Kanamycin resistance cassettes were inserted in the intergenic region located between SP1504 and SP1505. **b**. Microscopy of *S. pneumoniae* P$_{Zn}$-*gfp-cozEb* in the presence of daptomycin. Time-lapse fluorescence microscopy of P$_{Zn}$-*gfp-cozEb* in the absence and presence of 4 µg/mL daptomycin (+DAP). Representative micrographs are presented with a 2 µm scale bar; independent repetition (2) of the experiment showed similar results. **c**. 3D-structural illumination microscopy (3D-SIM) images of GFP-CozEb in live cells in the absence (left panel, chain of 3 cells) and presence (right panel, chain of 7 cells) of 5 µg/mL daptomycin. Signal intensity is represented by different colors, where blue is lower, green medium, and white higher signal. Representative micrographs are presented with a 2 µm scale bar. **d**. Snapshot

from time-lapse of GFP-CozEb (green signal) in live cells (cells outlines not shown) at 37 °C using TIRF (HILO) microscopy in the absence (left panel, chain of 16 cells) and presence (right panel, chain of 8 cells) of 5 µg/mL daptomycin. Images were taken every 45 ms. Representative micrographs are presented with a 5 µm scale bar. The movie can be found as Supplementary Data Movie 1. **e**. *cozEa* interacts with lipoprotein *lgt* and lspA. Lipoprotein posttranslational modification pathway of *S. pneumoniae*, including nonessential genes *lgt* and *lspA*. Both genes were identified in CRISPRi-TnSeq to have a negative interaction with *cozEa*. The heatmap values represent fitness reduction of Δ*lgt* or Δ*lspA* due to the knockdown of *cozEa* measured in a pooled assay. **f**. Growth curves of *cozEai*, *cozEai-*Δ*lgt*, and *cozEai-*Δ*lspA* in the presence and absence of 100 µM IPTG. Average of n = 3 replicates are plotted with standard error; independent repetition (2) of the experiment showed similar results.

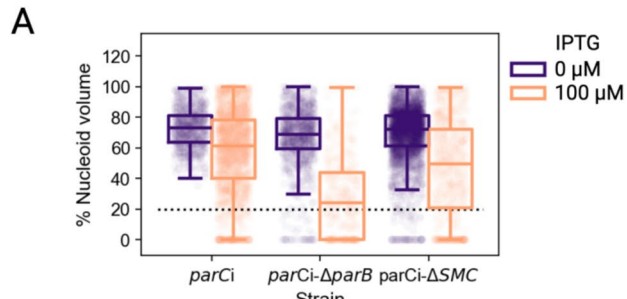

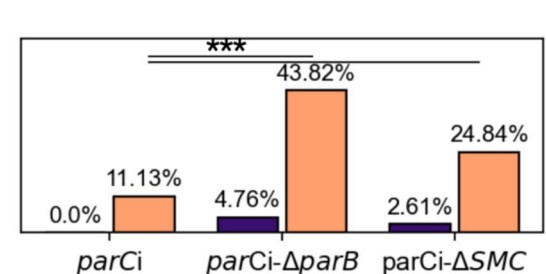

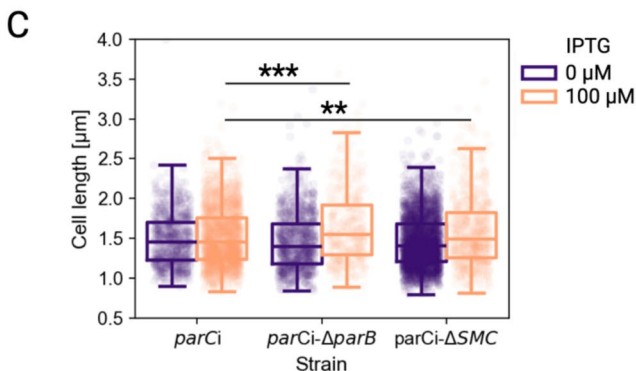

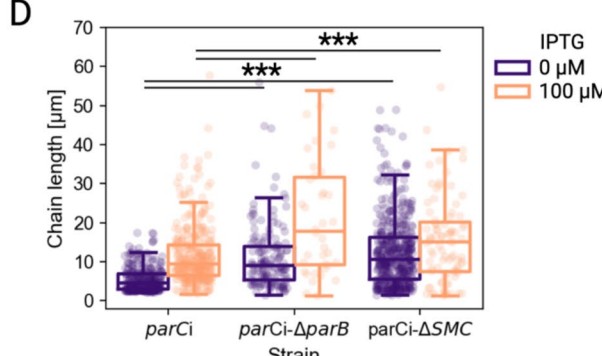

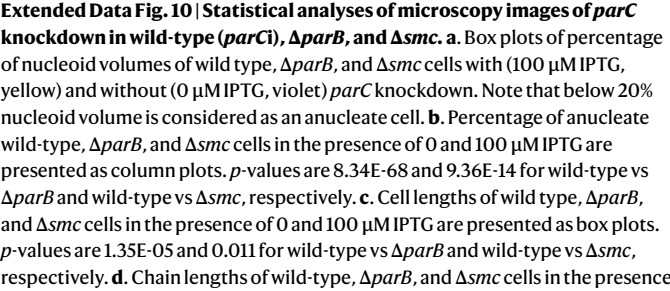

**Extended Data Fig. 10 | Statistical analyses of microscopy images of *parC* knockdown in wild-type (*parC*i), Δ*parB*, and Δ*smc*. a**. Box plots of percentage of nucleoid volumes of wild type, Δ*parB*, and Δ*smc* cells with (100 μM IPTG, yellow) and without (0 μM IPTG, violet) *parC* knockdown. Note that below 20% nucleoid volume is considered as an anucleate cell. **b**. Percentage of anucleate wild-type, Δ*parB*, and Δ*smc* cells in the presence of 0 and 100 μM IPTG are presented as column plots. *p*-values are 8.34E-68 and 9.36E-14 for wild-type vs Δ*parB* and wild-type vs Δ*smc*, respectively. **c**. Cell lengths of wild type, Δ*parB*, and Δ*smc* cells in the presence of 0 and 100 μM IPTG are presented as box plots. *p*-values are 1.35E-05 and 0.011 for wild-type vs Δ*parB* and wild-type vs Δ*smc*, respectively. **d**. Chain lengths of wild-type, Δ*parB*, and Δ*smc* cells in the presence

of 0 and 100 μM IPTG are presented as box plots. *p*-values are 8.95313E-05 and 0.001 for wild-type vs Δ*parB* and wild-type vs Δ*smc*, respectively. p-values of chromosome distribution and cell morphological differences between wild-type and mutants and with and without *parC* knockdown were calculated by two-sided Mann Whitney U test and images are labeled with *** (p < 0.001) or ** (p < 0.02) if the difference is significant. Details of statistical analyses are presented in Source Data Extended Fig. 10. The lower bound, centerline, and upper bound of the boxes in a. c. and d. represent the 1st (25th percentile), 2nd (median), and the 3rd quartile (75th percentile) of the dataset, respectively. The bottom and top whiskers represent values that deviate from the 1st and 3rd quartile by 1.5 times the interquartile range. Flier dots represent outliers.

|  |  |
|---|---|

# Reporting Summary

## Statistics

For all statistical analyses, confirm that the following items are present in the figure legend, table legend, main text, or Methods section.

| n/a | Confirmed | |
|---|---|---|
| ☐ | ☒ | The exact sample size (*n*) for each experimental group/condition, given as a discrete number and unit of measurement |
| ☐ | ☒ | A statement on whether measurements were taken from distinct samples or whether the same sample was measured repeatedly |
| ☐ | ☒ | The statistical test(s) used AND whether they are one- or two-sided *Only common tests should be described solely by name; describe more complex techniques in the Methods section.* |
| ☐ | ☒ | A description of all covariates tested |
| ☐ | ☒ | A description of any assumptions or corrections, such as tests of normality and adjustment for multiple comparisons |
| ☐ | ☒ | A full description of the statistical parameters including central tendency (e.g. means) or other basic estimates (e.g. regression coefficient) AND variation (e.g. standard deviation) or associated estimates of uncertainty (e.g. confidence intervals) |
| ☐ | ☒ | For null hypothesis testing, the test statistic (e.g. *F*, *t*, *r*) with confidence intervals, effect sizes, degrees of freedom and *P* value noted *Give P values as exact values whenever suitable.* |
| ☒ | ☐ | For Bayesian analysis, information on the choice of priors and Markov chain Monte Carlo settings |
| ☐ | ☒ | For hierarchical and complex designs, identification of the appropriate level for tests and full reporting of outcomes |
| ☐ | ☒ | Estimates of effect sizes (e.g. Cohen's *d*, Pearson's *r*), indicating how they were calculated |

*Our web collection on statistics for biologists contains articles on many of the points above.*

## Software and code

Policy information about availability of computer code

| | |
|---|---|
| Data collection | CRISPRi-TnSeq NGS data was collected on NextSeq sequencing platform (illumina). Growth data were collected on BioSpa spectrophotometer (Agilent). Fluorescence signal was recorded on Tecan Spectrometer. Microscopy images were captured on Olympus IX83 (Olympus) or Deltavision Elite (GE Healthcare). |
| Data analysis | CRISPRi-TnSeq NGS data was analyzed by Aerobio sequencing analyses platform, which is publicly available. Open source platform "Vega-Lite" and commercial software "GraphPad Prism 6" were used for fitness and growth data presentation, respectively. Microscopy images were processed by open software "Image J" and analyzed by python package "MOMIA" and "Scikit-image", available on Gitgub. 3D-Structural Illumination Microscopy images were processed by commercial software "SoftWoRx (GE Healthcare)". Gene set enrichment analysis (GSEA) was performed using open source GSEA 4.1.0 platform. Pathway analyses were performed using open source web-platform BioCyc. Genetic and chemical genetic interaction datasets were visualized with open source software Cytoscape 3.8.0. |

For manuscripts utilizing custom algorithms or software that are central to the research but not yet described in published literature, software must be made available to editors and reviewers. We strongly encourage code deposition in a community repository (e.g. GitHub). See the Nature Portfolio guidelines for submitting code & software for further information.

## Data

Policy information about availability of data

All manuscripts must include a data availability statement. This statement should provide the following information, where applicable:
- Accession codes, unique identifiers, or web links for publicly available datasets
- A description of any restrictions on data availability
- For clinical datasets or third party data, please ensure that the statement adheres to our policy

> All sequence data can be found in the NCBI Sequence Read Archive under the BioProject: PRJNA813307. NCBI Reference Genome Sequence file NC_008533.2 was used for the analysis of CRISPRi-TnSeq data. Source data for the figures are available in the corresponding source data tables.

## Research involving human participants, their data, or biological material

Policy information about studies with human participants or human data. See also policy information about sex, gender (identity/presentation), and sexual orientation and race, ethnicity and racism.

| | |
|---|---|
| Reporting on sex and gender | This is a study on bacterial genetic interaction mapping, so sex is not relevant. |
| Reporting on race, ethnicity, or other socially relevant groupings | This is a study on bacterial genetic interaction mapping, so race is not relevant. |
| Population characteristics | This is a study on bacterial genetic interaction mapping, so population characteristics is not relevant. |
| Recruitment | This is a study on bacterial genetic interaction mapping, so recruitment is not relevant. |
| Ethics oversight | This is a study on bacterial genetic interaction mapping, so ethics is not relevant. |

Note that full information on the approval of the study protocol must also be provided in the manuscript.

# Field-specific reporting

Please select the one below that is the best fit for your research. If you are not sure, read the appropriate sections before making your selection.

☒ Life sciences  ☐ Behavioural & social sciences  ☐ Ecological, evolutionary & environmental sciences

For a reference copy of the document with all sections, see nature.com/documents/nr-reporting-summary-flat.pdf

# Life sciences study design

All studies must disclose on these points even when the disclosure is negative.

| | |
|---|---|
| Sample size | 6 samples/condition. In previous articles we demonstrated that 6 samples per condition provides data with statistical rigor (van Opijnen et al., Nat Methods 2009; van Opijnen et al., Genome Res. 2012; van Opijnen et al., Curr. Protoc. Mol. Biol. 2014). |
| Data exclusions | When growth inhibition is severe Tn-Seq identifies non-specific interactions, therefore we excluded those samples which is described in the Materials section. |
| Replication | Each CRISPRi-TnSeq experiment was performed with at least 2 different IPTG concentrations and with 6 individual libraries. Growth and gene expression experiments were performed with 3 biological and 3 technical replicates. Independent repetition of all experiments produced similar results. |
| Randomization | A random genetic interaction dataset was created by permutation to compare with the experimentally observed dataset. This comparison shows that observed dataset is enriched with negative genetic interactions. |
| Blinding | While blinding is not relevant, extensive statistics and validation experiments all but guarantee the robustness of the dataset. |

# Reporting for specific materials, systems and methods

We require information from authors about some types of materials, experimental systems and methods used in many studies. Here, indicate whether each material, system or method listed is relevant to your study. If you are not sure if a list item applies to your research, read the appropriate section before selecting a response.

## Materials & experimental systems

| n/a | Involved in the study |
|-----|----------------------|
| ☐ | ☒ Antibodies |
| ☒ | ☐ Eukaryotic cell lines |
| ☒ | ☐ Palaeontology and archaeology |
| ☒ | ☐ Animals and other organisms |
| ☒ | ☐ Clinical data |
| ☒ | ☐ Dual use research of concern |
| ☒ | ☐ Plants |

## Methods

| n/a | Involved in the study |
|-----|----------------------|
| ☒ | ☐ ChIP-seq |
| ☒ | ☐ Flow cytometry |
| ☒ | ☐ MRI-based neuroimaging |

# Antibodies

| | |
|---|---|
| Antibodies used | Anti-Cyclic-di-AMP antibody |
| Validation | The above antibody was purchased with ELISA assay kit of Cayman chemicals, catalog number 501960. Details can be found on company's website "https://www.caymanchem.com/product/501960/cyclic-di-amp-elisa-kit". |

# Plants

| | |
|---|---|
| Seed stocks | n.a. |
| Novel plant genotypes | n.a. |
| Authentication | n.a. |

