## [Peer Review File · Nature Microbiology]

Peer Review Information

Journal: Nature Microbiology

Manuscript Title: CRISPRi-TnSeq: A genome-wide high-throughput tool for bacterial essential-nonessential genetic interaction mapping

Corresponding author name(s): Professor Tim van Opijnen

Reviewer Comments & Decisions:

Decision Letter, initial version:

Message: 20th July 2023

Dear Professor van Opijnen,

Thank you for your patience while your manuscript "CRISPRi-TnSeq: A genome-wide high-throughput tool for bacterial essential-nonessential genetic interaction mapping" was under peer-review at Nature Microbiology. It has now been seen by 4 referees, whose expertise and comments you will find at the end of this email. Although they find your work of some potential interest, they have raised a number of concerns that will need to be addressed before we can consider publication of the work in Nature Microbiology.

In particular, you will see that while they find your proposed workflow, of combining CRISPRi knockdown with TnSeq of some potential interest to the community, the referees raise multiple concerns over several aspects of the study. In particular, there were concerns over a lack of important controls including the use of non-targeting gRNAs, and possibility of other mutations or polar effects influencing the interpretation of the data. Several referees highlighted a need for further validation of interacting gene pairs identified in the gene interaction network and there were also concerns over reproducibility of some data and a need for more clarity in methodological details, replicates, statistical analysis and data presentation, as well as a need to tone down some conclusions. Referee #3 also raised the point that several studies using combinatorial CRISPRi knockdown approaches to explore gene interaction networks had been published which might detract also from the novelty of this approach. We feel that these are important points that need to be addressed for us to consider a revised manuscript, and that such a manuscript should discuss the advance presented by this CRISPRi-Tnseq approach over previously published methods for interrogating gene interaction networks. The rest of the referees' reports are clear and the remaining issues should be straightforward to address.

Should further experimental data allow you to address these criticisms, we would be happy to look at a revised manuscript.

We strongly support public availability of data. Please place the data used in your paper

2into a public data repository, if one exists, or alternatively, present the data as Source Data or Supplementary Information. If data can only be shared on request, please explain why in your Data Availability Statement, and also in the correspondence with your editor. For some data types, deposition in a public repository is mandatory - more information on our data deposition policies and available repositories can be found at <https://www.nature.com/nature-research/editorial-policies/reporting-standards#availability-of-data>.

Please include a data availability statement as a separate section after Methods but before references, under the heading "Data Availability". This section should inform readers about the availability of the data used to support the conclusions of your study. This information includes accession codes to public repositories (data banks for protein, DNA or RNA sequences, microarray, proteomics data etc...), references to source data published alongside the paper, unique identifiers such as URLs to data repository entries, or data set DOIs, and any other statement about data availability. At a minimum, you should include the following statement: "The data that support the findings of this study are available from the corresponding author upon request", mentioning any restrictions on availability. If DOIs are provided, we also strongly encourage including these in the Reference list (authors, title, publisher (repository name), identifier, year). For more guidance on how to write this section please see: <http://www.nature.com/authors/policies/data/data-availability-statements-data-citations.pdf>

* If you have not done so already we suggest that you begin to revise your manuscript so that it conforms to our Article format instructions at <http://www.nature.com/nmicrobiol/info/final-submission>. Refer also to any guidelines provided in this letter.

When submitting the revised version of your manuscript, please pay close attention to our [href="https://www.nature.com/nature-portfolio/editorial-policies/image-integrity">Digital Image Integrity Guidelines](https://www.nature.com/nature-portfolio/editorial-policies/image-integrity). and to the following points below:

- that unprocessed scans are clearly labelled and match the gels and western blots presented in figures.
- that control panels for gels and western blots are appropriately described as loading on

2sample processing controls

-- all images in the paper are checked for duplication of panels and for splicing of gel lanes.

Note: This url links to your confidential homepage and associated information about manuscripts you may have submitted or be reviewing for us. If you wish to forward this e-mail to co-authors, please delete this link to your homepage first.

Nature Microbiology is committed to improving transparency in authorship. As part of our efforts in this direction, we are now requesting that all authors identified as 'corresponding author' on published papers create and link their Open Researcher and Contributor Identifier (ORCID) with their account on the Manuscript Tracking System (MTS), prior to acceptance. This applies to primary research papers only. ORCID helps the scientific community achieve unambiguous attribution of all scholarly contributions. You can create and link your ORCID from the home page of the MTS by clicking on 'Modify my Springer Nature account'. For more information please visit please visit www.springernature.com/orcid.

If you wish to submit a suitably revised manuscript we would hope to receive it within 6 months. If you cannot send it within this time, please let us know. We will be happy to consider your revision, even if a similar study has been accepted for publication at Nature Microbiology or published elsewhere (up to a maximum of 6 months).

Yours sincerely,

Reviewer Expertise:

Referee #1: TnSeq, microbial genetics

Referee #2: CRISPRi, Streptococcus biology, protein-protein interactions

Referee #3: gene-protein interaction networks, systems biology, tools

Referee #4: S. pneumoniae cell biology

Reviewer Comments:

Reviewer #1 (Remarks to the Author):

Jana and colleagues present a new approach for investigate genetic interactions in bacteria by combining two existing approaches, CRISPR inference (CRISPRi) and transposon site sequencing (tn-seq), using *Streptococcus pneumoniae* as a model. In this workflow, a CRISPRi gene knockdown of an essential gene is performed in the background of a genome-wide tn-seq library, and the resulting “double mutant” library is assayed under standard rich media condition both in the absence and presence of an inducer to repress essential gene expression. The authors use 13 essential genes as queries, representing genes from different functional categories including cell wall synthesis, cell division, and DNA replication. Overall, the authors identify over 1,300 genetic interactions (both positive and negative) with non-essential genes. From their networks, they conclude that some non-essential genes with frequent genetic interactions with their 13 queries represent potential genetic buffers against mutation and environmental challenge. They then dissect certain parts of their genetic interaction network with more follow-up investigations and demonstrate new linkages between septal and peripheral peptidoglycan synthesis, cell wall synthesis and cellular division, and cyclic-di-AMP levels and cell turgor.

Overall, the new CRISPRi-TnSeq approach is a valuable addition to the bacterial functional genomics toolbox, as existing methods for studying genetic interactions at-scale in bacteria are limited. For instance, the older synthetic genetic array (eSGA) approach developed for *E. coli* requires conjugation, do not take advantage of next-generation sequencing, and the phenotyping is only semi-quantitative. It’s been 15 years since the publication of the initial eSGA paper, and I am not aware of this approach was ever applied outside of *E. coli*. So, I’m enthusiastic about CRISPRi-TnSeq, because as the authors note, both approaches individually are now well-established in many bacteria (including those that are “non-model”). Therefore, I can see a rapid path towards the implementation of a CRISPRi-TnSeq approach (maybe not exactly what the authors did here, see below) in diverse bacteria to uncover novel gene-gene associations, and to gain new insights that are missed with standard single-gene genetics.

Comments:

- L46: The authors should define a genetic interaction in the Introduction, even if briefly. In particular, that the measured phenotype of the double mutant (in this case a mutant and a gene knockdown) deviates from the expected phenotype based on the single mutants. Furthermore, it would be nice to distinguish “synthetic lethal” and negative genetic interactions, as the sometimes intermixed use of these terms in the paper can lead to confusion. Classically, synthetic lethals are cases where the combined loss of two non-essential genes leads to an inviable phenotype. In this work, one of the genes under study is always an essential gene, so technically no synthetic lethal gene pairs can be identified. This is a little nit-picky, but I think these distinctions are important to make.
- L50-52: Rephrase, reads as if CRISPRi is a TIS tool. Maybe group TIS and CRISPRi as bacterial functional genomics tools.
- L75: Sounds like all of the approaches mentioned above are array-based, but the human example (ref 9) is a pooled-based assay with next-generation sequencing.
- L117 and in Figure 1: Recommend changing the subpanels to letters to match the rest of the manuscript.

4

- L125-8: How is the expected phenotype of the transposon mutant-knockdown strain phenotype? This is important because it is the measured deviation from these values that defines a genetic interaction. In particular, what is the measured fitness of the knockdown strain alone at the different levels of induction? Or do the authors just assume that this value is uniform across the entire mutant library (because all strains have the same knockdown cassette), so that any deviation in fitness of the transposon mutants in the absence and presence of induction constitutes a genetic interaction (i.e. what's shown in Figure 1.5)? If this is the case, then it would be good to explicitly mention this here and/or in the Methods section.
- L130: Is it 12 essential genes? Because *clpP* is non-essential?
- L136-8: It appears that the uninduced libraries have varying fitness defects, suggesting some leakiness to the IPTG-inducible system (growth values in panel B range from ~0.25 for *parC* to ~0.45 for *atpF*). What is the likely reason for these different fitness values, and is there any impact on the identification of genetic interactions?
- L140: units should be μM .
- Figure 2: Panel B is hard to see due to its size and resolution, especially the y-axis labels. As far as I can tell panel C is not referred to in the manuscript. In panel C, where are the gene functional categories derived from? The meaning of the inset figure in panel G is unclear. In panel H, a legend showing the scale of the fitness values is needed.
- L157: I didn't see a succinct table that presents just the list of 1,334 gene pairs with genetic interactions, which would be useful to readers. Also, it would be nice to mention that some query genes had few interactions (*atpF*), while others had many (*ftsZ*).
- If I understand Supplementary Table S3, then ~half or more of the non-essential genome has a negative genetic interaction with *pbp2x* and *ftsZ* if you add enough IPTG to the system. For instance, at 25 μM IPTG, *pbp2x* has a negative genetic interaction with 1,266 different non-essential genes! This seems odd to me, because this must include many genes that only have a very indirect relationship to cell wall synthesis. At a lower induction level, there's considerably fewer genes (44) with a negative interaction with *pbp2x*. Obviously, filtering for informative interactions from >1,000 genes is not ideal, so it's clear that the induction level is critical for identifying a manageable number of interactions in the authors' CRISPRi-TnSeq approach. I'm curious if the insights that the authors draw from *pbp2x* are from the lower induction level (or example, the network presented in Figure 5A), or if they are selectively including genes in these networks based on the narrative they want to tell. Did the higher level of induction cause severe growth inhibition for these two genes? Unfortunately, these data are not included in Figure 2B.
- L172-4: Figure 2E and Supplemental Figure 3 are low resolution. I also recommend removing panel E from Figure 2 to make additional space for the other panels.
- L175: Figure 2F does not show the enrichment of genetic interactions around SPD2014-SPD2064.
- L188-9: How enriched are these gene functional categories in the *parC* and *fabH* datasets? And with what statistical significance?
- Figure 3: Explicitly state in the legend that the clustering of experiments in panel B is based on the entire datasets, and that only select genes are being shown in the heatmap. Also, a legend showing the scale for the fitness values in B should be included.
- Figure 4: The networks in panels A and B are low resolution. In particular, the gene labels in the panel B network nodes are nearly impossible to read. In panel C, what pairs of genes were tested for the 27 validations, and was the same IPTG concentration used in the validation and in the genome-wide data?

- L233: 7 interactions validated or 27 validated? The reference to Figure 4C seems to signify the latter.
- L234-9: Do these 7 non-essential genes with frequent negative genetic interactions with the essential query genes have negative fitness values in the C+Y growth medium used for the genome-wide assays? Is the interpretation that mutants in these 7 genes are already sick, and so an expression knockdown of an essential gene just exasperates the fitness defect.
- L239-47: Similar to above, what are the phenotypes of these 10 genes on C+Y (are they detrimental, i.e. transposon mutants have a fitness advantage in the pooled assay)?
- L250: Check spelling of pleiotropic.
- Figure 5: Gene labels in network in panel A are hard to read. Panels B and C would be easier to follow if the colors matched by strain. Why was a different IPTG concentration used in B and C?
- L311: This implies that node pvaA in Figure 5A should be colored blue.
- L353-4: Specify the overall overlap of genetic interactions between cozEa and pbp2x. Are these genes the most connected in the network?
- L372: overtime should be two words.
- Figure7: Why isn't the growth defect of pbp2xi+IPTG more severe (in panel B)? Was a lower concentration of IPTG used to highlight the more prominent fitness defect in ta delta-cozEb background? More broadly, the authors should indicate the concentration of IPTG used in each assay/figure panel, because it's possible that different concentrations were used to highlight the phenotypes (the growth curves and the cell images). In panels CDE, what is the expected impact of adding daptomycin? I'm not sure what I'm supposed to distinguish between the +DAP and -DAP images.
- Figure 8: Indicate the IPTG concentration used, in particular in panels A and B. The cozEai+IPTG strain is only moderately sick, so I assume that a low IPTG concentration was used. In panel G, what do the lgt and lspA heatmap values mean?
- L391: There's no rodZ data in Figure 8B.
- L399:400: The interaction between cozEa and ctpE is not that strong.
- L470-2: Another CRISPRi-TnSeq variation is to move the different CRISPRi query gene constructs into an existing transposon mutant library. The advantage here is that one wouldn't need to construct a new Tn-Seq library for each query gene under investigation.
- Discussion: One limitation of the current study is that only a single condition (C+Y media) was profiled. For many putative genetic interactions (for example, for metabolic genes or transporters), the GI will only be apparent under certain conditions. This could be touched upon briefly in the Discussion.
- L503-4: It appears that biological replicates were only performed in a few instances. Why? In the cases where replicates were performed, what the agreement between the assays?
- L518-24: Are the fitness values for different insertions in the same gene averaged?
- L526-28: How is the multiplicative fitness value calculated ($W_a \times W_b$)? More specifically, if W_b is the fitness of the CRISPRi gene knockdown, where does this value come from?
- L526:43: The code used to calculate delta-W and its associated statistics should be made publicly available.
-

Reviewer #2 (Remarks to the Author):

6This paper by Jana et al. presents CRISPRi-TnSeq, a method that combines CRISPR interference (CRISPRi) and transposon sequencing (TnSeq) to map interactions between essential and nonessential genes in *S. pneumoniae*. The process involves engineering a bacterial strain with CRISPRi to knock down essential genes and using a transposon mutant library to knock out nonessential genes. By analyzing the fitness effects of these gene manipulations, the study identifies and categorizes negative and positive genetic interactions. The results show a significant number of genetic interactions, with a higher concentration near the *oriC* region. Negative interactions are more prevalent, particularly among genes involved in replication, transcription, and energy metabolism.

The authors use multiple strategies to validate CRISPRi-TnSeq. First, they use antibiotic-TnSeq methods to identify overlapping interactions between nonessential genes and either CRISPRi-targeted or antibiotic-targeted essential genes. Hierarchical clustering purports to reveal gene clusters involved in specific biological processes, confirming known associations and discovering new ones.

The study also uncovers pleiotropic genes that interact with multiple essential genes, enhancing the organism's sensitivity to essential gene knockdown. These pleiotropic genes are proposed as "genetic capacitors" that protect against environmental and genetic stresses.

Furthermore, the study examines specific genetic interactions by creating nonessential gene deletion mutants in CRISPRi knockdown strains. Growth rates are measured with and without IPTG, allowing for fitness calculations affected by nonessential gene deletion and simultaneous CRISPRi-mediated target knockdown. The observed interactions align with those identified through CRISPRi-TnSeq analysis. Specific interactions, like *fabH* and *plsX*, reveal compensatory mechanisms between genes. Several additional experiments are reported that begin to examine mechanisms of interaction between essential and nonessential genes, particularly as they relate to cell wall synthesis, cell division, and morphology.

Specific critiques:

-Overall, this is a compelling paper that presents a new frontier in mapping bacterial genetic networks. By combining the ability to dynamically repress essential genes (CRISPRi) with TnSeq's ability to quantify nonessential gene fitness data in a massively multiplexed system, the strategy combines the strengths of the two approaches.

-One important control seems to be missing. How do the authors know that there are not off-target effects of their CRISPRi system? If the essential genes being targeted were even slightly confounded by nonspecific targeting of other chromosomal regions by dCas9, that would seriously compromise the networks that emerge. If it were a fixed nonspecific effect, those genes affected could end up looking like pleiotropic contributors to fitness because they would show up (erroneously) as being linked to multiple knocked down essential genes. The control to ensure this isn't the case would be to perform CRISPRi-TnSeq with one (or several) non-targeting/scrambled guide RNA templates. When the system is used with this design, there should be no significant difference from Tn-seq performed with WT.

7-The writing clear and easy to follow. However, the paper is very long and tries to capture a possibly overzealous number of topics. Many of my comments below are aimed at tightening the core story, which is a good one, by discarding and consolidating some of the more extraneous strands.

-The validation sections overuse speculative language. A good example of this is lines 182-186 ("Reducing ATP MAY thus also buffer against more general stress...atpF knockdown MIGHT increase intracellular pH...which MAY additionally contribute to buffer against stress"). Some of this can be moved to Discussion or removed altogether. The data can be presented as validating the technique without positing mechanisms for which there is not proof.

-Figure 1H seems out of place and the accompanying discussion doesn't really lead anywhere in the rest of the paper. I would remove this.

-Figure 2 requires extensive revision. Panel 2B is not well explained and doesn't—on the surface—make a lot of sense. The clustering does not look convincing. For instance, why does atpF15 cluster with atpF30 when the columns below look so divergent? How was this done? What scale is in the heatmap? What genes (or gene sets?) are on the vertical and horizontal axes? Currently very confusing.

-Figures 7 and 8 (and accompanying discussion of CozEa-related results): In my opinion, the strongest data in these two figures are the gene network map combined with the growth curves. Much of the rest of it, including the microscopy, is distracting, hard-to-interpret, and peripheral to CRISPRi-TnSeq (the photomicrographs in Figure 5 are more convincing and more directly related to the main thrust of the paper). Much of the related verbiage in the results section leans in a speculative direction. I would COMBINE and SIMPLIFY Figures 7 and 8, making a single figure that shows a SIMPLIFIED network with the genes whose interactions you're focusing on (leave out the many shown that you're not) and the relevant growth curves. The Figure 7 and 8 microscopy could, I feel, be moved to Supplemental.

-Figures throughout: Using white circles with thin outlines for the essential genes in your gene networks strikes me as a mistake. They're very hard to see (actually invisible in Figure 1A). Would choose a different color and possibly a different shape for these important network members.

-What is the difference between a "genetic capacitor" and a stress response? Aren't they both systems that interact with a lot of genes to help the cell adapt to an external or internal disruption? Do genes involved in canonical stress responses (e.g. the stringent response) show up as mediators of essential gene manipulation? If not, isn't that surprising?

-The technical details of performing CRISPRi-TnSeq in pneumococcus are reasonably well laid out, since both CRISPRi and TnSeq in this organism have been previously described in great detail. The authors' claim that the system would be easily transferred to any other bacterium is dubious, however. Neither TnSeq nor CRISPRi are—at this point—universally

available, and the steps needed to institute them in non-model bacteria are nontrivial. It seems reasonable to say that the system should be adaptable to other organisms, but only in cases where the two halves of the approach are established. To this end, I would remove the last sentence of the abstract as currently written.

-The Methods subsections outlining statistical analysis of the data (under the headings "Analysis of CRISPRi-transposon mutant libraries" and "Genetic interaction mapping") are too vague. Specifically, it's not clear enough how a genetic interaction is determined to be significant.

Minor comments:

Throughout the paper (e.g. lines 169, 171) the authors use mM whereas in Figures it's μM .

Line 372: "Over time" not "overtime."

Reviewer #3 (Remarks to the Author):

This manuscript by Jana et al. describes the development of an approach called CRISPRi-TnSeq that attempts to determine genetic interactions between an essential gene and all non-essential genes in *S. pneumoniae*. Their system is a simple composition of a single-gene CRISPR-interference followed by whole-genome transposon mutagenesis and sequencing. IPTG-induction of dCas9 enables control over the context and degree of essential gene knock-down. Comparison of gene transposon insertions in IPTG induced vs non-induced populations enables them to quantify positive/negative genetic interactions between the essential gene and all other non-essential genes in the genome. In motivating the utility of their approach, the authors imply that existing systems cannot easily identify essential x non-essential genetic interactions in a high-throughput manner. However, this is not true. In fact, Jiang et al. (*Cell*, 180(5):1002 (2020)) and recently Ellis et al. (<https://elifesciences.org/reviewed-preprints/86903>) have shown that libraries of double CRISPRi knockdowns can be generated in bacteria and enable study of pairwise and even multiplex genetic interactions, including essential x non-essential and in particular, essential x essential genes which is beyond the capacity of the authors' CRISPRi-TnSeq approach. This is highly relevant work that the authors don't seem to be aware of. As such, the technological novelty of the work is not that significant. However, the authors apply their approach to a collection of essential genes (enriched for antibiotic targets) to discover suppressive (positive) and aggravative (negative) genetic interactions. Out of a total of 24,000 potential interactions, they find 1,334 that are significant, with 754 negative and 580 positive. The authors then validate a set of these interactions and with the help of previous work/knowledge of the genes and further genetic experiments/imaging propose potential hypotheses for the mechanism by which these interactions arise. The areas of focus include lipid biosynthesis, connection between septal and peripheral peptidoglycan synthesis, the role of cyclic-di-AMP on intracellular K^+ and turgor, connections between peptidoglycan synthesis and cell morphology/envelope integrity, and connections between chromosome decantation and segregation. These are interesting vignettes that demonstrate the power of performing systematic genetic interaction studies. As a collective, they add incrementally to the long literature on the topic extensively explored in

9previous work. As individual stories, they would be of interest to *S. pneumoniae* cell biologists as brief forays that require further experimental follow-up.

A major high-level conclusion of the work, is summarized in the abstract: “we identify a set of 17 pleiotropic genes, of which a subset tentatively functions as genetic capacitors, dampening phenotypic outcomes and protecting against perturbations”. They further state that these genes have evolved to “induce robustness into a biological systems to protect against environmental or genetic insults”. They, in fact, claim that the activity of these genes is similar to the buffering properties of the HSP90 chaperone. This is quite a far reach and not at all supported by any of their experimental observations. Let me briefly explain why. The authors find 1,334 significant genetic interactions among 24,000 gene pairs. It is entirely not surprising that one would find a small subset of highly connected nodes in the resulting genetic interaction network. This just reflects a set of recurrent connections between biological processes that are connected to each other at a higher level of organization. There is no need to over-interpret these genes as having a special property of buffering akin to HSP90.

Other concerns:

- 1) Design of gRNA, off-target effects and polar effects. The authors should discuss how their gRNAs are designed as off-target effect is always a possibility. Ideally the authors should design at least two gRNAs free of off-target effect for each essential gene they are targeting, and use them as biological replicates of each other. The other concern is the polar effect of gRNAs as they can silence all downstream genes in the targeted operon. This is especially problematic if different genes in the operon encode very different functions. This is the case for a number of genes the authors are studying. For example, *gyrA* (DNA replication) is followed by *srtA* (cell wall) in an operon; *folA* (DNA/metabolism) is followed by multiple other genes including *clpX* (protein degradation) in an operon. This means the authors cannot be certain if the genetic interactions they study are for the gene of interest or a neighboring gene encoding a different function. Authors need to really clarify this.
- 2) Data reproducibility. The author selected 13 essential genes and studied their interactions with the rest of the genome (ie, 13-vs-all interactions). However, it's not clear to me how reproducible these datasets are among biological replicates, as they are not presented. Can the authors comment on that?
- 3) The validity of the clustering in Fig. 3B looks suspicious as some closely clustered columns don't even look similar (eg, *rpoC15* and *rpoC30* are right next to each other but the profiles clearly look different. Another example is *ftsZ15* and *ftsZ30*).
- 4) The growth consequences of IPTG-induced knockdown seem to be highly variable in different experiments. For example, there is a large difference in the effect of *pbp2x* knockdown between different figures. For example, compare *pbp2xi*+IPTG between figure panels 5B and 7B. What is this caused by? Are the authors using different IPTG concentrations in each experiment? In general, these growth curves should have replicates with error bars throughout.
- 5) Throughout the work, the authors should have more nuanced interpretations of experimental observations. For example, bottom of page 13, the authors state: “This suggests that a reduced amount of CozEa results in CtpE dysregulation and changes in intracellular calcium that lead to defects in cell envelope integrity”. What are alternative explanations?

106) Microscopic observations are sometimes subtle. The authors could be more quantitative throughout. For example, on page 14, the authors state: "Fluorescence microscopy shows that parC titration results in anucleate cells and uneven chromosome distribution (Fig. 9D), highlighting that chromosome distribution is dependent on decatenation by parC. Deletion of smc results in enhanced chaining (Fig. 9E), indicating that SMC-mediated chromosome segregation affects downstream cell division." They could quantify differences in anucleate cells, uneven chromosome distribution, and chaining.

7) At the end of the Discussion section, the authors propose future advances over the work presented here: "By incorporating two guide RNAs into a single cell, a dual-guide CRISPRi tool would have the potential to become a broadly applicable approach for genome-wide genetic interaction mapping. We are actively working on both approaches as we foresee that each will have its own advantages and disadvantages with respect to questions that can be answered as well as their differential ease of implementation and optimization into a new strain or species". As discussed above, these advances have already been made in the recent literature (Jiang et al, 2020 and Ellis et al, 2023).

Reviewer #4 (Remarks to the Author):

Comments to authors

This paper combined two powerful genetic screening tools of CRISPRi and Tn-Seq to identify genetic interaction between essential and non-essential genes on a genome-wide scale. This technique has great potential for studying gene interactions when one gene is essential. CRISPRi strains were obtained for 13 essential genes involved in different cellular processes in *Streptococcus pneumoniae*, and >1300 genetic interactions were detected in the genetic screen.

This paper will be very useful for research groups that would like to apply the method to other essential genes of *S. pneumoniae*, or to other organisms. Because of this reason, the methods, analysis and validation should be carefully documented. However, I find the details of the strain construction missing, the analysis tools and rationale not well-explained, and the validation data set small. Only 27 of the >1300 genetic interactions (2%) were shown in a figure. It is likely that there are others that were experimentally checked, but not included on the graph (for example the data on cosEa and cosEb, and many of the strains in the strain table). A reasonable proof-of-concept validation data set should include about 5 to 10 % of the data set.

This paper reports a potentially valuable genetic suppressor screen and interesting observations with some of suppressor pairs. This combined technique will advance the knowledge in the field, not just of *Streptococcus pneumoniae*, but also of other organisms. However, the lack of clear labels of supplemental tables and supplemental figures, incomplete explanation of data analysis concerning significant genetic interactions, inconsistency with the data in different figures, and incorrect information on certain gene essentiality (e.g., rodZ) make it difficult to understand and evaluate the results. I recommend that the authors provide a detailed and careful report on the methods,

11analysis, and validation results. The current manuscript draws big conclusions on many different aspects of gene interactions without thorough justifications. It may be more effective to concentrate on one topic of interest, such as the pbp2x and pde1 interaction, and provide extra experimental evidence for any model.

General comments on presentation and formats of supplemental tables and figures

1. All supplemental tables should have a heading of supplemental table # and a title of the table. Tables should have footnotes to explain the items on the table if needed. The lack of table numbers and titles makes it hard for readers to find the data.

2. Make sure all texts and lines in figures and supplemental figures are clearly visible. The black gene labels on dark circle in Fig. 4,5,7, and 9 are hard to read. The lines connecting the gene are too light to distinguish whether they are blue or red, or whether there are intensity differences in the lines. The bands in supplemental figure 1 are not visible.

Comments on presentation of methods to generate the strains for analysis

3. Supplemental table 1: list genetic backgrounds of strains. Arrange the strain names in a systematic way. The gfp-tagged cosEb is not listed, and it is in a TIGR4 background. It is not clear how each strain is constructed. The strains 'adopted' (line 488) from a previous study should be specified with a reference, and any new constructed strains for this study should be specified as such. Are the adopted strains the same as reported previously, or are they single colony isolated or remade for this study? A good format to use is appendix Table 1 of Liu et al. For example, the genotype for pbp2x Δ pde1 should be XL25 Δ pde1::camR. Some of the strains (e.g. fabH vraS) are not mentioned in the text and should be omitted from the table.

4. In Liu et al., the pbp2xi strain in the presence of IPTG (Fig.3G) show decrease in OD after 3h and no uptick in growth in 6 h (no data after 6h). However, in Fig. 5C, 6A, and 6B, there is an uptick of growth after 6h. More concerning is the graph in Fig. 7B, the same strain grew much better than shown in 5C, 6AB. Is there a suppressor mutation in this strain? Liu et al mentioned that suppressor mutations are sometimes found in the dcas9 by sequencing. Because of this concern, growth curves should be shown for each of the 13 parent strains instead of a growth value. It will be best if the parent strains are single-colony-isolated and whole-genome-sequenced before the highly involved Tn-seq experiment.

5. In figure 2B, what is the unit for growth in the y-axis? The values are between 0.1 to 0.5. Is this the maximum OD? This should be clearly written in Materials and Methods, and in the figure legend.

6. The author should refer to supplementary Table 1 instead of supplementary Table 9 for the strains in line 259. Supplemental Table 9 has a list of primers, but the use of these primers are not stated. The use of each primer in supplemental Table 9 should be briefly described in Materials and Methods. It appears that some of the primers are not relevant to this paper, because these genes are not mentioned in the text. They should be deleted in the table. Primers used for qPCR studies should be included in this table.

Questions and comments related to analysis of the CRISPRi-TnSeq data set

7. The raw data is presented in Supplemental Table 3, which should have an explanation of

12the heading in the experimental section. There is a tab named as 'description', which readers may easily miss. Each of the items under description should be clearly explained in Materials and Methods. Explain the headings for each column in supplemental Table 3 to guide the readers through the analysis.

8. Explain in line 535 to 538 "To make genetic interaction values comparable between different datasets produced for individual CRISPRi targets, individual fitness difference (Δw) and genetic interaction datasets were z-score normalized. All significant genetic interactions with $p\text{-adj} \leq 0.05$ were mapped in a genetic interaction network using Cytoscape". Explain z-score normalized.

9. In supplemental Table 3, there were entries with $p\text{-adj} \leq 0.05$, but was considered as false under significance. How are the true and false values assigned? In line 156, it is mentioned that 1334 significant genetic interactions are identified. Are these the ones that have true significance in Supplemental Table 3?

10. In Supplemental Table 4, arrange all data in a sheet that is easily understood. A footnote also would be helpful.

Questions and comments related to growth curves and Δw of constructed strains

11. The growth profiles of the parent CRISPRi strains used for Tn-seq is very important. As mentioned above, the growth curves of these strains should be shown, similar to those shown in Fig. 3 of Liu et al. instead of the bar graph of a selected strains in Fig. 2B. Explain the unit in Fig. 4c.

12. Line 140, it would be best if the parCi strain is whole-genome sequenced, or sequenced for the *dcas9* region to rule out suppressor mutations.

13. Line 156: >1300 genetic interactions were identified in the screen. However, only 27 were validated. How were the 27 selected? Is it based on the highest or lowest score? Are there any pairs that do not fit in the graph and data not shown? From Supplemental Table 1, there are more than 27 double strains. Are all the tested ones shown on the figure? In addition, the double *cosEa* strains shown in Fig. 8 and 9 are not included in the plot.

14. Line 259-262: Explain how growth rates and fitness values are obtained for each constructed double mutant in Materials and Methods. For supplementary table 11: use a footnote to explain how the Δw (CRISPRi-TnSeq) and Δw (mutant growth) are obtained. How many times was Δw determined?

15. Supplemental Table 10 showed raw data of time vs. OD of growth curves. A simple table of growth rate, growth yield and calculated fitness values of double vs single mutants should be listed.

Essentiality of *rodZ*

16. Line 389: As reported recently by Lamanna et al., 2022 in Molecular Microbiology, *RodZ* is essential in unencapsulated and encapsulated *S. pneumoniae* D39. Constructed *rodZ* mutants typically contain suppressor mutations. Therefore, the *rodZ* knockout reported here likely contains suppressor mutations. Please comment or explain.

Other comments:

13

17. Line 135, Supplementary Table S11, how is growth measured? What is measured in Fig. 2B? How is ΔW (mutant growth) determined in Supplementary Table S11? Primers used for qPCR should be in a supplementary table.
18. Line 171, refer to Fig. 2C.
19. Line 189: put boxes over parC and fabH interaction in supplementary Figure 4 to highlight the information in the text (line 189).
20. Line 215-229: should have boxes or arrows in Figure 3 to show the points shown in the text line 215-229. It is not clear that there are overlapping targets from this figure. Explain nodes and attributes in Materials and Methods.
21. Line 216: Supplemental Table 2 is qPCR results of expression, did not show hierarchical clustering.
22. Lines 230-256: 27 interactions were shown in Fig. 4C. It is not clear which 7 interactions were validated in line 233. Supplemental Figure 6 showed only fabH and clpI, and pbp2x and clpC. As mentioned above, all 27 growth curves should be shown.
23. Section starting from line 257, all 27 growth curves should be shown.
24. Line 259: supplemental table 9 are primers, not relevant to the sentence.
25. Line 266-267: put in references for fabH and plsX (line 266-267).
26. Line 292-302: murM, murN, and pbp2x are not the pairs for validation (not found in Supplemental Table 11). pgdA and pbp2x are also not one of the 27 pairs tested. Also pbp1a, pbp1b, or pbp2a are not tested. These pairs should be tested.
27. Line 305: pbp3 also is not tested with pbp2x. Likewise of pbp2x with sepF or pvaA.
28. Line 322: Supplemental Figure 7 and 8 did not show increased c-di-AMP level in the growth medium. c-di-AMP level was not measured in the growth media.
29. Line 342: K concentration is not stated.
30. Line 343: Fig. 6A shows cell growth, not cellular turgor.
31. Line 343-345: Figure 6 and 7, the light blue and light green lines are very confusing. Use different colors. Why does pbp2xi +IPTG show a growth at a later stage in Fig. 6A and 6B right panels? Please clarify.
32. Line 364: Figure 7 :pbp2xi +IPTG grew very well compared to Fig. 5B, Fig. 6A, and Fig. 6B (right panel). This figure casts doubt on pbp2x cozeB negative interaction.
33. Line 557: should specify that sp1505 is cozeB. It also should have been listed in the

strain table that the genetic background for this construct is TIGR4. This information should also be in the main text.

34. Line 571 When are the cells collected for the c-di-AMP assay? After how long in the presence or absence of IPTG? It is mentioned that the cells were grown up to OD of 0.5 to 1.0. However, the growth curve in Fig. 5B, cell did not reach OD of 0.5. Please clarify.

35. Line 361: *cozEa* and *cozEb* interactions were not confirmed by growth of constructed strains.

36. Supplementary Figure 1: the colors are very faint. Best if a section of the data is shown clearly. Show clearly some of the examples shown in other figures. On the X-axis, list the gene knock down. Examples of the following should be shown: 1(line 181) that *atpF* interaction are mostly positive; 2)(line 188) *parC* and *fabH* knockdown are enriched in DNA repair and lipid metabolism genes; and (3) (line 191) *adk*

Author Rebuttal to Initial comments

Reviewer 1. - Comments:

Reviewer 1 - Comment 1. - L46: The authors should define a genetic interaction in the Introduction, even if briefly. In particular, that the measured phenotype of the double mutant (in this case a mutant and a gene knockdown) deviates from the expected phenotype based on the single mutants. Furthermore, it would be nice to distinguish “synthetic lethal” and negative genetic interactions, as the sometimes intermixed use of these terms in the paper can lead to confusion. Classically, synthetic lethals are cases where the combined loss of two non-essential genes leads to an inviable phenotype. In this work, one of the genes under study is always an essential gene, so technically no synthetic lethal gene pairs can be identified. This is a little nit-picky, but I think these distinctions are important to make.

Author response:

While genetic interactions are defined in the Materials section, we have now also added the definition to the introduction of the manuscript on **P3L61-66**. The reviewer is right that if we adhere to the strict definition of a synthetic lethal interaction, this study, with its focus on mapping genetic interactions between essential and non-essential genes, indeed does not identify synthetic lethals per

15se. As suggested, we have substituted 'synthetic lethal' by 'negative genetic' interaction throughout the manuscript.

Reviewer 1 - Comment 2. - L50-52: Rephrase, reads as if CRISPRi is a TIS tool. Maybe group TIS and CRISPRi as bacterial functional genomics tools.

Author response:

As suggested, we have changed TIS for "genome-wide functional genomics".

Reviewer 1 - Comment 3. - L75: Sounds like all of the approaches mentioned above are array-based, but the human example (ref 9) is a pooled-based assay with next-generation sequencing.

Author response:

We have adjusted this section and clarified that not all are array-based methods.

Reviewer 1 - Comment 4. - L117 and in Figure 1: Recommend changing the subpanels to letters to match the rest of the manuscript.

Author response:

We changed the sub-panel numbers to letters.

Reviewer 1 - Comment 5. - L125-8: How is the expected phenotype of the transposon mutant-knockdown strain phenotype? This is important because it the measured deviation from this values that defines a genetic interaction. In particular, what is the measured fitness of the knockdown strain alone at the different levels of induction? Or do the authors just assume that this value is uniform

across the entire mutant library (because all strains have the same knockdown cassette), so that any deviation in fitness of the transposon mutants in the absence and presence of induction constitutes a genetic interaction (i.e. what's shown in Figure 1.5)? If this is the case, then it would be good to explicitly mention this here and/or in the Methods section.

Author response:

CRISPRi knockdown of an essential gene reduces the growth of the entire mutant library, the relative fitness of individual insertion mutants remains similar in the presence of IPTG unless the corresponding essential and nonessential gene pair genetically interact. Thus, a deviation in fitness of a transposon mutant in the absence and presence of essential gene knockdown constitutes a genetic interaction. The manner in which genetic interactions are scored is described in detail in the results section “**CRISPRi-TnSeq identifies nonessential genes mechanistically linked to essential genes**”. To make the point even more explicit we further highlight this on **P6L135-139** and again in the methods sections on **P21L604-P22L609**.

Reviewer 1 - Comment 6. - L130: Is it 12 essential genes? Because *clpP* is non-essential?

Author response:

Indeed, of the 13 targets, 12 are essential genes, and while *clpP* is essential in many species and even certain *S. pneumoniae* strains, it is not essential in the strain we used here. A *clpP* KO may, depending on the strain, be very difficult to obtain. Importantly, in this manuscript *clpP* knockdown (KD) does not trigger a defect. However, due to the combination of a KD with a KO, we were able to score interactions, even by knocking down a gene (*clpP*) that in this particular strain is non-essential, which illustrates a strength of our approach and the value of genetic interactions in general. We have clarified in the manuscript that *clpP* is non-essential (**P6L141/149**). Furthermore, we discuss on **P18L505-508** that combining KDs with KOs can also be used in scoring genetic interactions between non-essential genes, for which *clpP* serves as an example.

Reviewer 1 - Comment 7. - L136-8: It appears that the uninduced libraries have varying fitness defects, suggesting some leakiness to the IPTG-inducible system (growth values in panel B range from ~ 0.25 for *parC* to ~ 0.45 for *atpF*). What is the likely reason for these different fitness values, and is there any impact on the identification of genetic interactions?

Author response:

The CRISPRi strains are adopted from our previous study where we introduced CRISPRi for *Streptococcus pneumoniae* (Liu et al. 2017). Using *luc* reporter assays, we clearly demonstrated that there is no observable leakiness in our CRISPRi system (Liu et al. 2017 Fig. EV1C). In the present manuscript, the values in Fig. 2B are OD 600nm values, not fitness. To avoid confusion, we have substituted the labels. The presented values should only be compared within each figure, the variation between figures represents normal variation in growth. Importantly, growth curves of individual libraries show that there is no fitness effect of the presence of the uninduced CRISPRi cassette.

Reviewer 1 - Comment 8. - L140: units should be μM .

Author response:

Adjusted.

Reviewer 1 - Comment 9. - Figure 2: Panel B is hard to see due its size and resolution, especially the y-axis labels. As far as I can tell panel C is not referred to in the manuscript. In panel C, where are the gene functional categories derived from? The meaning of the inset figure in panel G is unclear. In panel H, a legend showing the scale of the fitness values is needed.

Author response:

Fig. 2B is substituted with a high-resolution image. Fig. 2C (revised Fig. 2D) is added on **P8L183**. Functional categories / biological processes are adopted from *Streptococcus pneumoniae* the NCBI Reference Sequence file NC_008533.2, which is indicated in the Methods section. Fig. 2G inset (revised Fig. 2E) is added on **P8L192**. As suggested panels E and H are moved to Supplemental Figure 4 and 6, and a fitness scale is added to Supplemental Fig. 6.

Reviewer 1 - Comment 10. - L157: I didn't see a succinct table that presents just the list of 1,334 gene pairs with genetic interactions, which would be useful to readers. Also, it would be nice to mention that some query genes had few interactions (*atpF*), while others had many (*ftsZ*).

Author response:

This information is summarized in Supplemental Table 4, which is referred to on **P7L169**.

Reviewer 1 - Comment 11. - If I understand Supplementary Table S3, then ~half or more of the non-essential genome has a negative genetic interaction with *pbp2x* and *ftsZ* if you add enough IPTG to the system. For instance, at 25 uM IPTG, *pbp2x* has a negative genetic interaction with 1,266 different non-essential genes! This seems odd to me, because this must include many genes that only have very indirect relationship to cell wall synthesis. At a lower induction level, there's considerably less genes (44) with a negative interaction with *pbp2x*. Obviously, filtering for informative interactions from >1,000 genes is not ideal, so it's clear that the induction level is critical for identifying a manageable number of interactions in the authors CRISPRi-TnSeq approach. I'm curious if the insights that the authors draw from *pbp2x* are from the lower induction level (or example, the network presented in Figure 5A), or if they are selectively including genes in these networks based on the narrative they want to tell. Did the higher level of induction cause severe growth inhibition for these two genes? Unfortunately, these data are not included in Figure 2B.

Author response:

Our data indeed shows that a situation can be created where IPTG is increased to such an extent, that the resulting knockdown of the essential gene becomes so large that any other disruption to the system (e.g., a non-essential gene KO) can tip the balance and cause a disruption that results in a very strong fitness defect, indicative of a genetic interaction. However, this defect is not the result of a genetic interaction but merely due to the organism being on the ‘verge of collapse’ (e.g., see our work on entropy *Zhu et al., 2021. Nature Comm*, <https://doi.org/10.1038/s41467-020-18134-z>). This is exactly what happens to *pbp2x* and *ftsZ* in the presence of the highest IPTG concentration, resulting in a very high number of interactions many of them being false positives. This highlights that it is critical to screen at different concentrations; a low concentration to identify the strongest interactions, and a higher concentration to increase the possible number of interactions that can be confidently scored. While the highest *pbp2x* and *ftsZ* IPTG concentrations (25 and 20 μM respectively) included the interactions scored for the 15 μM concentrations, the highest concentration could not be used to expand the scored number of interactions, due to the systems instability issues discussed above. Importantly, it really depends on the essential gene and likely the corresponding sgRNA what ends up to be a concentration that is too high, as for some essential genes we were able to use up to 100 μM . We agree that these considerations are important, and we have included these in the Methods section on **P22L622**. Additionally, we have also indicated this on **P8L184**.

Reviewer 1 - Comment 12. - L172-4: Figure 2E and Supplemental Figure 3 are low resolution. I also recommend removing panel E from Figure 2 to make additional space for the other panels.

Author response:

We removed panel 2E from Figure 2. Additionally, Supplemental Figure 3 (revised Supplemental Figure 4) is substituted with a high-resolution version.

Reviewer 1 - Comment 13. - L175: Figure 2F does not show the enrichment of genetic interactions around SPD2014-SPD2064.

Author response:

The label of Fig. 2F is changed; “Highest” now also indicated “SPD2014-2064”, and lowest refers to “SPD0229”.

Reviewer 1 - Comment 14. - L188-9: How enriched are these gene functional categories in the *parC* and *fabH* datasets? And with what statistical significance?

Author response:

Normalized enrichment scores of DNA repair and Lipid metabolism categories are 1.98 with FDR 0.01 and 1.93 with FDR 0.0004 for *parC* and *fabH* knockdown, respectively (see Supplemental Table 6).

Reviewer 1 - Comment 15. - Figure 3: Explicitly state in the legend that the clustering of experiments in panel B is based on the entire datasets, and that only select genes are being shown in the heatmap. Also, a legend showing the scale for the fitness values in B should be included.

Author response:

We now mention in the legend that clustering of experiments is based on the entire dataset. The fully clustered dataset can be found in Supplemental Figure 7. Additionally, a fitness scale is added to panel B of Figure 3.

Reviewer 1 - Comment 16. - Figure 4: The networks in panels A and B are low resolution. In particular, the gene labels in the panel B network nodes are nearly impossible to read. In panel C, what pairs of genes were tested for the 27 validations, and was the same IPTG concentration used in the validation and in the genome-wide data?

Author response:

Figure 4 low-resolution images are substituted with high resolution ones. Tested gene pairs of Fig. 4 panel C are listed in Supplemental Table 11, which is indicated in the legend. To ensure robustness of the scored interactions, genetic interactions were tested in the validation experiments, both below and above the IPTG concentration(s) used in genome-wide experiments. This is indicated in the legend.

Reviewer 1 - Comment 17. - L233: 7 interactions validated or 27 validated? The reference to Figure 4C seems to signify the latter.

Author response:

We validated 32 interactions and 7 of those are interactions with pleotropic genes. We edited the sentence on **P9L246** to clarify this.

Reviewer 1 - Comment 18. - L234-9: Do these 7 non-essential genes with frequent negative genetic interactions with the essential query genes have negative fitness values in the C+Y growth medium used for the genome-wide assays? Is the interpretation that mutants in these 7 genes are already sick, and so an expression knockdown of an essential gene just exasperates the fitness defect.

Author response:

These 7 genes are nonessential in C+Y medium, and do not have a fitness defect, which is mentioned on **P10L248** and can be found in Supplemental Table 3. As described in the manuscript this suggests that these genes are important for phenotypic robustness of the organism.

Reviewer 1 - Comment 19. - L239-47: Similar to above, what are the phenotypes of these 10 genes on C+Y (are they detrimental, i.e. transposon mutants have a fitness advantage in the pooled assay)?

Author response:

These genes are not detrimental. Tn-mutants of these genes only have a small fitness defect in C+Y media. These data can be found in Supplemental Table 3.

Reviewer 1 - Comment 20. - L250: Check spelling of pleiotropic.

Author response:

Typos are corrected.

Reviewer 1 - Comment 21. - Figure 5: Gene labels in the network in panel A are hard to read. Panels B and C would be easier to follow if the colors matched by strain. Why was a different IPTG concentration used in B and C?

Author response:

We changed the color of Figure 5 panel A labels to make them more visible. Consensus is established between panel B and C in terms of colors. 20 μ M IPTG was used in both panel B and C which is indicated in the legend.

Reviewer 1 - Comment 22. - L311: This implies that node *pvaA* in Figure 5A should be colored blue.

Author response:

pvaA nodes are changed to blue.

Reviewer 1 - Comment 23. - L353-4: Specify the overall overlap of genetic interactions between *cozEa* and *pbp2x*. Are these genes the most connected in the network?

23Author response:

cozEa shares all the interactions with *pbp2x* and are the most connected in the network. This overlap is now mentioned on **P13L369**.

Reviewer 1 - Comment 24. - L372: overtime should be two words.

Author response:

Correction is made.

Reviewer 1 - Comment 25. - Figure7: Why isn't the growth defect of *pbp2xi*+IPTG more severe (in panel B)? Was a lower concentration of IPTG used to highlight the more prominent fitness defect in a Δ -*cozEb* background? More broadly, the authors should indicate the concentration of IPTG used in each assay/figure panel, because it's possible that different concentrations were used to highlight the phenotypes (the growth curves and the cell images). In panels CDE, what is the expected impact of adding daptomycin? I'm not sure what I'm supposed to distinguish between the +DAP and -DAP images.

Author response:

A lower concentration (12.5 μ M) of IPTG was used for the growth curve of the Figure 7 panel to highlight the sensitivity of the interaction. We have checked and made sure to include IPTG concentrations in the legend and/or figures (where possible) throughout the manuscript to avoid any unclarity. Since a *cozEb* mutant is hypersusceptible to daptomycin, we hypothesized that daptomycin may have an impact on CozEb's positioning and activity in the cell membrane. However, we did not observe any differences between the presence and absence of daptomycin. We have added this on **P14L393**.

Reviewer 1 - Comment 26. - Figure 8: Indicate the IPTG concentration used, in particular in panels A and B. The *cozEai*+IPTG strain is only moderately sick, so I assume that a low IPTG concentration was used. In panel G, what do the *lgt* and *lspA* heatmap values mean?

Author response:

In all the growth and microscopy experiments related to Figure 8 (revised Fig. 7) 100 μ M IPTG was used to knockdown *cozEa*. The IPTG concentration is mentioned in the figure legend. The heatmap values of panel G represent fitness reduction of *lgt* or *lspA* mutant due to the knockdown of *cozEa* measured in the pooled assay (revised Supplemental Fig. 16A). A scale has been added to 8G (revised Supplemental Fig. 16A) and details are indicated in the legend of the figure.

Reviewer 1 - Comment 27. - L391: There's no rodZ data in Figure 8B.

Author response:

Correction is made on P15L415.

Reviewer 1 - Comment 28. - L399:400: The interaction between *cozEa* and *ctpE* is not that strong.

Author response:

Based on the fitness values, the *cozEa-ctpE* interaction may not be the strongest but it is significant. Importantly, the validation growth experiments, and the microscopy experiments indicate that the interaction has biological significance, i.e., the absence of CtpE rescues the growth of cells experiencing *cozEa* knockdown, and microscopy images show that it reduces/prevents cell blebbing. Moreover, the interaction highlights the sensitivity of CRISPRi-TnSeq and the importance of suppressor interactions.

Reviewer 1 - Comment 29. - L470-2: Another CRISPRi-TnSeq variation is to move the different CRISPRi query gene constructs into an existing transposon mutant library. The advantage here is that one wouldn't need to construct a new Tn-Seq library for each query gene under investigation.

Author response:

We considered doing this but it is challenging to ensure incorporation of a CRISPRi cassette into all transposon mutants in a pool with equal efficiency. The result is that a large number of transposon mutants will drop out from the pool, likely resulting in a skewed distribution. Therefore, generating a transposon mutant library in a CRISPRi strain background will result in a more complex pool. Additionally, it is often not that difficult to generate transposon pools. However, while we agree that an easy way to generate a pool that could sample all genetic interactions simultaneously would be great, there are serious complications that will make this difficult. We note this in the Methods section on **P20L568**.

Reviewer 1 - Comment 30. - Discussion: One limitation of the current study is that only a single condition (C+Y media) was profiled. For many putative genetic interactions (for example, for metabolic genes or transporters), the GI will only be apparent under certain conditions. This could be touched upon briefly in the Discussion.

Author response:

We have included this in the discussion on **P18L497**.

Reviewer 1 - Comment 31. - L503-4: It appears that biological replicates were only performed in a few instances. Why? In the cases where replicates were performed, what the agreement between the assays?

Author response:

All CRISPRi-TnSeq experiments were performed using 6 independent transposon mutant libraries, and each library was tested with at least 2 IPTG concentrations. Each experiment thus consists of multiple technical and biological replicates. This is described in the Methods section on **P20L563**.

Reviewer 1 - Comment 32. - L518-24: Are the fitness values for different insertions in the same gene averaged?

Author response:

Yes, fitness values for different insertions in the same gene are averaged, which is in concurrence with the original Tn-Seq approach, and which is described on **P21L593**.

Reviewer 1 - Comment 33. - L526-28: How is the multiplicative fitness value calculated ($W_a \times W_b$)? More specifically, if W_b is the fitness of the CRISPRi gene knockdown, where does this value come from?

Author response:

As mentioned above in the response to comment 5, since CRISPRi knockdown of an essential gene target reduces the growth of the entire mutant library, the relative fitness of individual insertion mutants remains similar in the presence of IPTG unless the corresponding essential and nonessential gene pair genetically interact. Thus, a significant deviation in fitness of the transposon mutants in the absence and presence of essential gene knockdown constitutes a genetic interaction. This is described in detail on **P6L136** and in the Methods section.

Reviewer 1 - Comment 34. - L526:43: The code used to calculate delta-W and its associated statistics should be made publicly available.

Author response:

The approach has been described in detail in many different publications that we reference, however we have now also included the fitness equation in the Methods section on **P21L585** and relevant references of Δw calculation on **P22L609**.

Reviewer 2. Major comments:

Reviewer 2 - Comment 1. -Overall, this is a compelling paper that presents a new frontier in mapping bacterial genetic networks. By combining the ability to dynamically repress essential genes (CRISPRi) with TnSeq's ability to quantify nonessential gene fitness data in a massively multiplexed system, the strategy combines the strengths of the two approaches.

Author response:

We appreciate the reviewer's comments.

Reviewer 2 - Comment 2. -One important control seems to be missing. How do the authors know that there are not off-target effects of their CRISPRi system? If the essential genes being targeted were even slightly confounded by nonspecific targeting of other chromosomal regions by dCas9, that would seriously compromise the networks that emerge. If it were a fixed nonspecific effect, those genes affected could end up looking like pleiotropic contributors to fitness because they would show up (erroneously) as being linked to multiple knocked down essential genes. The control to ensure this isn't the case would be to perform CRISPRi-TnSeq with one (or several) non-targeting/scrambled guide RNA templates. When the system is used with this design, there should be no significant difference from Tn-seq performed with WT.

Author response:

The original CRISPRi system in *Streptococcus pneumoniae* that we build our current approach on was described and extensively validated in our previous study Liu et al, Mol Syst Biol 5:931, 2017. The CRISPRi strains that we used in the current CRISPRi-TnSeq manuscript are directly derived from that published work (Liu et al, 2017). All of this is described in this manuscript and specifically on **P6L143**. In the above-mentioned previous study, among other approaches we also performed high-depth RNAseq to characterize CRISPRi strains with and without sgRNA to a *luc* reporter gene in the presence or absence of 1 mM IPTG. The RNAseq data showed that LacI repressed *dcas9* strongly and 1 mM IPTG increased *dcas9* expression by ~600-fold. As expected, the presence of IPTG reduced *luc* expression (~84 fold) in the sgRNA-*luc* strain (Liu et al, 2017). By comparing expression profiles of with and without sgRNA-*luc* we found that the CRISPRi system repressed *luc* only in the presence of sgRNA-*luc*, confirming specificity and sensitivity. Furthermore, the CRISPRi-TnSeq data in this manuscript shows that interactions of pleiotropic genes are specific. In this study genes having interactions with more than half of the targets are considered pleiotropic. Of these pleiotropic genes we found that they did not have interactions with all of our CRISPRi target genes, and these interactions were not the same between pleiotropic genes. Moreover, interactions between pleiotropic genes and their corresponding interactors are mixed negative and positive interactions, which would be counterintuitive if they were non-specific interactions. Furthermore, some genes are also identified as pleiotropic in other species such as *clpP* in yeast. Lastly, if we had a systematic issue with off-target effects in our system our read-out would be littered with genes that had fitness calls with very high variance, which would result in a very low hit rate (i.e., significant fitness differences) and a very low validation success rate. Identified hits also differ depending on which gene was targeted by CRISPRi, demonstrating no significant systematic off target effects.

Reviewer 2 - Comment 3. -The writing clear and easy to follow. However, the paper is very long and tries to capture a possibly overzealous number of topics. Many of my comments below are aimed at tightening the core story, which is a good one, by discarding and consolidating some of the more extraneous strands.

Author response:

We appreciate this comment, and we appreciate the advice. We recognize the manuscript is long and contains a lot of information. However, the data is rich in content, and we feel it is extremely

29important to present a manuscript that is extensively validated and not only contributes technologically, but also highlights the biology that can be uncovered with this new advance. Importantly, we have focused on tightening up the language, some of the stories as well as the figures, which we hope has helped.

Reviewer 2 - Comment 4. -The validation sections overuse speculative language. A good example of this is lines 182-186 (“Reducing ATP MAY thus also buffer against more general stress...atpF knockdown MIGHT increase intracellular pH...which MAY additionally contribute to buffer against stress”). Some of this can be moved to Discussion or removed altogether. The data can be presented as validating the technique without positing mechanisms for which there is not proof.

Author response:

We removed additional explanations when they are not critical to the core message.

Reviewer 2 - Comment 5. -Figure 1H seems out of place and the accompanying discussion doesn't really lead anywhere in the rest of the paper. I would remove this.

Author response:

We believe the reviewer means Figure 2H, which we removed from Fig. 2, as it does not illustrate a key point clearly. Actually, CRISPRi-TnSeq identified genes of both purine biosynthesis and degradation pathways, which are presented in Supplemental Figure 5. This complete version of the figure will help to understand how elegantly CRISPRi-TnSeq unveils two opposite pathways that are interconnected through substrate availability.

Reviewer 2 - Comment 6. -Figure 2 requires extensive revision. Panel 2B is not well explained and doesn't—on the surface—make a lot of sense. The clustering does not look convincing. For instance, why does atpF15 cluster with atpF30 when the columns below look so divergent? How was this

done? What scale is in the heatmap? What genes (or gene sets?) are on the vertical and horizontal axes? Currently very confusing.

Author response:

It seems the reviewer means Figure 3B. It is expected that genes identified at lower IPTG concentration (e.g., 15 μ M) will also be identified at higher concentration (e.g., 30 μ M) during target knockdown. Therefore these 2 gene sets have high overlap and cluster together. In panel B, a selected portion of the heat map is highlighted to show gene clusters or potential protein complexes. We added the complete cluster analysis to Supplemental Figure 7, which should clarify that clustering is based on the entire dataset, and not just a fraction. To increase clarity, we also mention this in the figure legend of the revised manuscript. All nonessential genes and conditions are on the vertical and horizontal axis respectively. We also added a scale to the heatmap.

Reviewer 2 - Comment 7. -Figures 7 and 8 (and accompanying discussion of CozEa-related results): In my opinion, the strongest data in these two figures are the gene network map combined with the growth curves. Much of the rest of it, including the microscopy, is distracting, hard-to-interpret, and peripheral to CRISPRi-TnSeq (the photomicrographs in Figure 5 are more convincing and more directly related to the main thrust of the paper). Much of the related verbiage in the results section leans in a speculative direction. I would COMBINE and SIMPLIFY Figures 7 and 8, making a single figure that shows a SIMPLIFIED network with the genes whose interactions you're focusing on (leave out the many shown that you're not) and the relevant growth curves. The Figure 7 and 8 microscopy could, I feel, be moved to Supplemental.

Author response:

We have combined and simplified Figures 7 and 8, by limiting the microscopy images that are shown, which have been moved to Supplemental Figures 15, 16. We do want to keep several of the microscopy images because it nicely illustrates that effects on growth can be paired with an array of morphological effects that can be triggered or resolved by genetic interactions. Importantly, while we can identify many genetic interactions by measuring effects on growth in pooled assays (e.g., with

CRISPRi-TnSeq) it only tells a part of the story, and that story can be expanded upon by monitoring morphological changes. Moreover, such changes will give you additional information on gene function and interaction origin. We also retained the network in Figure 7A as we find it important to highlight interactions beyond the ones we validate, as this highlights the inherent complexity that comes with genetic interaction mapping. However, if the editor finds that we should make further cuts on figures and reduce complexity, we will do so.

Reviewer 2 - Comment 8. -Figures throughout: Using white circles with thin outlines for the essential genes in your gene networks strikes me as a mistake. They're very hard to see (actually invisible in Figure 1A). Would choose a different color and possibly a different shape for these important network members.

Author response:

Essential gene nodes are reshaped into rectangles with a thicker outline.

Reviewer 2 - Comment 9. -What is the difference between a “genetic capacitor” and a stress response? Aren't they both systems that interact with a lot of genes to help the cell adapt to an external or internal disruption? Do genes involved in canonical stress responses (e.g. the stringent response) show up as mediators of essential gene manipulation? If not, isn't that surprising?

Author response:

A stress response is defined as a set of genes that display a transcriptional change in response to a specific perturbation. A capacitor is a gene that mediates or protects against a wide array of perturbations, and here, as others have done, we specifically define it as a gene that protects against a variety of (genetic) perturbations by buffering the phenotypic effects. While capacitors may to some extent be connected to stress response genes they are not the same thing. Importantly, genes that respond transcriptionally to a perturbation (stress response genes) are rarely the same genes that are involved in the phenotypic response, albeit they are closely connected (Jensen et al., 2017). Furthermore, the stringent response as mentioned by the reviewer is often triggered in response to

nutrient(s) deprivation. All the experiments were performed in rich C+Y media and cells were grown for 3-4 hours which shouldn't trigger the stringent response. Additionally, it's unlikely/there is no evidence that any of the knockdowns would sufficiently trigger this response either.

Reviewer 2 - Comment 10. -The technical details of performing CRISPRi-TnSeq in pneumococcus are reasonably well laid out, since both CRISPRi and TnSeq in this organism have been previously described in great detail. The authors' claim that the system would be easily transferred to any other bacterium is dubious, however. Neither TnSeq nor CRISPRi are—at this point—universally available, and the steps needed to institute them in non-model bacteria are nontrivial. It seems reasonable to say that the system should be adaptable to other organisms, but only in cases where the two halves of the approach are established. To this end, I would remove the last sentence of the abstract as currently written.

Author response:

We have made these changes by adjusting the last sentence from the abstract and the Discussion section.

Reviewer 2 - Comment 11. -The Methods subsections outlining statistical analysis of the data (under the headings “Analysis of CRISPRi-transposon mutant libraries” and “Genetic interaction mapping”) are too vague. Specifically, it's not clear enough how a genetic interaction is determined to be significant.

Author response:

Genetic interactions are defined in the results section and in the Methods section it is further outlined in detail how fitness is calculated and how a significant interaction is scored.

Reviewer 2. Minor comments:

1. Throughout the paper (e.g. lines 169, 171) the authors use mM whereas in Figures it's uM.
2. Line 372: “Over time” not “overtime.”

Author response:

Changes are made in the revised manuscript.

Reviewer 3. Major comments:

Reviewer 3 - Comment 1. This manuscript by Jana et al. describes the development of an approach called CRISPRi-TnSeq that attempts to determine genetic interactions between an essential gene and all non-essential genes in *S. pneumoniae*. Their system is a simple composition of a single-gene CRISPR-interference followed by whole-genome transposon mutagenesis and sequencing. IPTG-induction of dCas9 enables control over the context and degree of essential gene knock-down. Comparison of gene transposon insertions in IPTG induced vs non-induced populations enables them to quantify positive/negative genetic interactions between the essential gene and all other non-essential genes in the genome. In motivating the utility of their approach, the authors imply that existing systems cannot easily identify essential x non-essential genetic interactions in a high-throughput manner. However, this is not true. In fact, Jiang et al. (*Cell*, 180(5):1002 (2020)) and recently Ellis et al. (<https://elifesciences.org/reviewed-preprints/86903>) have shown that libraries of double CRISPRi knockdowns can be generated in bacteria and enable study of pairwise and even multiplex genetic interactions, including essential x non-essential and in particular, essential x essential genes which is beyond the capacity of the authors' CRISPRi-TnSeq approach. This is highly relevant work that the authors don't seem to be aware of.

Author response:

The first paper mentioned by the reviewer (*Cell*, 180(5):1002 (2020)) uses a CRISPR adaptation system (CALM) that randomly produces crRNA libraries using transformed genomic DNA. While a very cool approach it is hard to directly compare it to CRISPRi-TnSeq as it serves a different purpose/can potentially answer different questions, while also having a mostly unclear sensitivity,

34and which likely works on a different (less sensitive) scale. First, due to the random nature of crRNAs, adapted crRNA mediated CRISPRi knockdown systems are not as efficient as synthetic CRISPRi systems which use a sgRNA to knockdown a target gene. While, in principle, it is possible to knockdown all genes in a genome by using an adapted crRNA library, due to its random nature and low efficiency, it requires 100-1000s of crRNAs to knockdown a gene. A typical bacterial genome contains several thousands of genes, which could require in excess of a million crRNAs to cover the entire genome. Targeting two genes simultaneously to score genetic interactions could require more than a billion crRNAs, and bring you beyond what can be realistically generated, tracked and measured. Second, a possibly even bigger challenge is measuring and calculating the precise fitness effect of individual crRNAs. It is critical to understand what the exact fitness effect of a knockdown is, and that this effect is consistent. Because different crRNAs targeting the same gene will have different effects on gene knockdown and therefore trigger different effects on fitness, an accurate measure of the exact effect on fitness is very hard to achieve. This challenge therefore gets compounded in digenic interaction studies, which depend on accurate measurements of single knockdowns, which are used to calculate the expected double knockdown fitness (the multiplicative), and accurate measurements of the double knockdown. Third, due to the much higher number of targets (crRNAs) to track a random crRNA library needs up to several orders of magnitude deeper sequencing to measure the abundance of crRNAs in the library, which can rapidly drive-up costs but also exceed sequencing capabilities. Lastly, it remains unknown whether the CALM system can be as broadly adapted to other bacteria, as CRISPRi and Tn-Seq have.

The second study that the reviewer mentioned (Ellis et al. (<https://elifesciences.org/reviewed-preprints/86903>)) also uses adapted crRNAs to study genetic interactions of a set of 44 effector genes. Much of the same issues that we described above apply to this. Moreover, this publication further highlights that the gene knockdown efficiency of the system is highly random and variable, where most crRNAs show either nothing or up to ~2-fold knockdown. These variables and the lack of phenotypic effects due to weak and variable knockdown make accurate single and double gene knockdown fitness measurements very difficult.

In the revised manuscript we address these two manuscripts and the developed tools in the discussion section on **P18L521** and contrast them to CRISPRi-TnSeq. Also see next comment.

Reviewer 3 - Comment 2. As such, the technological novelty of the work is not that significant. However, the authors apply their approach to a collection of essential genes (enriched for antibiotic

35targets) to discover suppressive (positive) and aggravative (negative) genetic interactions. Out of a total of 24,000 potential interactions, they find 1,334 that are significant, with 754 negative and 580 positive. The authors then validate a set of these interactions and with the help of previous work/knowledge of the genes and further genetic experiments/imaging propose potential hypotheses for the mechanism by which these interactions arise. The areas of focus include lipid biosynthesis, connection between septal and peripheral peptidoglycan synthesis, the role of cyclic-di-AMP on intracellular K⁺ and turgor, connections between peptidoglycan synthesis and cell morphology/envelope integrity, and connections between chromosome decantation and segregation. These are interesting vignettes that demonstrate the power of performing systematic genetic interaction studies. As a collective, they add incrementally to the long literature on the topic extensively explored in previous work. As individual stories, they would be of interest to S. pneumonia cell biologists as brief forays that require further experimental follow-up.

Author response:

We respectfully and adamantly disagree with the technological novelty comment. As described in the previous comment existing technology has serious disadvantages to accurately and quantitatively score genetic interactions. Moreover, even using a designed CRISPRi approach for genetic interactions studies, by means of knocking down two genes with two separate but specifically designed guide RNAs, remains a very difficult endeavor. The difficulty stems from at least two issues: 1. CRISPRi-mediated non-essential gene knockdown is often so relatively subtle that it does not result in a phenotype; 2. Polar effects in bacterial systems are well known, which can throw-off any measurement. Consequently, knocking down two genes can simultaneously trigger very different levels of knockdown due to these issues, further making fitness scoring for single and double knockdowns difficult and less accurate. Here, by combining CRISPRi with Tn-Seq we can reduce the variability by having a stably knocked out non-essential gene, and a CRISPRi mediated knocked down essential gene. We show by validating a wide array of interactions that this indeed results in accurately measurable fitness effects. We want to stress that this level of validation is absolutely key in showing the strength and accuracy of our, and for that matter, any approach. Many tools get developed and published with large scale datasets, where validation is often seen as an afterthought. We believe and argue that by validating such a relatively large number of interactions that represent a range of interactions from weak to very strong as well as include a wide-array of functions and pathways we show: 1) how sensitive and versatile our approach is, and 2) That most any genetic interaction, independent of function or pathway can be mapped and scored with CRISPRi-TnSeq.

36Lastly, because we map interactions with essential genes and focus on key pathways that are critical across many different bacterial species, many of our follow-up experiments should carry over to other species. Thus, this work presents a tool, CRISPRi-TnSeq, that can be directly carried over to any species in which Tn-Seq and CRISPRi have been implemented, it presents a strategy to uncover gene function and untangle pathways, as well as new biology on genes that are highly conserved across species.

Reviewer 3 - Comment 3. A major high-level conclusion of the work, is summarized in the abstract: “we identify a set of 17 pleiotropic genes, of which a subset tentatively functions as genetic capacitors, dampening phenotypic outcomes and protecting against perturbations”. They further state that these genes have evolved to “induce robustness into a biological systems to protect against environmental or genetic insults”. They, in fact, claim that the activity of these genes is similar to the buffering properties of the HSP90 chaperone. This is quite a far reach and not at all supported by any of their experimental observations. Let me briefly explain why. The authors find 1,334 significant genetic interactions among 24,000 gene pairs. It is entirely not surprising that one would find a small subset of highly connected nodes in the resulting genetic interaction network. This just reflects a set of recurrent connections between biological processes that are connected to each other at a higher level of organization. There is no need to over-interpret these genes as having a special property of buffering akin to HSP90.

Author response:

We by no means want to oversell our findings and regret if the reviewer finds offense in some of our interpretations. Throughout the manuscript we try to remain objective and let the data speak for itself. We do of course interpret our findings and try to place them into a biological context. The fact that we find a number of highly connected non-essential genes was indeed to be expected, however one still needs to identify and map them, especially since we believe these carry interesting consequences. For instance, 7 of these genes are dominated by negative interactions and their absence sensitizes the organism to a wide variety of (genetic) perturbations, indicating their potential in maintaining a robust system. These interactions and phenotypes reflect the important roles that these non-essential genes play in the genome, which highlights a key strength of CRISPRi-TnSeq. Moreover, we place these genes in context by comparing them with HSP90, a gene that is also able to

buffer against a wide variety of perturbations. Importantly, besides showing that there are key non-essential genes in a network that could even be exploited, we do not make any definitive claims of their exact function. Lastly, we are convinced it is very important that we map these types of interactions as it gives insight into the architecture of a genomic network, what provides stability and what could destabilize a network. However, to make sure our writing is not easily interpreted as overselling we have toned down our wording in the respective section and have for instance removed the capacitor designation.

Reviewer 3 - Comment 4. Design of gRNA, off-target effects and polar effects. The authors should discuss how their gRNAs are designed as off-target effect is always a possibility. Ideally the authors should design at least two gRNAs free of off-target effect for each essential gene they are targeting, and use them as biological replicates of each other. The other concern is the polar effect of gRNAs as they can silence all downstream genes in the targeted operon. This is especially problematic if different genes in the operon encode very different functions. This is the case for a number of genes the authors are studying. For example, *gyrA* (DNA replication) is followed by *srtA* (cell wall) in an operon; *folA* (DNA/metabolism) is followed by multiple other genes including *clpX* (protein degradation) in an operon. This means the authors cannot be certain if the genetic interactions they study are for the gene of interest or a neighboring gene encoding a different function. Authors need to really clarify this.

Author response:

Indeed, in bacterial systems polar effects are a major drawback of CRISPRi mediated target knockdown (also see Comment 2). Therefore, we selected the last gene of the operon as CRISPRi targets when possible. Moreover, all gRNAs were designed and validated for each target in our previous study, Liu et al. Mol Syst Biol 5:931, 2017. In the revised manuscript we discuss the potential impact of polar effects for *gyrA* and *folA* on **P18L500**.

Reviewer 3 - Comment 5. Data reproducibility. The author selected 13 essential genes and studied their interactions with the rest of the genome (ie, 13-vs-all interactions). However, it's not clear to

me how reproducible these datasets are among biological replicates, as they are not presented. Can the authors comment on that?

Author response:

Six biological replicates (6 independently constructed Tn libraries) were grown in each CRISPRi-TnSeq experiment for each essential gene target. Correlation data included in Supplemental File 3 shows a high degree of replicability between libraries. Moreover, Supplemental Table 3 shows average fitness values measured for each gene in each experiment, as well as variation, standard deviation, error, and significance, and also illustrates a very high degree of reproducibility. Lastly, all data are used to evaluate genetic interactions. The large number of replicates thus decreases the chances of false positives.

Reviewer 3 - Comment 6. The validity of the clustering in Fig. 3B looks suspicious as some closely clustered columns don't even look similar (eg, rpoC15 and rpoC30 are right next to each other but the profiles clearly look different. Another example is ftsZ15 and ftsZ30).

Author response:

In panel B, a selected portion of the heat map is presented to highlight gene clusters and potential protein complexes. The entire heatmap is presented in Supplemental Figure 7, which better illustrates conditions that cluster. For instance, it is expected that genes identified at a lower IPTG concentration (e.g., 15 μ M) should also be identified at a higher concentration (e.g., 30 μ M) during target knockdown. These conditions indeed cluster, and this clustering can be better observed in the entire heatmap. To improve clarity, we highlight this in the figure legend of the revised manuscript, and we also point the reader to the Supplemental Figure.

Reviewer 3 - Comment 7. The growth consequences of IPTG-induced knockdown seem to be highly variable in different experiments. For example, there is a large difference in the effect of *pbp2x* knockdown between different figures. For example, compare *pbp2xi*+IPTG between figure panels 5B

and 7B. What is this caused by? Are the authors using different IPTG concentrations in each experiment? In general, these growth curves should have replicates with error bars throughout.

Author response:

In growth studies, both WT and mutant were grown at least as independent triplicates. In each figure, an average of these triplicates is shown. However, since they are so consistent the error bars are very small, and therefore not clearly visible. Indeed, different IPTG concentrations were used in the experiments of figures 5B (20 μ M) and 7B (12.5 μ M). To clarify this, IPTG concentrations are now clearly added to each figure/legend and it is mentioned on **P10L275**.

Reviewer 3 - Comment 8. Throughout the work, the authors should have more nuanced interpretations of experimental observations. For example, bottom of page 13, the authors state: “This suggests that a reduced amount of CozEa results in CtpE dysregulation and changes in intracellular calcium that lead to defects in cell envelope integrity”. What are alternative explanations?

Author response:

Where we believe it was appropriate, we have added alternative explanations.

Reviewer 3 - Comment 9. Microscopic observations are sometimes subtle. The authors could be more quantitative throughout. For example, on page 14, the authors state: “Fluorescence microscopy shows that parC titration results in anucleate cells and uneven chromosome distribution (Fig. 9D), highlighting that chromosome distribution is dependent on decatenation by parC. Deletion of smc results in enhanced chaining (Fig. 9E), indicating that SMC-mediated chromosome segregation affects downstream cell division.”. They could quantify differences in anucleate cells, uneven chromosome distribution, and chaining.

Author response:

We have added quantitative analysis data of microscopy images in the Supplemental Fig. 17 and Supplemental Table 12 and described the results in the context of these quantitative analysis.

Reviewer 3 - Comment 10. At the end of the Discussion section, the authors propose future advances over the work presented here: “By incorporating two guide RNAs into a single cell, a dual-guide CRISPRi tool would have the potential to become a broadly applicable approach for genome-wide genetic interaction mapping. We are actively working on both approaches as we foresee that each will have its own advantages and disadvantages with respect to questions that can be answered as well as their differential ease of implementation and optimization into a new strain or species”. As discussed above, these advances have already been made in the recent literature (Jiang et al, 2020 and Ellis et al, 2023).

Author response:

As discussed above there are critical disadvantages with existing approaches as there will also be disadvantages when two specifically designed guide RNAs are incorporated in a single cell to knockdown two different genes. While we are developing dual-guided CRISPRi employing gRNAs this will not replace CRISPRi-TnSeq because it addresses a different need, for instance it will be easier to implement, and we expect it will remain more sensitive and accurate. We have added such considerations to the Discussion on **P18L518**.

Reviewer 4. Comments to authors:

Reviewer 4 - Comment a. This paper will be very useful for research groups that would like to apply the method to other essential genes of *S. pneumoniae*, or to other organisms. Because of this reason, the methods, analysis and validation should be carefully documented. However, I find the details of the strain construction missing, the analysis tools and rationale not well-explained, and the validation data set small. Only 27 of the >1300 genetic interactions (2%) were shown in a figure. It is likely that there are others that were experimentally checked, but not included on the graph (for example the

data on cosEa and cosEb, and many of the strains in the strain table). A reasonable proof-of-concept validation data set should include about 5 to 10 % of the data set.

Author response:

We have made sure, in the revision, to be as complete as we can in any of the methods, analysis and validation descriptions. Concerning the CRISPRi strains that were used here, these were constructed and validated in our previous study. The details of those strain construction can be found in Liu et al, Mol Syst Biol 5:931, 2017. We have made sure to highlight this more clearly in the revised manuscript. Additionally, we have updated Fig. 4C including the validation results (e.g., *cozEa-cozEb*) and ensured consensus between Fig. 4C and Supplemental Table 1. Lastly, concerning the comment that a decent validation set should contain 5-10% of the data set is a wonderful idea, but especially the 10% is a bit unrealistic with respect to money and time. However, we do agree that a substantial validation set is very important when a new tool, such as CRISPRi-TnSeq is developed and/or implemented/used. Moreover, there is an important trade-off between the size of the validation set and the depth of follow-up experiments. Here we chose to explore a significant number of interactions (32). There was one interaction (*fabH-vraS*) that did not recapitulate the CRISPRi-TnSeq data, which could have been due to the rise of a spontaneous suppressor mutation. We have added this possibility in the revised manuscript on **P18L509**. Furthermore, we spend significant time and effort on follow-up experiments to highlight the diversity of the interactions we picked up and the biological consequences. Importantly, the dataset harbors many more interesting interactions that can fuel a swath of projects, but which would not be realistically pursuable in this project/manuscript.

Reviewer 4 - Comment b. This paper reports a potentially valuable genetic suppressor screen and interesting observations with some of suppressor pairs. This combined technique will advance the knowledge in the field, not just of *Streptococcus pneumoniae*, but also of other organisms. However, the lack of clear labels of supplemental tables and supplemental figures, incomplete explanation of data analysis concerning significant genetic interactions, inconsistency with the data in different figures, and incorrect information on certain gene essentiality (e.g., *rodZ*) make it difficult to understand and evaluate the results.

Author response:

We thoroughly scrutinized the manuscript and believe we have fixed any discrepancies, including those mentioned by the reviewer.

Reviewer 4 - Comment c. I recommend that the authors provide a detailed and careful report on the methods, analysis, and validation results. The current manuscript draws big conclusions on many different aspects of gene interactions without thorough justifications. It may be more effective to concentrate on one topic of interest, such as the *pbp2x* and *pde1* interaction, and provide extra experimental evidence for any model.

Author response:

As described in the comment above, there is an important trade-off to consider when validating a new tool/large dataset. Here we successfully opted to balance the number of validated interactions with additional follow-up experiments, which enabled us to illustrate how diverse the interactions are that can be identified with CRISPRi-TnSeq, which in turn highlights the power and applicability of the approach. We of course do not want to oversell any of our data, and we believe we have been very careful and considerate in the experiments we performed and the conclusions that we draw from them. However, to ensure that our conclusions and discussions are measured and balanced we have made several changes in the revised version of this manuscript that reflect this.

Reviewer 4 - General comments on presentation and formats of supplemental tables and figures

Reviewer 4 - Comment 1. All supplemental tables should have a heading of supplemental table # and a title of the table. Tables should have footnotes to explain the items on the table if needed. The lack of table numbers and titles makes it hard for readers to find the data.

Author response:

The recommended changes have been made.

Reviewer 4 - Comment 2. Make sure all texts and lines in figures and supplemental figures are clearly visible. The black gene labels on dark circle in Fig. 4,5,7, and 9 are hard to read. The lines connecting the gene are too light to distinguish whether they are blue or red, or whether there are intensity differences in the lines. The bands in supplemental figure 1 are not visible.

Author response:

Among other things we improved the readability of Figs. 4, 5, 7 and 9 (revised Fig. 8) and the edges of the network and increased the intensity of the band in Supplemental Figure 1 (revised Supplemental Figure 2).

Reviewer 4 - Comments on presentation of methods to generate the strains for analysis

Reviewer 4 - Comment 3. Supplemental table 1: list genetic backgrounds of strains. Arrange the strain names in a systematic way. The *gfp*-tagged *cosEb* is not listed, and it is in a TIGR4 background. It is not clear how each strain is constructed. The strains ‘adopted’ (line 488) from a previous study should be specified with a reference, and any new constructed strains for this study should be specified as such. Are the adopted strains the same as reported previously, or are they single colony isolated or remade for this study? A good format to use is appendix Table 1 of Liu et al. For example, the genotype for *pbp2x Δpde1* should be XL25 *Δpde1::camR*. Some of the strains (e.g. *fabH vraS*) are not mentioned in the text and should be omitted from the table.

Author response:

In the revised version of Supplemental Table 1, strain names are arranged in a systematic way following the reviewer’s suggestions. The *gfp*-tagged *cozEb* strain is included and *fabH-vraS* is mentioned in the revised manuscript. The strains adopted from a previous study are specified with

the corresponding references. The strains adopted from a previous study are isolated from single colonies, which is highlighted in the methods section.

Reviewer 4 - Comment 4. In Liu et al., the *pbp2xi* strain in the presence of IPTG (Fig.3G) show decrease in OD after 3h and no uptick in growth in 6 h (no data after 6h). However, in Fig. 5C, 6A, and 6B, there is an uptick of growth after 6h. More concerning is the graph in Fig. 7B, the same strain grew much better than shown in 5C, 6AB. Is there a suppressor mutation in this strain? Liu et al mentioned that suppressor mutations are sometimes found in the *dcas9* by sequencing. Because of this concern, growth curves should be shown for each of the 13 parent strains instead of a growth value. It will be best if the parent strains are single-colony-isolated and whole-genome-sequenced before the highly involved Tn-seq experiment.

Author response:

Targeting cell wall genes induces cell lysis, and the rate and extent of lysis increases with IPTG concentration which determines the level of gene repression. In our previous study Liu et al. 2017, we also observed the uptick in growth in the presence of IPTG after 6 hours, which can be found in the supplemental figure EV4 of Liu et al. 2017, which contains full growth curves. For clarity we have now also included full growth curves in Supplemental Table 10. Importantly, the 'uptick' in growth of bacterial CRISPRi strains was already shown in Peters et al. 2016, Cell (<https://doi.org/10.1016/j.cell.2016.05.003>), and can ultimately be expected due to spontaneous suppressor mutations (commonly in *dcas9*) that arise in the population. In the present study, the *pbp2x* strain grew better in Fig. 7B because of a lower IPTG concentration (12.5 μ M). We recognize that these concentrations were not always clearly indicated, which we have corrected in the revised manuscript and we mention in multiple places. Importantly, we are/were vigilant and aware of the possibility of spontaneous mutations arising in our experiments. Therefore, growth of each CRISPRi strain was confirmed with and without IPTG before performing any high-throughput experiments. Single colony isolation is and has been a standard practice in our labs. For instance, for any gene deletion mutant construction single colony selection is required, and strains are confirmed by sequencing. The same stocks were used for growth studies, including those presented in Fig. 2B. Importantly in Fig. 2B, growth is presented as bar plots to compare them to gene repression. The entire growth curves are now presented in Supplemental Figure 1. While we have taken extensive

45care to ensure that spontaneous suppressor mutations did not affect our data, there is always a possibility that these may have arisen during an experiment. Such incidents are not preventable and is not something that whole genome sequencing would prevent, although we did use WGS at several points in time to confirm strain integrity. To further reduce the chances of suppressor mutations taking over the population and being carried forward to follow-up experiments, strains were always grown in rich medium without IPTG, and multiple stock tubes were prepared from the same culture and stored and used for the entire study. Our consistent and reproducible data and validation experiments indicate that we had no suppressor mutations arise that significantly affected this work. To highlight these considerations, we have included a paragraph in the Methods section on **P20L553** that addresses this.

Reviewer 4 - Comment 5. In figure 2B, what is the unit for growth in the y-axis? The values are between 0.1 to 0.5. Is this the maximum OD? This should be clearly written in Materials and Methods, and in the figure legend.

Author response:

The units are OD at 600 nm, which was added to the Y-axis.

Reviewer 4 - Comment 6. The author should refer to supplementary Table 1 instead of supplementary Table 9 for the strains in line 259. Supplemental Table 9 has a list of primers, but the use of these primers are not stated. The use of each primer in supplemental Table 9 should be briefly described in Materials and Methods. It appears that some of the primers are not relevant to this paper, because these genes are not mentioned in the text. They should be deleted in the table. Primers used for qPCR studies should be included in this table.

Author response:

The manuscript is adjusted with respect to the strains in Supplemental Table 1. The use of primers are described in the Methods section, irrelevant primers were removed from Supplemental Table 9 and qPCR primers were added.

Reviewer 4 - Questions and comments related to analysis of the CRISPRi-TnSeq data set

Reviewer 4 - Comment 7. The raw data is presented in Supplemental Table 3, which should have an explanation of the heading in the experimental section. There is a tab named as 'description', which readers may easily miss. Each of the items under description should be clearly explained in Materials and Methods. Explain the headings for each column in supplemental Table 3 to guide the readers through the analysis.

Author response:

In the revised manuscript we have made the suggested changes to Supplemental Table 3 and include a description in the Methods section.

Reviewer 4 - Comment 8. Explain in line 535 to 538 "To make genetic interaction values comparable between different datasets produced for individual CRISPRi targets, individual fitness difference (Δw) and genetic interaction datasets were z-score normalized. All significant genetic interactions with $p\text{-adj} \leq 0.05$ were mapped in a genetic interaction network using Cytoscape". Explain z-score normalized.

Author response:

The rationale for z-score normalization is included on **P22L610**.

Reviewer 4 - Comment 9. In Supplemental Table 3, there were entries with $p\text{-adj} \leq 0.05$, but was considered as false under significance. How are the true and false values assigned? In line 156, it is mentioned that 1334 significant genetic interactions are identified. Are these the ones that have true significance in Supplemental Table 3?

Author response:

To determine whether fitness values are ‘True’, three conditions are imposed, which are the same as a standard Tn-Seq experiment: 1. W_i is calculated from 3 or more data points; 2. The fitness difference between conditions is larger than 0.1 (10%), and 3. the fitness difference is statistically significant ($p\text{-adj} < 0.05$) in a one sample t -test with Bonferroni or Benjamini-Hochberg correction for multiple testing. This description is listed in the Methods section. Indeed, 1334 ‘True’ interactions are identified by CRISPRi-TnSeq experiments, which can be found in Supplemental Table 3.

Reviewer 4 - Comment 10. In Supplemental Table 4, arrange all data in a sheet that is easily understood. A footnote also would be helpful.

Author response:

In the revised manuscript, we have clarified the data in Supplemental Table 4, and included a footnote.

Reviewer 4 - Questions and comments related to growth curves and Δw of constructed strains

Reviewer 4 - Comment 11. The growth profiles of the parent CRISPRi strains used for Tn-seq is very important. As mentioned above, the growth curves of these strains should be shown, similar to those shown in Fig. 3 of Liu et al. instead of the bar graph of a selected strains in Fig. 2B. Explain the unit in Fig. 4c.

Author response:

Growth curves of CRISPRi strains used in the study are included as Supplemental Figure 1.

Reviewer 4 - Comment 12. Line 140, it would be best if the parCi strain is whole-genome sequenced, or sequenced for the *dcas9* region to rule out suppressor mutations.

Author response:

The parCi strain was sequenced and no suppressors were identified. Importantly, as we describe in the main text of the manuscript the observed delayed response upon knockdown, not only confirms that while delayed knockdown is successful and results in a clear phenotype, but this phenotype has also been seen in *Mycobacterium*. Lastly, the *smc*-knock out, combined with parCi without IPTG has no phenotype, while a strong genetic interaction is confirmed in the presence of IPTG, further indicating the validity of the parCi strain.

Reviewer 4 - Comment 13. Line 156: >1300 genetic interactions were identified in the screen. However, only 27 were validated. How were the 27 selected? Is it based on the highest or lowest score? Are there any pairs that do not fit in the graph and data not shown? From Supplemental Table 1, there are more than 27 double strains. Are all the tested ones shown on the figure? In addition, the double *cosEa* strains shown in Fig. 8 and 9 are not included in the plot.

Author response:

As described above, we don't think that the 'only 27 were validated' comment does justice to the extensive validation and follow-up experiments that were performed in this study and that are described in detail in this manuscript. Actually, 32 Interactions were validated which were selected from those with moderate to strong fitness effects, and we aimed to cover a wide variety of interactions and the potential roles that they could describe/be involved in. Only 1 selected interaction (*fabH-vraS*) did not recapitulate the CRISPRi-TnSeq fitness score, which could have

been due to the rise of a spontaneous suppressor mutation. This possibility has been added to the revised manuscript P18L509.

Reviewer 4 - Comment 14. Line 259-262: Explain how growth rates and fitness values are obtained for each constructed double mutant in Materials and Methods. For supplementary table 11: use a footnote to explain how the Δw (CRISPRi-TnSeq) and Δw (mutant growth) are obtained. How many times was Δw determined?

Author response:

This is described in detail in the methods section and more succinct in the results section, moreover a footnote is added to Supplemental Table 11 to explain how Δw (CRISPRi-TnSeq) and Δw (mutant growth) are obtained. Δw was determined at least 3 times from independent experiments.

Reviewer 4 - Comment 15. Supplemental Table 10 showed raw data of time vs. OD of growth curves. A simple table of growth rate, growth yield and calculated fitness values of double vs single mutants should be listed.

Author response:

Growth rate and fitness values are summarized in Supplemental Table 11 for the corresponding growth data presented in Supplemental Table 10.

Reviewer 4 - Essentiality of rodZ

Reviewer 4 - Comment 16. Line 389: As reported recently by Lamanna et al., 2022 in Molecular Microbiology, RodZ is essential in unencapsulated and encapsulated *S. pneumoniae* D39. Constructed

rodZ mutants typically contain suppressor mutations. Therefore, the rodZ knockout reported here likely contains suppressor mutations. Please comment or explain.

Author response:

While we did not identify a suppressor, as explained above it is possible that suppressor mutants arise during experiments. The data from Lamanna et al., 2022 indeed would argue for the possibility of a suppressor mutation in the *rodZ* knock out, albeit a partial suppressor. The *cozEai-rodZ* with IPTG phenotype shows that while a *rodZ* suppressor might be present, it is not capable to rescue the *cozEa* knockdown, instead it confirms the strong negative genetic interaction that likely exists between the two genes. And thus, while a suppressor may be present, it would only be a partial suppressor, i.e., the growth and microscopy data show that *rodZ* is crucial to maintain growth and morphology of *S. pneumoniae* D39 when *cozEa* is knocked down. This suggests that a suppressor mutation would only partially compensate for RodZ's function(s). In the revised manuscript, we discuss the possibility and likelihood of the partial *rodZ* suppressor mutation on **P15L418**.

Reviewer 4 - Other comments:

Reviewer 4 - Comment 17. Line 135, Supplementary Table S11, how is growth measured? What is measured in Fig. 2B? How is ΔW (mutant growth) determined in Supplementary Table S11? Primers used for qPCR should be in a supplementary table.

Author response:

Both wild-type and mutant growth is measured in C+Y medium using an Agilent BioSpa including an automatic plate handler, reader and incubator. The strains were grown exactly the same way as Tn-mutant libraries were grown for the CRISPRi-TnSeq experiments described in the Methods section. In Fig. 2B, growth of CRISPRi mutant libraries without and with different concentrations of IPTG is presented, full growth curves are shown in Supplemental Fig. 1. The details of ΔW (mutant growth) calculations are included in the Methods section of the revised manuscript **P23L652**.

Reviewer 4 - Comment 18. Line 171, refer to Fig. 2C.

Author response:

Adjusted.

Reviewer 4 - Comment 19. Line 189: put boxes over *parC* and *fabH* interaction in supplementary Figure 4 to highlight the information in the text (line 189).

Author response:

We added boxes to Supplemental Figure 4 (revised Supplemental Figure 5) to highlight the datasets of *parC* and *fabH*.

Reviewer 4 - Comment 20. Line 215-229: should have boxes or arrows in Figure 3 to show the points shown in the text line 215-229. It is not clear that there are overlapping targets from this figure. Explain nodes and attributes in Materials and Methods.

Author response:

Boxes are added to Fig. 3B to highlight the overlap. Nodes and attributes are included in the Methods section.

Reviewer 4 - Comment 21. Line 216: Supplemental Table 2 is qPCR results of expression, did not show hierarchical clustering.

Author response:

The correct reference to Supplemental Table 3 is added.

Reviewer 4 - Comment 22. Lines 230-256: 27 interactions were shown in Fig. 4C. It is not clear which 7 interactions were validated in line 233. Supplemental Figure 6 showed only *fabH* and *clpI*, and *pbp2x* and *clpC*. As mentioned above, all 27 growth curves should be shown.

Author response:

Validated pleiotropic interactions are mentioned in Supplemental Table 11. In Supplemental Figure 6 (revised Supplemental Figure 8) a pair of pleiotropic interactions of *clpC* are highlighted. Growth curves are plotted in Supplemental Table 10.

Reviewer 4 - Comment 23. Section starting from line 257, all 27 growth curves should be shown.

Author response:

Growth curves are plotted in Supplemental Table 10.

Reviewer 4 - Comment 24. Line 259: supplemental table 9 are primers, not relevant to the sentence.

Author response:

Table number is corrected to Supplemental Table 1.

Reviewer 4 - Comment 25. Line 266-267: put in references for *fabH* and *plsX* (line 266-267).

Author response:

References are added.

Reviewer 4 - Comment 26. Line 292-302: murM, murN, and pbp2x are not the pairs for validation (not found in Supplemental Table 11). pgdA and pbp2x are also not one of the 27 pairs tested. Also pbp1a, pbp1b, or pbp2a are not tested. These pairs should be tested.

Author response:

We refer to the genetic interactions identified by CRISPRi-TnSeq, not the validation dataset. We have made it clear in the revised manuscript.

Reviewer 4 - Comment 27. Line 305: pbp3 also is not tested with pbp2x. Likewise of pbp2x with sepF or pvaA.

Author response:

As mentioned above, we referred to CRISPRi-TnSeq identified genetic interactions, not the validation dataset. We have mentioned this in the revised manuscript.

Reviewer 4 - Comment 28. Line 322: Supplemental Figure 7 and 8 did not show increased c-di-AMP level in the growth medium. c-di-AMP level was not measured in the growth media.

Author response:

c-di-AMP levels in the growth medium are included in the revised manuscript as Supplemental Figure 9.

Reviewer 4 - Comment 29. Line 342: K concentration is not stated.

Author response:

K⁺ concentration is now included.

Reviewer 4 - Comment 30. Line 343: Fig. 6A shows cell growth, not cellular turgor.

Author response:

Corrected to Fig. 6C.

Reviewer 4 - Comment 31. Line 343-345: Figure 6 and 7, the light blue and light green lines are very confusing. Use different colors. Why does pbp2xi +IPTG show a growth at a later stage in Fig. 6A and 6B right panels? Please clarify.

Author response:

As clarified above, in our previous study Liu et al. 2017, we have also observed the later stage of growth in the presence of IPTG. Such later stage growth of bacterial CRISPRi strains is quite typical and was for instance also observed in Peter et al. 2016, Cell. Specifically, this late-stage growth is likely due to the emergence of spontaneous suppressor mutations (commonly in dcas9). We have highlighted this in the Methods section.

Reviewer 4 - Comment 32. Line 364: Figure 7 :pbp2xi +IPTG grew very well compared to Fig. 5B, Fig. 6A, and Fig. 6B (right panel). This figure casts doubt on pbp2x cozEB negative interaction.

Author response:

55As described above, in the growth experiment of Figure 7 a lower IPTG concentration (12.5 μ M) was used that imposes minimal knockdown. We have clarified IPTG concentrations in all relevant figures/legends throughout the manuscript.

Reviewer 4 - Comment 33. Line 557: should specify that sp1505 is *cozEb*. It also should have been listed in the strain table that the genetic background for this construct is TIGR4. This information should also be in the main text.

Author response:

We specified this in the Methods section and in the main text on **P14L384**. We also included the strains in Supplemental Table 1.

Reviewer 4 - Comment 34. Line 571 When are the cells collected for the *c-di-AMP* assay? After how long in the presence or absence of IPTG? It is mentioned that the cells were grown up to OD of 0.5 to 1.0. However, the growth curve in Fig. 5B, cell did not reach OD of 0.5. Please clarify.

Author response:

Fig. 5B growth curves were performed in 96 well plates using an Agilent BioSpa like all other growth curves. Additionally, *pbp2x* and its *pde1* mutant were grown in 15 mL tubes with and without IPTG for 4.5 hours for the *c-di-AMP* assay due to the requirement of a large number of cells, while OD 600nm was measured in 1mm cuvettes. The scale of growth is different between 96 well plates 15ml tubes explaining the difference in OD reached.

Reviewer 4 - Comment 35. Line 361: *cozEa* and *cozEb* interactions were not confirmed by growth of constructed strains.

Author response:

Indeed, this double mutant was not constructed, partially because the observed strong negative genetic interaction between *cozEa* and *cozEb* was expected based on a previous study (Stamsås et al., 2020 MBio) indicated the functional redundancy between CozEa and CozEb. Thus, since the existing information all but confirms the interaction, it seemed better to validate other interactions, for which less or no information is available.

Reviewer 4 - Comment 36. Supplementary Figure 1: the colors are very faint. Best if a section of the data is shown clearly. Show clearly some of the examples shown in other figures. On the X-axis, list the gene knock down. Examples of the following should be shown: 1(line 181) that *atpF* interaction are mostly positive; (2)(line 188) *parC* and *fabH* knockdown are enriched in DNA repair and lipid metabolism genes; and (3) (line 191) *adk*.

Author response:

We have increased the intensity of the bands and highlighted in Supplemental Figure 1 (revised Supplemental Figure 2) in a network that *atpF* interactions are mostly positive. In Supplemental Figure 5, we highlighted with a red box that DNA repair and lipid metabolism genes are enriched in *parC* and *fabH* knockdown datasets, respectively. In Supplemental Figure 6, we illustrate that purine biosynthesis and degradation pathways are advantageous and disadvantageous, respectively, when *adk* is knocked down.

Decision Letter, first revision:

Message: 23rd February 2024

Dear Professor van Opijnen,

57Thank you for your patience while your manuscript "CRISPRi-TnSeq: A genome-wide high-throughput tool for bacterial essential-nonessential genetic interaction mapping" was under peer-review at Nature Microbiology. It has now been seen by 4 referees, whose expertise and comments you will find at the of this email. You will see from their comments below that while they find your work of interest, some important points are raised. We are very interested in the possibility of publishing your study in Nature Microbiology, but would like to consider your response to these concerns in the form of a revised manuscript before we make a final decision on publication.

In particular, you will see that Reviewer #1 had several concerns over how the total numbers of gene interactions were determined, and how true interactions are distinguished from false positives particularly under high levels of CRISPRi induction, among a number of other comments. Referee #3 also had a number of concerns over the comparison of different methods in the discussion. In light of this reviewer's comments we think that a revised manuscript would need to address these points and tone down these aspects of the discussion accordingly. There were also a number of more minor points outlined among the reviewer reports which we believe are straightforward and should be feasible to address.

If you have not done so already please begin to revise your manuscript so that it conforms to our Article format instructions at <http://www.nature.com/nmicrobiol/info/final-submission/>

The usual length limit for a Nature Microbiology Article is six display items (figures or tables) and 3,000 words. We have some flexibility, and can allow a revised manuscript at 3,500 words, but please consider this a firm upper limit. There is a trade-off of ~250 words per display item, so if you need more space, you could move a Figure or Table to Supplementary Information.

Some reduction could be achieved by focusing any introductory material and moving it to the start of your opening 'bold' paragraph, whose function is to outline the background to your work, describe in a sentence your new observations, and explain your main conclusions. The discussion should also be limited. Methods should be described in a separate section following the discussion, we do not place a word limit on Methods.

Nature Microbiology titles should give a sense of the main new findings of a manuscript, and should not contain punctuation. Please keep in mind that we strongly discourage active verbs in titles, and that they should ideally fit within 90 characters each (including spaces).

We strongly support public availability of data. Please place the data used in your paper into a public data repository, if one exists, or alternatively, present the data as Source Data

or Supplementary Information. If data can only be shared on request, please explain why in your Data Availability Statement, and also in the correspondence with your editor. For some data types, deposition in a public repository is mandatory - more information on our data deposition policies and available repositories can be found at <https://www.nature.com/nature-research/editorial-policies/reporting-standards#availability-of-data>.

Please include a data availability statement as a separate section after Methods but before references, under the heading "Data Availability". This section should inform readers about the availability of the data used to support the conclusions of your study. This information includes accession codes to public repositories (data banks for protein, DNA or RNA sequences, microarray, proteomics data etc...), references to source data published alongside the paper, unique identifiers such as URLs to data repository entries, or data set DOIs, and any other statement about data availability. At a minimum, you should include the following statement: "The data that support the findings of this study are available from the corresponding author upon request", mentioning any restrictions on availability. If DOIs are provided, we also strongly encourage including these in the Reference list (authors, title, publisher (repository name), identifier, year). For more guidance on how to write this section please see: <http://www.nature.com/authors/policies/data/data-availability-statements-data-citations.pdf>

To improve the accessibility of your paper to readers from other research areas, please pay particular attention to the wording of the paper's opening bold paragraph, which serves both as an introduction and as a brief, non-technical summary in about 150 words. If, however, you require one or two extra sentences to explain your work clearly, please include them even if the paragraph is over-length as a result. The opening paragraph should not contain references. Because scientists from other sub-disciplines will be interested in your results and their implications, it is important to explain essential but specialised terms concisely. We suggest you show your summary paragraph to colleagues in other fields to uncover any problematic concepts.

If your paper is accepted for publication, we will edit your display items electronically so they conform to our house style and will reproduce clearly in print. If necessary, we will re-size figures to fit single or double column width. If your figures contain several parts, the parts should form a neat rectangle when assembled. Choosing the right electronic format at this stage will speed up the processing of your paper and give the best possible results in print. We would like the figures to be supplied as vector files - EPS, PDF, AI or postscript (PS) file formats (not raster or bitmap files), preferably generated with vector-graphics software (Adobe Illustrator for example). Please try to ensure that all figures are non-flattened and fully editable. All images should be at least 300 dpi resolution (when figures are scaled to approximately the size that they are to be printed at) and in RGB colour format. Please do not submit Jpeg or flattened TIFF files. Please see also 'Guidelines for Electronic Submission of Figures' at the end of this letter for further detail.

Figure legends must provide a brief description of the figure and the symbols used, within 350 words, including definitions of any error bars employed in the figures.

When submitting the revised version of your manuscript, please pay close attention to our [href="https://www.nature.com/nature-research/editorial-policies/image-integrity">Digital Image Integrity Guidelines](https://www.nature.com/nature-research/editorial-policies/image-integrity). and to the following points below:

Please include a statement before the acknowledgements naming the author to whom correspondence and requests for materials should be addressed.

Finally, we require authors to include a statement of their individual contributions to the paper -- such as experimental work, project planning, data analysis, etc. -- immediately after the acknowledgements. The statement should be short, and refer to authors by their initials. For details please see the Authorship section of our joint Editorial policies at http://www.nature.com/authors/editorial_policies/authorship.html

- * include a point-by-point response to any editorial suggestions and to our referees. Please include your response to the editorial suggestions in your cover letter, and please upload your response to the referees as a separate document.
- * ensure it complies with our format requirements for Letters as set out in our guide to authors at www.nature.com/nmicrobiol/info/gta/
- * state in a cover note the length of the text, methods and legends; the number of references; number and estimated final size of figures and tables
- * resubmit electronically if possible using the link below to access your home page:

*This url links to your confidential homepage and associated information about manuscripts you may have submitted or be reviewing for us. If you wish to forward this e-mail to co-authors, please delete this link to your homepage first.

Please ensure that all correspondence is marked with your Nature Microbiology reference number in the subject line.

Nature Microbiology is committed to improving transparency in authorship. As part of our efforts in this direction, we are now requesting that all authors identified as 'corresponding author' on published papers create and link their Open Researcher and Contributor Identifier (ORCID) with their account on the Manuscript Tracking System (MTS), prior to acceptance. This applies to primary research papers only. ORCID helps the scientific community achieve unambiguous attribution of all scholarly contributions. You can create and link your ORCID from the home page of the MTS by clicking on 'Modify my Springer Nature account'. For more information please visit please visit www.springernature.com/orcid.

We hope to receive your revised paper within three weeks. If you cannot send it within this time, please let us know.

Yours sincerely,

Reviewers Comments:

Reviewer #1 (Remarks to the Author):

- L168-170: It's unclear to me where the reported total number of 1,334 genetic interactions is derived from. Figure S2 is just a heat map, while Supplemental Table 3 is hard to parse in this respect because of the replicate experiments and multiple levels of IPTG for the same query gene. I tried summing values in Supplemental Table 4 to 1,334 without success. Importantly, how do the authors distinguish a true genetic interaction from what they refer to as "false positives" when the cells are on the "verge of collapse" (L625-8) at high induction levels? For example, adk has over 300 genetic interactions (summing both positive and negative) according to Supplemental Table 4, which is over 10% of the nonessential genome. To me, this seems like a very large number and many (most?) of these genes are only distantly connected in cellular function to adk. Why are these adk interactions considered true genetic interactions but those detected at high IPTG concentration with parC are not? How do the authors recommend making this distinction

61moving forward, especially considering their purported statement on L116-7 about providing a “clear experimental and analytical roadmap” for future CRISPRi-TnSeq studies. And while I appreciate the new text at L622-634 that addresses this issue, it doesn’t provide rigorous criteria for when high inducer concentration data should be excluded.

- L64: There are different ways to define genetic interactions including “additive” and “multiplicative” models (see <https://doi.org/10.1073/pnas.071225510>), so the authors should be careful in their working. My understanding is that they’re using a multiplicative calculation, so they should use consistent language throughout the manuscript.

- L143-4: I recommend expanding this sentence slightly and include some statement about the lack of off-target effects detected in the 2017 paper (i.e. move some of the authors’ rebuttal here).

- L939-940: Figure 2F legend states that interactions are shown per 50 kB in the genome, do they mean three 50kB windows around oriC, SPD2014-64, and SPD0229 specifically (this is what’s shown in the plot)?

- L181-2: It’s more accurate to say that on average 65% of the GIs identified at the lower IPTG level were also identified at the higher IPTG level (per the number in Supp. Table 4). Because the reverse certainly isn’t true.

Reviewer #2 (Remarks to the Author):

The authors' revisions and responses to the first round of reviews are thorough and thoughtful.

Reviewer #3 (Remarks to the Author):

The authors’ modifications to the manuscript have improved the interpretation of their scientific findings.

However, the authors’ Discussion misrepresents the most relevant previous work in the field. In particular, the authors claim that: “While, in principle, it is possible to knockdown all genes in a genome by using an adapted crRNA library, due to its random nature and low efficiency, it likely requires hundreds of crRNAs to knockdown a gene”. This is not true. There is no evidence that crRNAs are less efficient than gRNAs. It is well known in the field that gRNAs/crRNAs that target the coding strand are mostly effective while those targeting the non-coding strand are mostly weak/ineffective. This means that roughly half of the generated crRNAs (ie, coding strand) are effective, as has been shown in Jiang et al 2020 (citation 67).

The authors also claim: "different crRNAs targeting the same gene will have different effects on gene knockdown and fitness, making it hard to achieve an accurate measure of the exact effect on fitness". While it is true that individual crRNAs (or gRNAs) likely have variable knockdown efficiency, having many of them increases the sensitivity to detect fitness effects. In fact, most published work utilizing CRISPRi screening design ≥ 10 gRNAs for each gene, and this is true for both prokaryotic and eukaryotic systems. Gene fitness is quantified by averaging all gRNAs specific to that gene. This is the norm rather than an exception in the field, as using multiple gRNAs, rather than only one gRNA, can increase sensitivity and provide higher statistical robustness, in addition to buffering the occasional off-target effect of gRNAs.

The authors also claim: "Third, due to the much higher number of targets (crRNAs) to track, a random crRNA library needs up to several orders of magnitude deeper sequencing to measure the abundance of crRNAs in the library, which can rapidly drive-up costs but also exceed sequencing capabilities". This is inaccurate. Studies that design multiple gRNAs/crRNAs per gene do not require "several orders of magnitude deeper sequencing". Wang et al 2018 (citation 27) stated that "Approximately 10 million reads were collected for each library". Jiang et al 2020 (citation 67) stated that "7–25 million sequencing reads provided sufficient coverage". Rousset et al 2018 (PLOS Genetics) stated that they use "17 million reads per sample" and "4.6 million reads per sample". These sequencing depths are comparable, affordable, and represent the norm in the field.

It is most useful for the authors to differentiate their approach in terms of its niche applications rather than to misrepresent existing alternative approaches. For example, if one has a particular essential gene in mind, the ability to tune its knockdown level by variable induction, and to test its interactions with knockouts of all non-essential genes is a useful capability (as the authors' multiple vignettes demonstrate).

Reviewer #4 (Remarks to the Author):

In general, I am satisfied with the modification of the manuscript. The materials and methods section and supplementary materials are expanded to provide details on the experiments.

I am pleased to see that the dataset for validation was expanded from 27 pairs to 32 pairs, and that the growth profiles are detailed in supplemental table 11.

Below are some simple corrections that the authors should consider:

1. The section 'When cell wall synthesis is compromised, c-di-AMP can maintain growth and morphology by controlling turgor'. The connection between C-di-AMP and turgor has not been demonstrated directly in *Streptococcus pneumoniae* in the literature or in this study. I suggest to delete 'by controlling turgor'.

Page 12, line 338: 'However, increased c-di-AMP levels in the growth medium alone are not sufficient to rescue pbb2x knockdown (Supplemental Fig. 10)'. From the trend in

supplemental fig.10, I would rephrase it to 'increased c-di-AMP levels in the growth medium partially rescue pbp2x knockdown.'

Page 12, line 344 'such as LytB, are altered in Δ pde147. The reference here (47) should instead use publication on *Streptococcus pneumoniae* (Garcia et al., 1999) for lytB, and Bai et al., 2013, which reported on pde1 and chain length in spn.

Page 13, line 359 'Indeed, when the extracellular K⁺ concentration is increased to 100 mM, the effects of Δ pde1 on cellular turgor, and the rescuing effect on pbp2x knockdown is nullified (Fig. 6C)'. Previous version cited Fig. 6A, which I mentioned that it did not show data on cellular turgor. Fig 6C in the current version is also only a cartoon, not data. I suggested that the author should change the sentence to not mention cellular turgor.

Also need to comment on the potassium concentration in the regular media. Should state 'K⁺ concentration is increased from xx to 100 mM'.

Bai et al., 2014 should be cited in relationship to pde1 and effect of potassium in *S. pneumoniae*.

2. In supplemental table 1, the cdai and cdai Δ pde1 strains should be included.

3. The legend to figure 2B should explain

(a) What is being recorded here on the y axis? It is listed as cell density. Is it the cell density after a fixed period of growth?

(b) What are the green, blue and yellow bars?

4. In Fig. 4B, 5A, 7, 8, S2, the dark circles with black letters are still not readable. Change the color of the circle to lighter color.

Garcia, P., Gonzalez, M.P., Garcia, E., Lopez, R., and Garcia, J.L. (1999) LytB, a novel pneumococcal murein hydrolase essential for cell separation. *Molecular microbiology* 31: 1275-1281.

Bai, Y., Yang, J., Eisele, L.E., Underwood, A.J., Koestler, B.J., Waters, C.M., Metzger, D.W., and Bai, G. (2013) Two DHH subfamily 1 proteins in *Streptococcus pneumoniae* possess cyclic di-AMP phosphodiesterase activity and affect bacterial growth and virulence. *Journal of bacteriology* 195: 5123-5132.

Bai, Y., Yang, J., Zarrella, T.M., Zhang, Y., Metzger, D.W., and Bai, G. (2014) Cyclic di-AMP impairs potassium uptake mediated by a cyclic di-AMP binding protein in *Streptococcus pneumoniae*. *Journal of bacteriology* 196: 614-623.

Author Rebuttal, first revision:

Reviewer 1. - Comments:

Reviewer 1 - Comment 1. - L168-170: It's unclear to me where the reported total number of 1,334 genetic interactions is derived from. Figure S2 is just a heat map, while Supplemental Table 3 is hard to parse in this respect because of the replicate experiments and multiple levels of IPTG for the same query gene. I tried summing values in Supplemental Table 4 to 1,334 without success. Importantly, how do the authors distinguish a true genetic interaction from what they refer to as "false positives" when the cells are on the "verge of collapse" (L625-8) at high induction levels? For example, *adk* has over 300 genetic interactions (summing both positive and negative) according to Supplemental Table 4, which is over 10% of the nonessential genome. To me, this seems like a very large number and many (most?) of these genes are only distantly connected in cellular function to *adk*. Why are these *adk* interactions considered true genetic interactions but those detected at high IPTG concentration with *parC* are not? How do the authors recommend making this distinction moving forward, especially considering their purported statement on L116-7 about providing a "clear experimental and analytical roadmap" for future CRISPRi-TnSeq studies. And while I appreciate the new text at L622-634 that addresses this issue, it doesn't provide rigorous criteria for when high inducer concentration data should be excluded.

Author response:

As we mentioned in the methods section the impact on growth due to essential gene-knockdown depends on multiple factors, including stability and abundance of the corresponding gene product. Moreover, the fitness effect of a genetic interaction increases with the extent of growth inhibition. To enable comparisons of genetic interactions across different datasets produced for individual CRISPRi targets, genetic interaction datasets (Δw) were z-score normalized. All significant genetic interactions ($Z > 1.5$ and $Z < -1.5$) with $p\text{-adj} \leq 0.075$ were mapped in a network and presented in Figure 2A. To make significant interactions more visible, the list of 1334 significant genetic interactions is now added to Table 3 in the revised version. With these selection criteria the number of significant genetic interactions for *adk* is 83 (not 300). Importantly, there is only one dataset that was removed from the analyses due to a condition that was too harsh in its selection, which is *parC* grown twice in the presence of 100

65mM IPTG. As we have shown multiple times before, if Tn-Seq library selection is performed in an environment in which the selection pressure is too strong or somehow erratic this can result in Tn-mutants randomly disappearing from the population in relatively large numbers. The population/library is at that point basically pulled through a bottleneck, resulting in a substantial loss of insertion mutants. When this loss is at least 10% or more the library is excluded. The libraries selected in this study have a loss between 0.1 and 4%, except for parC-100 mM, which has a 27% loss, putting it far above the threshold, and warranting its removal. These details are further clarified in the methods section P13L360.

Reviewer 1 - Comment 2. - L64: There are different ways to define genetic interactions including “additive” and “multiplicative” models (see <https://doi.org/10.1073/pnas.071225510>), so the authors should be careful in their working. My understanding is that they’re using a multiplicative calculation, so they should use consistent language throughout the manuscript.

Author response:

The multiplicative model and its definition is used throughout the manuscript.

Reviewer 1 - Comment 3. - L143-4: I recommend expanding this sentence slightly and include some statement about the lack of off-target effects detected in the 2017 paper (i.e. move some of the authors’ rebuttal here).

Author response:

A statement about the lack of off-target effects is made on P4L76 .

Reviewer 1 - Comment 4. - L939-940: Figure 2F legend states that interactions are shown per 50 kB in the genome, do they mean three 50kB windows around oriC, SPD2014-64, and SPD0229 specifically (this is what’s shown in the plot)?

Author response:

The reviewer is correct, which has been added to the legend of the revised supplementary figure 5.

Reviewer 1 - Comment 5. - L181-2: It's more accurate to say that on average 65% of the GIs identified at the lower IPTG level were also identified at the higher IPTG level (per the number in Supp. Table 4). Because the reverse certainly isn't true.

Author response:

This section has been shortened and updated in the manuscript.

Reviewer 2. Comments:

The authors' revisions and responses to the first round of reviews are thorough and thoughtful.

Author response:

We appreciate the reviewer's remarks.

Reviewer 3. Comments:

Reviewer 3 - Comment 1. The authors' modifications to the manuscript have improved the interpretation of their scientific findings.

Author response:

We appreciate the reviewer's acknowledgement.

Reviewer 3 - Comment 2. However, the authors' Discussion misrepresents the most relevant previous work in the field. In particular, the authors claim that: "While, in principle, it is possible to knockdown all genes in a genome by using an adapted crRNA library, due to its random nature and low efficiency, it likely requires hundreds of crRNAs to knockdown a gene". This is not true. There is no evidence that crRNAs are less efficient than gRNAs. It is well known in the field that gRNAs/crRNAs that target the coding strand are mostly effective while those targeting the non-coding strand are mostly weak/ineffective. This means that roughly half of the generated crRNAs (ie, coding strand) are effective, as has been shown in Jiang et al 2020 (citation 67).

The authors also claim: "different crRNAs targeting the same gene will have different effects on gene knockdown and fitness, making it hard to achieve an accurate measure of the exact effect on fitness". While it is true that individual crRNAs (or gRNAs) likely have variable knockdown efficiency, having many of them increases the sensitivity to detect fitness effects. In fact, most published work utilizing CRISPRi screening design ≥ 10 gRNAs for each gene, and this is true for both prokaryotic and eukaryotic systems. Gene fitness is quantified by averaging all gRNAs specific to that gene. This is the norm rather than an exception in the field, as using multiple gRNAs, rather than only one gRNA, can increase sensitivity and provide higher statistical robustness, in addition to buffering the occasional off-target effect of gRNAs.

The authors also claim: "Third, due to the much higher number of targets (crRNAs) to track, a random crRNA library needs up to several orders of magnitude deeper sequencing to measure the abundance of crRNAs in the library, which can rapidly drive-up costs but also exceed sequencing capabilities". This is inaccurate. Studies that design multiple gRNAs/crRNAs per gene do not require "several orders of magnitude deeper sequencing". Wang et al 2018 (citation 27) stated that "Approximately 10 million reads were collected for each library". Jiang et al 2020 (citation 67) stated that "7–25 million sequencing reads provided sufficient coverage". Rousset et al 2018 (PLOS Genetics) stated that they use "17 million reads per sample" and "4.6 million reads per sample". These sequencing depths are comparable, affordable, and represent the norm in the field.

Author response:

68We believe there is a bit of an unfortunate misunderstanding about what we wrote and how the reviewer interpreted things. We take full responsibility for this of course, however, in broad strokes, we stand by what we wrote, library complexity will be (much) higher, sensitivity lower, and tractability more difficult compared to rationally designed sgRNA approaches. However, we think the new crRNA approaches are wonderful and will certainly find great applications. Importantly, we have shortened and clarified this section, as well as simultaneously toned down what we believe are important considerations.

Reviewer 3 - Comment 3. It is most useful for the authors to differentiate their approach in terms of its niche applications rather than to misrepresent existing alternative approaches. For example, if one has a particular essential gene in mind, the ability to tune its knockdown level by variable induction, and to test its interactions with knockouts of all non-essential genes is a useful capability (as the authors' multiple vignettes demonstrate).

Author response:

We apologize if the reviewer felt that we misrepresented things, which was never our intention. While we certainly appreciate all the reviewer's comments we hope they accidentally mislabeled the presented approach as a niche application, which doesn't do justice to the amount of work and effort invested, as well as its presented application/implementation.

Reviewer 4. Comments:

In general, I am satisfied with the modification of the manuscript. The materials and methods section and supplementary materials are expanded to provide details on the experiments.

I am pleased to see that the dataset for validation was expanded from 27 pairs to 32 pairs, and that the growth profiles are detailed in supplemental table 11.

Author response:

We appreciate the reviewer's positive remarks and acknowledgments.

Reviewer 4: Below are some simple corrections that the authors should consider:

Reviewer 4 - Comment 1. The section 'When cell wall synthesis is compromised, c-di-AMP can maintain growth and morphology by controlling turgor'. The connection between C-di-AMP and turgor has not been demonstrated directly in *Streptococcus pneumoniae* in the literature or in this study. I suggest to delete 'by controlling turgor'.

Page 12, line 338: 'However, increased c-di-AMP levels in the growth medium alone are not sufficient to rescue pbp2x knockdown (Supplemental Fig. 10)'. From the trend in supplemental fig.10, I would rephrase it to 'increased c-di-AMP levels in the growth medium partially rescue pbp2x knockdown.'

Page 12, line 344 'such as LytB, are altered in $\Delta pde147$. The reference here (47) should instead use publication on *Streptococcus pneumoniae* (Garcia et al., 1999) for lytB, and Bai et al., 2013, which reported on pde1 and chain length in spn.

Page 13, line 359 'Indeed, when the extracellular K⁺ concentration is increased to 100 mM, the effects of $\Delta pde1$ on cellular turgor, and the rescuing effect on pbp2x knockdown is nullified (Fig. 6C)'. Previous version cited Fig. 6A, which I mentioned that it did not show data on cellular turgor. Fig 6C in the current version is also only a cartoon, not data. I suggested that the author should change the sentence to not mention cellular turgor.

Also need to comment on the potassium concentration in the regular media. Should state 'K⁺ concentration is increased from xx to 100 mM'.

Bai et al., 2014 should be cited in relationship to pde1 and effect of potassium in *S. pneumoniae*.

Author response:

All the suggested changes were made.

Reviewer 4 - Comment 2. In supplemental table 1, the cdai and cdai Δ pde1 strains should be included.

Author response:

Both strains are added to Supplemental Table 1.

Reviewer 4 - Comment 3. The legend to figure 2B should explain

- (a) What is being recorded here on the y axis? It is listed as cell density. Is it the cell density after a fixed period of growth?
- (b) What are the green, blue and yellow bars?

Author response:

- (a) The reviewer is correct. Cell density was measured after \sim 3 hours of growth.
- (b) Green, blue, and yellow bars represent no, low, and high IPTG, respectively.

These details are now included in the legend of revised Figure 1C.

Reviewer 4 - Comment 4. In Fig. 4B, 5A, 7, 8, S2, the dark circles with black letters are still not readable. Change the color of the circle to lighter color.

Author response:

Color gradients represent different biological processes. Therefore, changing colors would not be ideal. Instead, we changed the black letters to a lighter color for the darker nodes.

Reviewer 4 - Comment 5.

Garcia, P., Gonzalez, M.P., Garcia, E., Lopez, R., and Garcia, J.L. (1999) LytB, a novel pneumococcal murein hydrolase essential for cell separation. *Molecular microbiology* 31: 1275-1281.

Bai, Y., Yang, J., Eisele, L.E., Underwood, A.J., Koestler, B.J., Waters, C.M., Metzger, D.W., and Bai, G. (2013) Two DHH subfamily 1 proteins in *Streptococcus pneumoniae* possess cyclic di-AMP phosphodiesterase activity and affect bacterial growth and virulence. *Journal of bacteriology* 195: 5123-5132.

Bai, Y., Yang, J., Zarrella, T.M., Zhang, Y., Metzger, D.W., and Bai, G. (2014) Cyclic di-AMP impairs potassium uptake mediated by a cyclic di-AMP binding protein in *Streptococcus pneumoniae*. *Journal of bacteriology* 196: 614-623.

Author response:

Existing ref 52 (revised 42) is Bai et al., 2014. Other recommended references are included in the revised manuscript.

Manuscript information:

Length of Text – 3452

Length of Methods and Legends – 2661 and 1673

Number of References – 74

Number of Figures – 6

Estimated size of Figures – 4x 1/2 page and 2x 2/3rd page

Decision Letter, second revision:

72Message: Our ref: NMICROBIOL-23051149B

7th May 2024

Dear Dr. van Opijnen,

Thank you for your patience as we've prepared the guidelines for final submission of your Nature Microbiology manuscript, "CRISPRi-TnSeq: A genome-wide high-throughput tool for bacterial essential-nonessential genetic interaction mapping" (NMICROBIOL-23051149B). Please carefully follow the step-by-step instructions provided in the attached file, and add a response in each row of the table to indicate the changes that you have made. Ensuring that each point is addressed will help to ensure that your revised manuscript can be swiftly handed over to our production team.

In recognition of the time and expertise our reviewers provide to Nature Microbiology's editorial process, we would like to formally acknowledge their contribution to the external peer review of your manuscript entitled "CRISPRi-TnSeq: A genome-wide high-throughput tool for bacterial essential-nonessential genetic interaction mapping". For those reviewers who give their assent, we will be publishing their names alongside the published article.

Nature Microbiology offers a Transparent Peer Review option for new original research manuscripts submitted after December 1st, 2019. As part of this initiative, we encourage our authors to support increased transparency into the peer review process by agreeing to have the reviewer comments, author rebuttal letters, and editorial decision letters published as a Supplementary item. When you submit your final files please clearly state in your cover letter whether or not you would like to participate in this initiative. Please note that failure to state your preference will result in delays in accepting your manuscript for publication.

Cover suggestions

COVER ARTWORK: We welcome submissions of artwork for consideration for our cover. For

73more information, please see our guide for cover artwork.

Nature Microbiology has now transitioned to a unified Rights Collection system which will allow our Author Services team to quickly and easily collect the rights and permissions required to publish your work. Approximately 10 days after your paper is formally accepted, you will receive an email in providing you with a link to complete the grant of rights. If your paper is eligible for Open Access, our Author Services team will also be in touch regarding any additional information that may be required to arrange payment for your article.

Please note that *Nature Microbiology* is a Transformative Journal (TJ). Authors may publish their research with us through the traditional subscription access route or make their paper immediately open access through payment of an article-processing charge (APC). Authors will not be required to make a final decision about access to their article until it has been accepted. Find out more about Transformative Journals

<https://mts-nmicrobiol.nature.com/cgi-bin/main.plex?el=A4Cg5BoD4A5nwu6J6A9ftdAOW8RnVPb2D2cCRFYyyI9QZ>

Best regards,

Reviewer #1:

Remarks to the Author:

The authors adequately addressed by concerns. I look forward to seeing this combined tn-seq and crispri approach applied to other bacteria.

Reviewer #4:

Remarks to the Author:

The current manuscript is acceptable for publication if the author take note of the following.

'The reviewer is correct. Cell density was measured after ~ 3 hours of growth.

(b) Green, blue, and yellow bars represent no, low, and high IPTG, respectively.

These details are now included in the legend of revised Figure 1C'

Legend to fig. 1c still do not have the information on the length of time of growth when the OD is recorded in the graph.

Final Decision Letter:

Message 12th June 2024

:

Dear Professor van Opijnen,

I am pleased to accept your Article "CRISPRi-TnSeq: A genome-wide high-throughput tool for bacterial essential-nonessential genetic interaction mapping" for publication in Nature Microbiology. Thank you for having chosen to submit your work to us and many congratulations.

You may wish to make your media relations office aware of your accepted publication, in case they consider it appropriate to organize some internal or external publicity. Once your paper has been scheduled you will receive an email confirming the publication details. This

75is normally 3-4 working days in advance of publication. If you need additional notice of the date and time of publication, please let the production team know when you receive the proof of your article to ensure there is sufficient time to coordinate. Further information on our embargo policies can be found here:

<https://www.nature.com/authors/policies/embargo.html>

Please note that *Nature Microbiology* is a Transformative Journal (TJ). Authors may publish their research with us through the traditional subscription access route or make their paper immediately open access through payment of an article-processing charge (APC). Authors will not be required to make a final decision about access to their article until it has been accepted. Find out more about Transformative Journals

With kind regards,